# DistMatch: Adaptive Binning via Distribution Matching for Robust Sequential Conformal Prediction

Enver Menadjiev [* 1]    Jihyeon Seong [* 1]    Jisu Yeo [1]    Jaesik Choi [1 2]

## Abstract

Sequential conformal prediction (CP) provides valid uncertainty quantification under the assumption of residual exchangeability. However, this assumption is often violated in real-world time series due to temporal dependencies and distributional shifts. While recent methods attempt to approximate exchangeability through reweighting, identifying optimal weights remains an open challenge. To address this limitation, we propose Dist-Match[1], a binning-based method that recursively partitions residuals within a binary tree using the Kolmogorov–Smirnov (KS) statistic. We theoretically show that this partitioning induces approximately exchangeable leaves, thereby avoiding the need for reweighting. By applying quantile regression with online updates within each leaf, DistMatch enables locally adaptive inference and improves robustness to distributional shifts. Extensive experiments demonstrate that DistMatch outperforms existing sequential CP methods.

## 1. Introduction

Conformal Prediction (CP) provides valid uncertainty quantification for arbitrary predictive models under the assumption of exchangeability (Vovk et al., 2005; Shafer & Vovk, 2008). Exchangeability implies that the joint distribution of the data is invariant under permutations, which guarantees that prediction intervals attain the desired coverage on unseen samples (Kim et al., 2020). In sequential settings, however, distributional shifts often violate this assumption, motivating existing sequential CP methods to approximate exchangeability via residual-based reweighting.

*Equal contribution [1]Korea Advanced Institute of Science and Technology, South Korea [2]INEEJI, South Korea. Correspondence to: Jaesik Choi <jaesik.choi@kaist.ac.kr>.

*Proceedings of the 43rd International Conference on Machine Learning*, Seoul, South Korea. PMLR 306, 2026. Copyright 2026 by the author(s).

[1]https://github.com/enver1323/dist_match_conformal

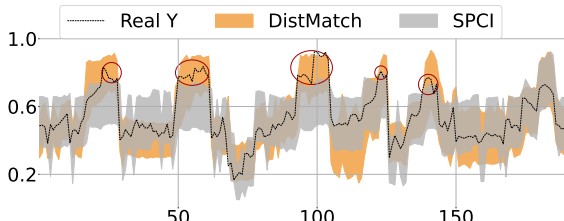

*Figure 1.* The figure compares sequential CP methods on the Solar dataset under extreme distribution shifts. DistMatch effectively captures these shifts through local adaptive quantile estimation via distribution-based binning, as highlighted by the red circles, in contrast to SPCI, which relies on sliding-window model updating.

Reweighting schemes assign data-dependent, continuous weights to residuals; however, they face a fundamental challenge in accurately estimating these weights, which can distort the empirical residual distribution. For instance, temporal reweighting methods (Barber et al., 2023) track global discrepancy by upweighting recent residuals, discarding informative early samples and amplifying weight misspecification under abrupt shifts. To address this limitation, similarity-based retrieval approaches (Auer et al., 2023; Lee et al., 2025) emphasize past residuals that are deemed similar to an observed input; nevertheless, their performance remains highly sensitive to retrieval quality, as even small similarity estimation errors can assign non-negligible weight to unrelated or noisy samples.

Binning-based approaches, by contrast, offer a promising alternative by grouping similar samples into independent subgroups without retrieval. Unlike reweighting-based methods, binning preserves empirical distributions by avoiding explicit weighting schemes, provided that samples within each bin are approximately exchangeable. However, ensuring approximate local exchangeability hinges on defining an appropriate splitting criterion, which is nontrivial in practice. To this end, we employ the Kolmogorov–Smirnov (KS) statistic (Massey, 1951), a nonparametric measure that partitions samples purely based on distributional similarity, without relying on temporal assumptions such as global stationarity (Gong & Huang, 2012). As illustrated in Figure 2, the KS statistic effectively distinguishes highly skewed distributions, a common characteristic of real-world residuals.

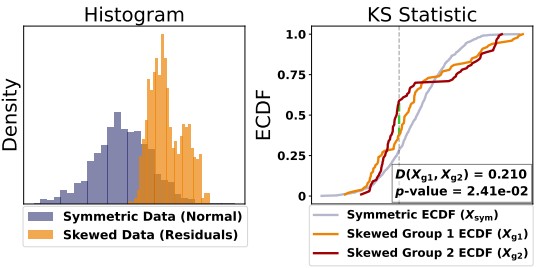

*Figure 2.* While real-world residuals exhibit highly skewed distributions (left subplot), the KS statistic successfully distinguishes between two different, highly skewed distributions (right subplot).

Motivated by these observations, we propose DistMatch, a binary tree–based framework that recursively bins residuals using the KS statistic, yielding approximately exchangeable leaves with bounded distribution. Specifically, DistMatch represents the local distributional context at each time step using a residual patch $\tilde{\varepsilon}_t$ paired with its target residual $\varepsilon_{t+1}$, capturing the relationship between recent and future residual behavior, and performs partitioning based on the patch similarity. When distributional shifts are detected, the tree expands to form branches that group samples with tightly bounded empirical distributions. Finally, at each leaf, DistMatch applies quantile regression with independent online updates, enabling locally adaptive and robust inference.

We introduce a notion of approximate local exchangeability based on the KS statistic, admitting samples whose residual distributions satisfy an exchangeability bound and conferring robustness to mild shifts. In contrast, Total Variation (TV)-based approaches (Barber et al., 2023) bound only global discrepancies, yielding weaker guarantees. Under this definition, we prove that DistMatch attains target-level approximate local exchangeability despite patch-level training, ensuring valid coverage. Under severe shifts, robustness is sustained by diverse split anchors from an ensemble of subsampled trees and online leaf-wise quantile regression updates that adapt local quantiles without altering tree structure. Consequently, DistMatch achieves state-of-the-art performance even under extreme shifts (Figure 1). To our knowledge, DistMatch is the first binning-based sequential CP framework with guarantees grounded in approximate local exchangeability. Our contributions are as follows:

- We propose DistMatch, a binning-based method for sequential CP that replaces reweighting with KS-based distribution matching.

- We theoretically show that DistMatch induces approximately exchangeable leaves, thereby guaranteeing valid conformal coverage.

- We show that DistMatch consistently outperforms existing baselines across diverse real-world datasets.

## 2. Related Work

Recent sequential CP methods attempt to approximate exchangeability through various reweighting strategies when exact exchangeability is violated (Stankeviciute et al., 2021). Temporal reweighting schemes (Tibshirani et al., 2019; Barber et al., 2023) rely on likelihood ratios or kernel-based decay to track global discrepancy while prioritizing recent samples; however, inaccurate tracking of distributional changes can lead to substantial coverage loss. To overcome these limitations, similarity- and copula-based methods (Auer et al., 2023; Lee et al., 2025; Sun & Yu, 2024) assign continuous weights via learned similarity or explicit dependency models; however, misspecification of either propagates into distorted quantile estimates. Methods that avoid explicit reweighting, such as ensemble- and update-based approaches (Xu & Xie, 2021; 2023), use sliding windows or adaptive retraining; however, truncating historical data may impair coverage under long-range dependence. Although some methods explicitly address distributional shifts via auxiliary or labeled data, or global correction mechanisms (Jeary et al., 2024; Xu et al., 2025; Kasa et al., 2025), such requirements limit their practical applicability.

Probabilistic forecasting offers an alternative for uncertainty quantification, yet most methods rely on modeling assumptions that can limit robustness under distribution shift. Gaussian Processes (GP) (Williams & Rasmussen, 1995) assume Gaussian priors that may be sensitive to heavy-tailed data, while Mixture Density Networks (MDN) (Bishop, 1994) depend on parametric mixtures whose calibration can degrade under degeneracy. Monte Carlo Dropout (MCD) (Gal & Ghahramani, 2016) relies on a variational approximation that may become miscalibrated over shifted time. Consequently, their coverage is not guaranteed under severe shifts.

## 3. Preliminary

### 3.1. Exchangeability

Consider a multivariate time series $X = \{(x_t, y_t)\}_{t=1}^T$, where each input $x_t \in \mathbb{R}^d$ is a $d$-dimensional vector, $y_t$ is a scalar response, and $T$ denotes the total number of time steps. Let an arbitrary point prediction model $\phi$ produce outputs $\hat{y}_t = \phi(x_t)$ and residuals be defined as $\varepsilon_t = y_t - \hat{y}_t$.

**Definition 3.1** (Exchangeability of Residuals). The sequence of residuals $\varepsilon_1, \varepsilon_2, \ldots, \varepsilon_T$ is said to be *exchangeable* if, for any permutation $\pi$ of the indices $\{1, 2, \ldots, T\}$, the joint distribution remains invariant as,

$$P(\varepsilon_1, \varepsilon_2, \ldots, \varepsilon_T) = P(\varepsilon_{\pi(1)}, \varepsilon_{\pi(2)}, \ldots, \varepsilon_{\pi(T)}). \quad (1)$$

### 3.2. Kolmogorov–Smirnov Statistic

The Kolmogorov–Smirnov (KS) statistic measures the distance between two empirical distribution functions.

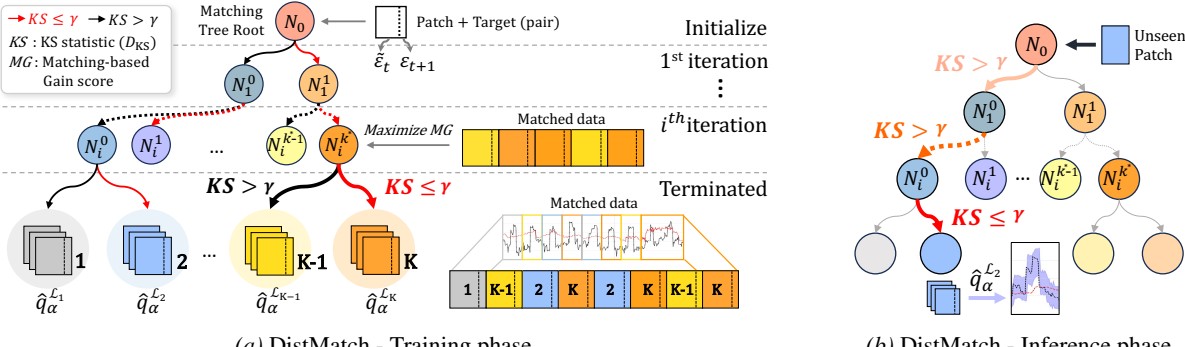

*(a)* DistMatch - Training phase

*(b)* DistMatch - Inference phase

*Figure 3.* DistMatch operates in two phases. During training, it learns a split anchor ($s$) by maximizing the MG score over all candidate patches in the calibration set, thereby identifying an optimal split node $N(s, \mathcal{T}_R, \mathcal{T}_L)$. Sample pairs are routed to the left or right child depending on whether their distributions match or differ. During inference, only the learned split anchors are used to assign unseen patches to the appropriate leaf, where the quantile regressor $\hat{q}_\alpha^{\mathcal{L}(\cdot)}$ estimates target residuals based on the local distribution within that leaf. Similar colors (e.g., red–orange–yellow) denote samples from the same parent, reflecting shared underlying distributions.

**Definition 3.2** (Two-Sample KS Statistic). Let $S(\varepsilon) = \{\varepsilon_1, \ldots, \varepsilon_t\}$ be a finite sample of residuals. Its empirical cumulative distribution function (ECDF) is defined as

$$F_{S(\varepsilon)}(x) = \frac{1}{|S(\varepsilon)|} \sum_{t \in \mathcal{I}(S(\varepsilon))} \mathbb{1}\{\varepsilon_t \leq x\}, \qquad (2)$$

where $\mathcal{I}(\cdot)$ denotes corresponding index set. Then, the *KS statistic (distance)* between $S(\varepsilon_i)$ and $S(\varepsilon_j)$ is defined as

$$D_{\text{KS}}(S(\varepsilon_i), S(\varepsilon_j)) = \sup_{x \in \mathbb{R}} \left| F_{S(\varepsilon_i)}(x) - F_{S(\varepsilon_j)}(x) \right|. \quad (3)$$

Several studies (Gong & Huang, 2012; Gibson et al., 2018) advocate using the raw KS statistic as a direct measure of distributional similarity, avoiding reliance on $p$-values and temporal assumptions such as global stationarity. Compared to alternatives such as the Wasserstein distance (WD) (Villani, 2009), KL divergence (Kullback & Leibler, 1951), or mutual information (MI) (Shannon, 1948)—which often require density estimation, shared support, or incur high computational cost—the KS statistic is a density-free criterion that enables efficient and stable discrimination between residuals under skewed distributions (Appendix C).

## 4. Method

### 4.1. Split Conformal Prediction

Given a multivariate time series $X$, our objective is to construct the narrowest possible prediction region $\hat{\mathcal{C}}_t^\alpha(x_{t+1})$ such that $y_{t+1} \in \hat{\mathcal{C}}_t^\alpha(x_{t+1})$ with probability at least $1 - \alpha$. The marginal and conditional coverage, as defined by Vovk et al. (2005), are respectively given by:

$$\mathbb{P}\left(y_{t+1} \in \hat{\mathcal{C}}_t^\alpha(x_{t+1})\right) \geq 1 - \alpha \qquad (4)$$

$$\mathbb{P}\left(y_{t+1} \in \hat{\mathcal{C}}_t^\alpha(x_{t+1}) \mid x_{t+1}\right) \geq 1 - \alpha. \qquad (5)$$

Originally introduced by Papadopoulos et al. (2007), the split conformal prediction (SCP) framework partitions the data into two disjoint subsets $X = \left\{ X^{(\text{Tr})}, X^{(\text{Cal})} \right\}$, where $X^{(\text{Tr})}$ is used to train a predictor $\phi \colon \mathbb{R}^d \to \mathbb{R}$ and $X^{(\text{Cal})}$ is used for calibration. After training $\phi$ on $X^{(\text{Tr})}$, nonconformity scores are computed on $X^{(\text{Cal})}$ as univariate absolute residuals: $|\varepsilon_t| = |y_t - \phi(x_t)|$, which are then used to construct prediction intervals around the predictions from $\phi$.

However, SPCI (Xu & Xie, 2023) uses signed residuals to preserve directional information, allowing the model to learn asymmetric conditional quantiles. Following this setting, DistMatch adopts signed residuals to capture the full distributional structure, where $\{\varepsilon_t\}_{t=1'}^T \in Z^{(c)}$ and $Z^{(c)}$ denotes the set of signed residuals from $X^{(\text{Cal})}$. Let $\hat{q}_{1-\alpha}$ denote the $(1 - \alpha)$ empirical quantile of the residuals. The resulting CP interval for a new input $x_{t+1}$ is given by:

$$\hat{\mathcal{C}}_t^\alpha(x_{t+1}) = \left[ \hat{y}_{t+1} + \hat{q}_{\alpha/2}, \ \hat{y}_{t+1} + \hat{q}_{1-\alpha/2} \right]. \qquad (6)$$

### 4.2. DistMatch

We propose DistMatch, a novel binning method that groups samples with bounded distributions from heterogeneous data. As illustrated in Figure 3, DistMatch first segments residuals into patches ($\tilde{\varepsilon}_t$) and pairs them with their corresponding target residuals ($\varepsilon_{t+1}$), forming patch-target pairs ($\tilde{\varepsilon}_t, \varepsilon_{t+1}$). These pairs are then passed through a binary matching tree, where recursive KS statistic–based matching scores partition them into approximately exchangeable leaves. Finally, each leaf is equipped with a quantile regressor that performs locally adaptive online quantile estimation.

#### 4.2.1. PATCHING

Given $Z^{(c)}$, we employ a patching technique (Nie et al., 2023) that enables KS-based comparisons over finite residual sets (Definition 3.2). We define each patch as $\tilde{\varepsilon}_t =$

$\{\varepsilon_{t-w+1}, \varepsilon_{t-w+2}, \ldots, \varepsilon_t\}$, using a sliding window of size $w \in \mathbb{Z}^+$. We then construct the input as a sequence of pairs $\mathcal{D} = \{(\tilde{\varepsilon}_t, \varepsilon_{t+1})\}_{t=w}^{T-1}$, where indices beyond $T - 1$ correspond to unseen samples. Patching captures local temporal dependence, ensuring convergence and target-level approximate local exchangeability (Section 5).

### 4.2.2. MATCHING TREE

Given $\mathcal{D}$, DistMatch builds a binary tree grouping similar residual patches. In the absence of labels, we define a matching-based gain score (MG) to guide tree construction, which quantifies how many samples are distributionally similar to a given candidate anchor $\tilde{\varepsilon}_i$, as defined by

$$\text{MG}(\tilde{\varepsilon}_i) = \sum_{j \in \mathcal{I}(\mathcal{D})} \mathbb{1}\{D_{\text{KS}}(\tilde{\varepsilon}_i, \tilde{\varepsilon}_j) \leq \gamma\}, \qquad (7)$$

where $D_{\text{KS}}$ denotes the KS statistic between residual patches following Eq. (3) and $\gamma \leq 0.1$ is a data-dependent threshold that controls the trade-off between within-leaf sample size and within-leaf empirical bounds.

As described in Algorithm 1, DistMatch grows a binary tree by recursively partitioning $\mathcal{D}$ at each split node $N$. At a given node, a split anchor $s$ is selected to maximize the number of samples whose residual patches lie within a KS distance of $\gamma$ from the anchor. Samples matched to the anchor are routed to the right child $\mathcal{T}_R$, while the remaining samples are routed to the left child $\mathcal{T}_L$. The recursion terminates either when 1) no further splitting is necessary, or 2) no split satisfying the minimum leaf size $n_{\min}$ is possible, yielding leaves $\{\mathcal{L}\}_{k=1}^{K}$ that satisfy approximate local exchangeability (Section 5). Note that DistMatch bootstraps $B$ trees with ratio $\theta$ to enhance robustness under shifts.

### 4.2.3. QUANTILE REGRESSION MODELS

Given $\{\mathcal{L}\}_{k=1}^{K}$, DistMatch independently applies a quantile regressor within each leaf. We adopt a Quantile Regression Forest (QRF) (Meinshausen, 2006), which preserves the local empirical distribution through tree-based partitioning (Appendix E.2.1). As illustrated in Figure 4, it defines the conditional empirical distribution of the target residual as

$$\hat{F}_{\mathcal{L}_{(\cdot)}}(y \mid \tilde{\varepsilon}_t) = \sum_{t \in \mathcal{I}(\mathcal{L}_{(\cdot)})} w_t(\tilde{\varepsilon}_t) \mathbb{1}\{\varepsilon_{t+1} \leq y\}, \quad (8)$$

where $w_t(\tilde{\varepsilon}_t)$ satisfies $\sum_{t \in \mathcal{I}(\mathcal{L}_{(\cdot)})} w_t(\tilde{\varepsilon}_t) = 1$. Then, $\tau$-quantile is given by

$$\hat{q}_\tau^{\mathcal{L}_{(\cdot)}}(\tilde{\varepsilon}_t) = \inf\{y \in \mathbb{R} : \hat{F}_{\mathcal{L}_{(\cdot)}}(y \mid \tilde{\varepsilon}_t) \geq \tau\}. \quad (9)$$

### 4.2.4. INFERENCE

At inference time, as shown in Figure 3b, the learned tree routes the unseen residual patch $\tilde{\varepsilon}_T$ by recursively comparing its distribution to the split anchors $(s_{(\cdot)})$ at each internal

---

**Algorithm 1** DistMatch Training

---

**Require:** Dataset $\mathcal{D} = \{(\tilde{\varepsilon}_t, \varepsilon_{t+1})\}_{t=w}^{T-1}$, KS threshold $\gamma$
**Ensure:** Matched Binary Tree ($\{N\} \in \mathcal{T}$)
1: **function** DistMatch($\mathcal{D}$)
2:    **for** $i = 1$ **to** $|\mathcal{D}|$ **do**
3:       $\mathcal{I}_i \leftarrow \{j \in \{1, \ldots, |\mathcal{D}|\} : \text{solve Eq.(7)}\};$
4:    **end for**
5:    $i^\star \leftarrow \arg\max_{1 \leq i \leq |\mathcal{D}|} |\mathcal{I}_i|;$
6:    $s \leftarrow \tilde{\varepsilon}_{i^\star};$
7:    **if** $|D| - |I_{i^\star}| < n_{\min}$ **or** $|\mathcal{I}_{i^\star}| = |\mathcal{D}|$ **then**
8:       **Return** *Leaf node ($\mathcal{L} = \mathcal{D}$)*
9:    **end if**
10:   $\mathcal{D}_R \leftarrow \{(\tilde{\varepsilon}_j, \varepsilon_{j+1}) : j \in \mathcal{I}_{i^\star}\};$
11:   $\mathcal{D}_L \leftarrow \{(\tilde{\varepsilon}_{j'}, \varepsilon_{j'+1}) : j' \notin \mathcal{I}_{i^\star}\};$
12:   $\mathcal{T}_R \leftarrow$ **DistMatch**($\mathcal{D}_R$);
13:   $\mathcal{T}_L \leftarrow$ **DistMatch**($\mathcal{D}_L$);
14:   **Return** $N(s, \mathcal{T}_R, \mathcal{T}_L)$
15: **end function**

---

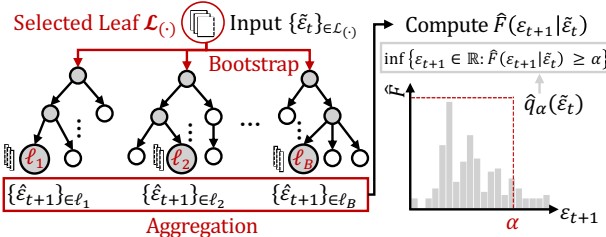

*Figure 4.* Given any arbitrary leaf $\mathcal{L}_{(\cdot)}$ from DistMatch, the QRF builds a forest by bootstrapping the input patches $\{\tilde{\varepsilon}_t\}_{\in \mathcal{L}_{(\cdot)}}$ and applies a quantile operator based on conditional ECDFs to estimate the target residual $\{\varepsilon_{t+1}\}_{\in \mathcal{L}_{(\cdot)}}$ via binning.

node ($N_{(\cdot)}$). Once a leaf node ($\mathcal{L}_{(\cdot)}$) is reached, the stored residual pairs are passed to a QRF, which is trained on $\{\tilde{\varepsilon}_t, \varepsilon_{t+1}\}_{\in \mathcal{L}_{(\cdot)}}$ and then used to estimate $\hat{\varepsilon}_{T+1}$. We further adopt the adaptive refinement strategy proposed by Xu & Xie (2023) to reduce unnecessary interval width under asymmetric or moderately shifted residual distributions, solving:

$$\delta^\star = \arg\min_{\delta \in [0, \alpha]} |\hat{q}_{1-\alpha+\delta}(\tilde{\varepsilon}_T) - \hat{q}_\delta(\tilde{\varepsilon}_T)|. \quad (10)$$

$$\hat{\mathcal{C}}_T^\alpha(x_{T+1}) = [\hat{y}_{T+1} + \hat{q}_{\delta^\star}(\tilde{\varepsilon}_T), \ \hat{y}_{T+1} + \hat{q}_{1-\alpha+\delta^\star}(\tilde{\varepsilon}_T)]. \quad (11)$$

Finally, we append the true target residual $\varepsilon_{T+1}$ to the assigned leaf ($\mathcal{L}_{(\cdot)}$), enabling continuous online updates of the QRF. By retraining the QRF after each new instance, this procedure continuously refines the local residual distributions within each leaf (Appendix E.2.6).

## 5. Theory

We analyze the distributional structure induced by recursive KS-based splitting and show that it suffices for valid coverage under the local temporal assumption.

## 5.1. Approximate Local Exchangeability

We first introduce KS-based approximate local exchangeability, which requires only within-leaf distributional similarity rather than global exchangeability.

**Definition 5.1** ($\gamma^\star$-Approximate Local Exchangeability). Let $\mathcal{L}_{(k^\star)}$ denote the leaf selected by $\tilde{\varepsilon}_T$. The target residuals $\{\varepsilon_{t+1}\}_{t \in \mathcal{I}(\mathcal{L}_{(k^\star)})}$ are $\gamma^\star$-approximately locally exchangeable with respect to the unseen target $\varepsilon_{T+1}$ if

$$\max_{t \in \mathcal{I}(\mathcal{L}_{(k^\star)})} D_{\mathrm{KS}}(P_{t+1}, P_{T+1}) \leq \gamma^\star, \tag{12}$$

where $P_{t+1}$ and $P_{T+1}$ denote the distributions of $\varepsilon_{t+1}$ and $\varepsilon_{T+1}$, respectively. Notably, $\gamma^\star = 0$ recovers the i.i.d. case, while $\gamma^\star > 0$ allows mild shifts between samples.

Definition 5.1 is well aligned with quantile estimation, as it directly controls discrepancies in cumulative distribution functions (Appendix B.4). We further introduce local temporal assumptions to ensure that KS-based splitting meaningfully separates heterogeneous patch distributions into approximately exchangeable leaves at the target level.

**Assumption 5.2** (Local Stationarity and Patch Mixing). The observed process is locally stationary in the sense of (Dahlhaus, 1997), and the predictor $\phi$ is a smooth mapping, thus the residual process $\{\varepsilon_t = y_t - \phi(x_t)\}$ inherits local stationarity. Consequently, the patch sequence $\{\tilde{\varepsilon}_t\}$ is $\beta$-mixing with coefficients $\{\beta_m\}_{m \geq 1}$ satisfying $\sum \beta_m^{1/2} < \infty$. In addition, there exists a conditional distribution kernel $\mathcal{K}$ with $\varepsilon_{t+1} \mid \tilde{\varepsilon}_t \sim \mathcal{K}(\tilde{\varepsilon}_t)$ that is KS-stable: for any patch distributions $\tilde{P}, \tilde{Q}$,

$$D_{\mathrm{KS}}\left(\tilde{P}\mathcal{K}, \tilde{Q}\mathcal{K}\right) \leq C\, D_{\mathrm{KS}}(\tilde{P}, \tilde{Q}), \tag{13}$$

where $\tilde{P}\mathcal{K}$ denotes the marginal distribution of $\varepsilon_{t+1}$ when $\tilde{\varepsilon}_t \sim \tilde{P}$.

*Remark* 5.3. Local stationarity (Dahlhaus, 1997) is standard for sliding-window analyses of nonstationary time series and provides both physical and theoretical grounding for the $\beta$-mixing condition. Under the Dahlhaus (1997) framework, the joint distribution of two well-separated, non-overlapping windows factorizes asymptotically as $T \to \infty$; thus, cross-window cumulants vanish with separation. In finite samples, this manifests as the joint distribution of two distant patches approaching the product of their marginals, ruling out infinitely long memory and supporting the decay of $\beta_m$ as $m$ grows. For overlapping patches generated by a sliding window of size $w$, dependence between $\tilde{\varepsilon}_{t_1}$ and $\tilde{\varepsilon}_{t_2}$ is mechanically induced by shared elements only when $m = |t_1 - t_2| \leq w$; once $m > w$ the patches are disjoint and the residual dependence reflects solely the underlying process's decaying memory, which diminishes under local stationarity in a manner consistent with $\beta$-mixing.

## 5.2. Convergence of the Matching Tree

We now formalize the convergence behavior of the KS-based matching tree at the patch level.

**Theorem 5.4** (Convergence). *Suppose Assumption 5.2 holds. For any initial calibration set, Algorithm 1 satisfies:*

**1. (Approximate local exchangeability)** *Any matched terminal leaf $\mathcal{L}_{(\cdot)}$ satisfies*

$$\max_{i,j \in \mathcal{I}(\mathcal{L}_{(\cdot)})} D_{\mathrm{KS}}(P_i, P_j) \leq 2\gamma. \tag{14}$$

**2. (Minimum leaf size)** *All leaves satisfy $|\mathcal{L}_{(\cdot)}| \geq n_{\min}$.*

*Proof.* Appendix B.1. □

*Remark* 5.5. Samples that do not satisfy the matching criterion are routed to a leftmost leaf. Empirically, these samples constitute around or less than 1% of the calibration set (Appendix E.2.4).

Theorem 5.4 shows that terminal leaves exhibit uniformly bounded patch-level distributions, while unmatched branches continue recursive splitting until they either produce a matched leaf or reach the minimum leaf size $n_{\min}$. Consequently, the tree construction KS threshold theoretically bounds the within-leaf KS diameter up to $2\gamma$, ensuring patch-level approximate local exchangeability.

## 5.3. Coverage Guarantee via Distributional Stability

We now provide a theoretical coverage guarantee for DistMatch based on a matching tree constructed from a fixed calibration set with adaptive quantile refinement.

**Theorem 5.6** (Coverage Guarantee). *Suppose Assumption 5.2 holds. Let $\mathcal{L}_{(k^\star)}$ be the leaf selected by the unseen patch $\tilde{\varepsilon}_T$ among $\{\mathcal{L}_k\}_{k=1}^K$, and let $\hat{q}_{1-\alpha+\delta^\star}^{\mathcal{L}_{(k^\star)}}$ be any consistent estimator of the $(1-\alpha+\delta^\star)$-conditional quantile constructed within $\mathcal{L}_{(k^\star)}$, with $\delta^\star \in [0, \alpha]$ from Eq. (10). Define the within-leaf target divergence*

$$m_{\mathcal{L}_{(k^\star)}} := \max_{t \in \mathcal{I}(\mathcal{L}_{(k^\star)})} D_{\mathrm{KS}}(P_{t+1}, P_{T+1}) \leq 2C\gamma + \mathcal{O}(\sigma_{\mathrm{mix}}), \tag{15}$$

*where $C = \mathcal{O}(1)$ depends only on the regularity of the conditional kernel $\mathcal{K}$ and $\sigma_{\mathrm{mix}} = \mathcal{O}\left(\sum_{m \geq 1} \sqrt{\beta_m}\right)$. Then, with probability at least $1 - \xi$,*

$$\left|\mathbb{P}\left(\varepsilon_{T+1} \leq \hat{q}_{1-\alpha+\delta^\star}^{\mathcal{L}_{(k^\star)}}\right) - (1-\alpha)\right| \leq m_{\mathcal{L}_{(k^\star)}} + \varrho_{\mathcal{L}_{(k^\star)}}(\xi)$$
$$+ \delta^\star + 1/|\mathcal{L}_{(k^\star)}|. \tag{16}$$

*where $\varrho_{\mathcal{L}_{(k^\star)}}(\xi) = \mathcal{O}\left(\sigma_{\mathrm{mix}}\sqrt{\log(1/\xi)/|\mathcal{L}_{(k^\star)}|}\right)$.*

*Proof.* Appendix B.2. □

*Remark* 5.7. Theorem 5.6 decomposes the coverage error into a bias term $m_{\mathcal{L}}$, which decreases with finer partitioning (smaller $\gamma$), and a variance term $\varrho_{\mathcal{L}}$, which increases due to finite-sample effects as $|\mathcal{L}|$ shrinks. Given the KS threshold $\gamma$, DistMatch balances this trade-off by stopping splits when further reductions in $m_{\mathcal{L}}$ are outweighed by increases in $\varrho_{\mathcal{L}}$, where $\gamma$ is optimized using only the calibration set.

*Remark* 5.8. Definition 5.1 characterizes exchangeability at the *target level*, while Algorithm 1 routes based on *patch-level* distributions. Under $\beta$-mixing, delayed dependence decays with patch size $w$, so the future target residual distribution evolves smoothly relative to the patch distribution. The KS bound thus enforces uniform closeness between the unseen and leaf-wise target distributions, yielding only a controlled additive coverage error. The $\mathcal{O}(\sigma_{\mathrm{mix}})$ term, which quantifies the propagation error from temporal dependence, diminishes as $\{\beta_m\}$ decay faster.

### 5.4. Bootstrapping and Online Updating for Robustness

Building on the fixed tree structure, we finally show that bootstrap averaging extends the marginal guarantee of Theorem 5.6 to the online sequence. Beyond Assumption 5.2, we additionally require exponential mixing decay, bootstrap routing coverage at a nontrivial probability level, and stability of quantile averaging across the ensemble.

**Assumption 5.9** (Fast Mixing Rate). The $\beta$-mixing coefficients from Assumption 5.2 decay exponentially:

$$\beta_m \leq C_\beta e^{-\lambda m}, \tag{17}$$

for some constants $\lambda > 0$ and $C_\beta > 0$.

**Assumption 5.10** (Bootstrap Diversity). Let $\{\mathcal{T}^{(b)}\}_{b=1}^B$ denote a bootstrap ensemble of matching trees constructed with sampling ratio $\theta \in (0,1]$. For any unseen patch $\tilde{\varepsilon}_T$, with probability at least $p_{\min}$ over the bootstrap randomness and tree construction, there exists at least one tree that routes $\tilde{\varepsilon}_T$ to a matched leaf:

$$\mathbb{P}\left(\exists\, b \in \{1,\ldots,B\} \text{ s.t. } D_{\mathrm{KS}}(\tilde{\varepsilon}_T, s^{(b)}) \leq \gamma\right) \geq p_{\min}, \tag{18}$$

where $s^{(b)}$ denotes the split anchor associated with the selected leaf in tree $b$.

**Assumption 5.11** (Quantile Averaging Stability). Let $\{q^{(b)}\}_{b=1}^B$ denote the $\tau$-quantile estimators obtained from the bootstrap ensemble, and define the averaged quantile

$$\bar{q} = \frac{1}{B} \sum_{b=1}^B q^{(b)}. \tag{19}$$

There exists a constant $\kappa_{\mathrm{avg}} = O(B^{-1})$ such that

$$|\mathbb{P}(Y \leq \bar{q}) - \tau| \leq \frac{1}{B} \sum_{b=1}^B \left|\mathbb{P}(Y \leq q^{(b)}) - \tau\right| + \kappa_{\mathrm{avg}}. \tag{20}$$

Assumptions 5.9–5.11 ensure 1) a sharpened concentration rate, 2) that most unseen patches reach a matched right leaf, and 3) controlled aggregation across the ensemble.

**Proposition 5.12** (Online Robust Coverage via Bootstrap Averaging). *Let* $\{\mathcal{T}^{(b)}\}_{b=1}^B$ *be a bootstrap ensemble of matching trees on a fixed calibration set, and define the ensemble quantile* $\hat{q}_\tau^{\mathrm{ens}}(\tilde{\varepsilon}_t) := B^{-1} \sum_{b=1}^B \hat{q}_\tau^{\mathcal{L}_t^{(b)}}(\tilde{\varepsilon}_t)$, *where* $\mathcal{L}_t^{(b)}$ *is the leaf selected by tree* $b$ *for* $\tilde{\varepsilon}_t$. *For* $t = T, \ldots, T + n_{\mathrm{test}} - 1$, *form*

$$\hat{\mathcal{C}}_t^\alpha = \left[\hat{y}_{t+1} + \hat{q}_{\delta^\star}^{\mathrm{ens}}(\tilde{\varepsilon}_t),\; \hat{y}_{t+1} + \hat{q}_{1-\alpha+\delta^\star}^{\mathrm{ens}}(\tilde{\varepsilon}_t)\right], \tag{21}$$

*with* $\delta^\star \in [0, \alpha]$ *given by Eq. (10), and after observing* $\varepsilon_{t+1}$ *append it to* $\mathcal{L}_t^{(b)}$ *and retrain. Under Assumptions 5.2 and 5.9–5.11, with probability at least* $1 - \xi$ *over the calibration sampling and online process,*

$$\frac{1}{n_{\mathrm{test}}} \sum_{t=T}^{T+n_{\mathrm{test}}-1} \mathbb{K}\{y_{t+1} \in \hat{\mathcal{C}}_t^\alpha\} \;\geq\; 1 - \alpha - \epsilon_{\mathrm{online}}, \tag{22}$$

*where*

$$\begin{aligned}
\epsilon_{\mathrm{online}} &\leq (1 - p_{\min})m_{\max} + 2C\gamma + O(\sigma_{\mathrm{mix}}) \\
&\quad + O\left(\sqrt{\frac{\log(n_{\mathrm{test}}/\xi)}{|\mathcal{L}|_{\min}}}\right) \\
&\quad + O(\sqrt{\beta_w}) + O(B^{-1}) + \delta^\star.
\end{aligned} \tag{23}$$

*with* $|\mathcal{L}|_{\min} := \min_{b,t} |\mathcal{L}_t^{(b)}|$, $m_{\max} := \max_{b,t} m_{\mathcal{L}_t^{(b)}}$, $\sigma_{\mathrm{mix}} = O\left(\sum_{m \geq 1} \sqrt{\beta_m}\right)$, *and* $w$ *the patch window size.*

*Proof.* Appendix B.3. $\qquad\square$

*Remark* 5.13. Proposition 5.12 clarifies why DistMatch maintains coverage to unseen samples without assuming periodicity in the underlying process. Under a mild shift, Definition 5.1 admits any calibration sample whose residual distribution lies within $\gamma$ of the unseen patch; thus, homogeneous right leaves absorb a gradual distributional shift between calibration and unseen sets. Under severe shift, two complementary mechanisms preserve validity. First, the bootstrap ensemble of $B$ trees with diverse split anchors (Assumption 5.10) routes each unseen patch to a matched right leaf in at least one tree with probability $p_{\min}$; the resulting fallback error scales as $(1 - p_{\min})m_{\max}$ in $\epsilon_{\mathrm{online}}$ and decays as $B$ grows. Second, the online QRF updating appends each observed residual to its assigned leaf and refits the leaf-wise quantile estimator. Together, these mechanisms preserve validity over the online sequence without modifying the tree structure.

In summary, DistMatch's KS-based binning overcomes the limitations of reweighting with theoretical guarantees on both convergence and coverage: samples are treated uniformly within each leaf, weight estimation is replaced by a nonparametric binary routing rule, and coverage error is bounded transparently by the threshold $\gamma$.

*Table 1.* Quantitative evaluation on five real-world datasets at $\alpha = 0.1$ using RF point predictor. Best widths and Winkler (Win.) scores are in **bold**; methods with miscoverage above $0.25\alpha$ are greyed out. DistMatch consistently outperforms other baselines across datasets, achieving the smallest widths and Winkler scores among methods that satisfy the target coverage.

| UC Model | | DistMatch (Ours) | KOWCPI (2025) | HopCPT (2023) | SPCI (2023) | EnbPI (2021) | NexCP (2023) | CP (2021) | Copula (2024) |
|---|---|---|---|---|---|---|---|---|---|
| Elec. | Win. ↓ | $\mathbf{1.97^{\pm 0.01}}$ | $2.44^{\pm 0.00}$ | $3.35^{\pm 0.21}$ | $2.54^{\pm 0.06}$ | 3.36 | 3.51 | 3.93 | 3.77 |
| | Cov. ↑ | $0.92^{\pm 0.00}$ | $0.90^{\pm 0.00}$ | $0.91^{\pm 0.02}$ | $0.90^{\pm 0.00}$ | 0.85 | 0.92 | 0.97 | 0.96 |
| | Width ↓ | $\mathbf{0.27^{\pm 0.00}}$ | $0.38^{\pm 0.00}$ | $0.50^{\pm 0.05}$ | $0.28^{\pm 0.00}$ | 0.45 | 0.57 | 0.71 | 0.67 |
| Solar | Win. ↓ | $\mathbf{1.54^{\pm 0.01}}$ | $2.07^{\pm 0.00}$ | $1.90^{\pm 0.13}$ | $1.98^{\pm 0.15}$ | 2.55 | 3.74 | 3.56 | 3.54 |
| | Cov. ↑ | $0.91^{\pm 0.00}$ | $0.93^{\pm 0.00}$ | $0.92^{\pm 0.01}$ | $0.85^{\pm 0.01}$ | 0.88 | 0.89 | 0.86 | 0.88 |
| | Width ↓ | $\mathbf{60.00^{\pm 0.40}}$ | $109.31^{\pm 0.00}$ | $97.04^{\pm 8.21}$ | $47.36^{\pm 5.14}$ | 112.57 | 215.67 | 167.91 | 187.69 |
| Wind | Win. ↓ | $\mathbf{2.15^{\pm 0.01}}$ | $3.54^{\pm 0.00}$ | $3.72^{\pm 0.31}$ | $2.19^{\pm 0.00}$ | 4.37 | 3.98 | 4.40 | 4.10 |
| | Cov. ↑ | $0.90^{\pm 0.00}$ | $0.89^{\pm 0.00}$ | $0.90^{\pm 0.01}$ | $0.83^{\pm 0.00}$ | 0.85 | 0.89 | 0.74 | 0.82 |
| | Width ↓ | $\mathbf{69.04^{\pm 0.03}}$ | $139.25^{\pm 0.00}$ | $151.66^{\pm 12.82}$ | $63.14^{\pm 0.18}$ | 145.63 | 171.67 | 147.57 | 159.00 |
| META | Win. ↓ | $\mathbf{0.12^{\pm 0.00}}$ | $0.61^{\pm 0.00}$ | $0.25^{\pm 0.08}$ | $\mathbf{0.12^{\pm 0.00}}$ | 0.19 | 0.25 | 0.29 | 0.27 |
| | Cov. ↑ | $0.90^{\pm 0.02}$ | $0.91^{\pm 0.00}$ | $0.83^{\pm 0.10}$ | $0.96^{\pm 0.00}$ | 0.83 | 0.90 | 0.98 | 0.96 |
| | Width ↓ | $\mathbf{4.28^{\pm 0.03}}$ | $26.27^{\pm 0.00}$ | $8.08^{\pm 0.65}$ | $5.00^{\pm 0.04}$ | 5.77 | 11.25 | 14.77 | 12.96 |
| NVDA | Win. ↓ | $\mathbf{0.49^{\pm 0.00}}$ | $0.72^{\pm 0.00}$ | $1.59^{\pm 0.01}$ | $1.05^{\pm 0.00}$ | 1.01 | 1.42 | 2.94 | 2.94 |
| | Cov. ↑ | $0.90^{\pm 0.00}$ | $0.89^{\pm 0.00}$ | $0.75^{\pm 0.00}$ | $0.78^{\pm 0.01}$ | 0.79 | 0.88 | 0.90 | 0.89 |
| | Width ↓ | $\mathbf{2.71^{\pm 0.01}}$ | $4.42^{\pm 0.00}$ | $6.93^{\pm 0.01}$ | $3.23^{\pm 0.01}$ | 3.53 | 8.98 | 16.30 | 14.52 |

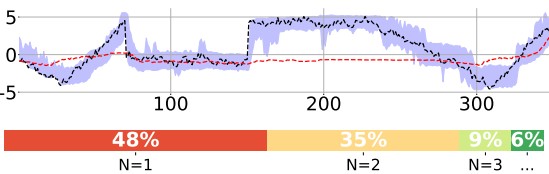

*Figure 5.* Study on synthetic data with two predefined clusters. DistMatch correctly recovers the two underlying clusters within the synthetic data, with only about 17% of noisy samples forming minor auxiliary leaves, demonstrating its robustness.

# 6. Experiments

## 6.1. Experimental Setup

**Datasets:** To quantitatively evaluate DistMatch, we utilize five real-world datasets: electricity consumption (*Elec.*), solar radiation (*Solar*), wind energy (*Wind*), and two stock datasets, META and NVDA. Additional experiments on heavy-tailed datasets are provided in the Appendix E.2.3.

**Point Predictor:** We use a Random Forest (RF) (Breiman, 2001) in Table 1 for fair comparison, and an LSTM (Hochreiter & Schmidhuber, 1997) for Table 2, since the corresponding baselines require a sequential backbone. Additional results using LGBM and TCN across various significance levels are provided in Appendix E.2.9.

**Hyperparameters:** We optimize hyperparameters via greedy search for all experiments, extending Auer et al. (2023). All results are averaged over four random seeds, with standard deviation reported.

**Evaluation Metrics:** 'Cov.' is the target coverage, 'Width' denotes interval size, and 'Win.' is the Winkler score penalizing miscoverage while accounting for width. Following

Auer et al. (2023), methods with miscoverage above $0.25\alpha$ are deemed invalid, regardless of interval width.

## 6.2. Synthetic Dataset

To evaluate DistMatch's clustering performance, we generate two-segmented synthetic time series by combining a piecewise deterministic trend with regime-dependent Gaussian noise (Appendix E.1.2). As shown in Figure 5, DistMatch correctly identifies the two major groups, which jointly contain about 83% of the samples, while assigning noisy points to small auxiliary leaves. This flexibility—recovering dominant clusters without forcing a fixed number of leaves—enhances robustness under real-world noise and distributional complexity.

## 6.3. Result Analysis

As shown in Table 1, DistMatch achieves the narrowest prediction intervals while maintaining valid coverage across all five datasets, whereas several baselines fail to meet the coverage target. HopCPT attains valid coverage but yields wider intervals due to the use of absolute residuals and misspecified retrieval, while SPCI and EnbPI suffer coverage degradation under extreme shifts from over-reliance on recent residuals. KOWCPI and CopulaCPTS, which are sensitive to model misspecification, as well as exchangeability-based methods such as NexCP and CP, preserve coverage but incur wider intervals than DistMatch. This gap is most highlighted on highly non-stationary datasets such as META and NVDA. Qualitatively (Figure 6a), DistMatch produces adaptive, narrow intervals, whereas HopCPT, EnbPI, and NexCP generate overly wide and nearly uniform intervals, and KOWCPI and SPCI struggle under shifts. With an

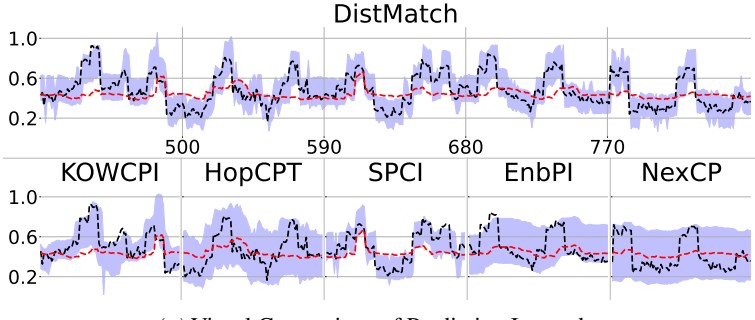

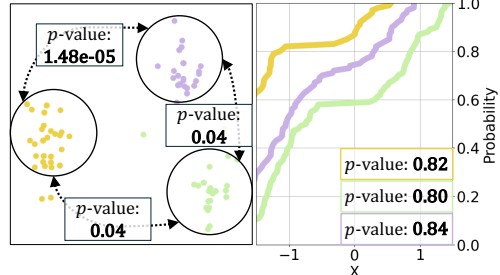

*(a)* Visual Comparison of Prediction Intervals.

*(b)* Clustered Results and *p*-value from DistMatch.

*Figure 6.* Figure (a) shows prediction interval visualizations for DistMatch and baselines on the *Elec.* dataset (Table 1). Black indicates ground truth, red points predictions, and purple estimated quantile intervals. Figure (b) presents projected residuals and ECDFs from three representative DistMatch leaf nodes, along with KS *p*-values for within- and cross-node comparisons on *Elec.* dataset (Table 1). DistMatch achieves narrower prediction intervals while capturing extreme shifts via distributionally bounded leaves.

*Table 2.* Comparison of DistMatch with probabilistic forecasting models at $\alpha = 0.1$ using an LSTM predictor. We report the coverage averaged over the three datasets (Avg. Cov.) and the Winkler score (Win.) for each dataset. Methods with miscoverage exceeding $0.25\alpha$ are greyed out. DistMatch achieves the best performance among methods satisfying the target coverage.

| Dataset | | Elec. | Solar | Wind |
|---|---|---|---|---|
| Metric | Avg. Cov. ↑ | Win. ↓ | Win. ↓ | Win. ↓ |
| DistMatch | 0.90 | $\mathbf{1.15^{\pm 0.02}}$ | $\mathbf{1.87^{\pm 0.11}}$ | $\mathbf{2.05^{\pm 0.01}}$ |
| MDN (1994) | 0.67 | $1.55^{\pm 0.34}$ | $1.53^{\pm 0.25}$ | $3.08^{\pm 0.34}$ |
| Gauss (1995) | 0.60 | $2.02^{\pm 0.39}$ | $1.53^{\pm 0.23}$ | $3.59^{\pm 0.15}$ |
| MCD (2016) | 0.53 | $1.82^{\pm 0.10}$ | $3.35^{\pm 0.20}$ | $3.55^{\pm 0.05}$ |

LSTM backbone (Table 2), DistMatch continues to satisfy the target coverage with a low Winkler score, whereas probabilistic forecasting baselines lose coverage and produce substantially wider intervals, inducing a higher Winkler score than DistMatch and limiting practical applicability.

To further validate DistMatch, we assess its ability to cluster distributionally bounded groups within each leaf node. Figure 6b visualizes three representative leaf nodes from the *Elec.* dataset (Table 1), where residuals are projected into two dimensions using Neighborhood Component Analysis (NCA) (Goldberger et al., 2004). We hypothesize that residuals within a node follow similar distributions (*p*-value $> 0.05$), whereas those across nodes differ significantly (*p*-value $\leq 0.05$). The results confirm this hypothesis, demonstrating that DistMatch's binning strategy effectively clusters residuals by distributional similarity.

### 6.4. Ablation Studies

**Hyperparameters:** Figure 7 examines the robustness of DistMatch under varying hyperparameters. We first examine the KS threshold $\gamma$, which controls tree construction and governs the bias–variance trade-off in Theorem 5.6. We set $\gamma \leq 0.1$ since larger values violate approximate exchangeability; within this range, DistMatch yields stable

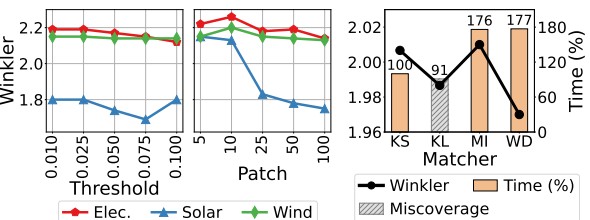

*(a)* Ablation study on KS threshold $\gamma$, patch length $w$, and distribution maching criteria.

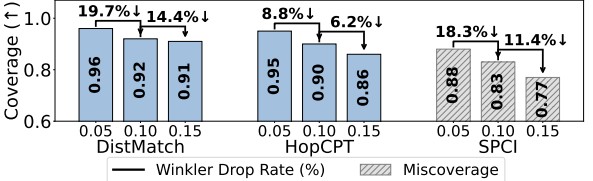

*(b)* Ablation study on significance level $\alpha$.

*Figure 7.* (a) presents an ablation study of the main hyperparameters of DistMatch on the *Elec.*, *Solar*, and *Wind* datasets under same setting with Table 1. (b) presents an ablation study on $\alpha$ using the RF predictor on the *Elec.* dataset, analyzing target coverage and the Winkler score drop rate.

Winkler scores by balancing the trade-off. For the patch length $w$, we adopt $w = 100$ as a conservative default for Table 1 to satisfy Theorem 5.6. In practice, smaller values are also viable: *Solar* benefits from $w = 25$, while other datasets are robust across a wide range. Among the considered distribution-matching criteria, the KS statistic achieves the most favorable Winkler scores, averaged across three datasets, while maintaining computational efficiency. In contrast, MI and WD incur substantially higher computational costs than KS, while KL fails to meet the target coverage. Finally, with respect to the significance level $\alpha$, DistMatch consistently attains the target coverage while resulting in the greatest decrease in the Winkler score. In summary, DistMatch performs robustly within the suggested parameter range, which is also theoretically grounded.

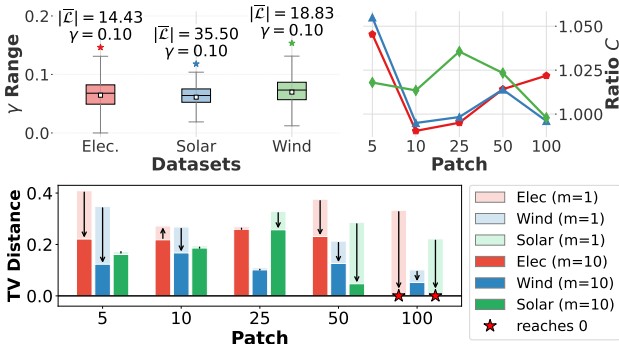

*Figure 8.* Top-left subplot shows the within-leaf KS distance at the patch-level with fixed $\gamma = 0.1$, while top-right reports the patch-to-target smoothness ratio $C$. Bottom-subplot presents $\beta$-mixing assumption validity by showing that the TV distance decays as the lag $m$ increases.

**Theoretical Validity:** Figure 8 examines the theoretical validity of DistMatch. We first empirically validate Theorem 5.4 by verifying that DistMatch maintains within-leaf KS distances below $2\gamma$ while preserving a sufficient average within-leaf sample size ($|\bar{\mathcal{L}}|$). As shown in Figure 8, we set $\gamma = 0.1$, under which residual patch distributions within each leaf are uniformly bounded by 0.2. The average leaf size also remains above 10 even when we set $n_{\min} = 0$ for the stopping criterion as default, since the matching criterion (Eq. (7)) naturally prevents extremely small leaves from forming. This is because a sample can only be routed to a leaf if it satisfies the distributional match with the anchor, which intrinsically limits fragmentation and keeps finite-sample errors controlled even for small $\gamma$. Next, we analyze the patch-to-target smoothness ratio $C$ in Assumption 5.2 using the KS distance by varying the patch length $w \in \{5, 10, 25, 50, 100\}$. As shown in Figure 8, a significant drop occurs when increasing the patch length from $w = 5$ to $w = 10$, and beyond this point, $C$ fluctuates due to a trade-off between patch-to-target smoothness and within-patch diversity, but converges at $w = 100$. This empirically supports the claim that $C = O(1)$ (Theorem 5.6).

Finally, for the patch-wise $\beta$-mixing assumption, we conduct nonparametric independence tests based on Total Variation (TV) distances. Specifically, to capture the full distribution of residual patches, each patch is mapped to its sorted quantile vector, serving as a sufficient statistic for the ECDF. We then discretize this high-dimensional space into a finite set of states using K-means clustering (Lloyd, 1982). Based on this discretization, we compute the TV distance between the empirical joint distribution and the product of marginals of two patches across various lags $m$. As shown in Figure 8, the TV distance generally decreases as the lag $m$ increases across various datasets and patch sizes; in particular, at $w = 100$ it converges to 0, providing empirical support for the $\beta$-mixing condition.

*Table 3.* Computational Complexity. Here $n$ is the calibration set size, $w$ is patch window size, and for HopCPT, $d$ is hidden dimension, $K$ is number of hops, $E_g$ is the number of graph edges, and $t$ is the current online step. Amortized costs assume $T \gg n$ and DistMatch's one-time calibration cost becomes negligible.

| Methods | Calibration | Per-step (online) |
|---|---|---|
| SPCI | $O(nw \log w)$ | $O((n + t)w \log w)$ |
| HopCPT | $O(E_g d + w \log w)$ | $O(Kd^2 + \text{retrain})$ |
| DistMatch | $O(n^2 w \log n)$ | $O(w \log n)$ |
| *Amortized cost over $T$ online steps ($T \gg n$)* | | |
| SPCI | $O(T^2 w \log w)$ | |
| HopCPT | $O(TKd^2 + T \cdot \text{retrain})$ | |
| DistMatch | $O(Tw \log n)$ | |

### 6.5. Computational Complexity

The computation of DistMatch consists of two components: the matching tree and the leaf-wise QRF. First, patch-level ECDFs are precomputed once in $O(nw \log w)$ and cached, so that each subsequent KS comparison runs in $\mathcal{O}(w)$ without additional sorting. Tree construction is then dominated by KS-based matching over these cached representations, with worst-case complexity $O(n^2 w \log n)$ as shown in Table 3. In Appendix E.2.5, however, we show that DistMatch maintains robust performance under limited calibration sizes, enabling efficient deployment with small calibration budgets. After training, online inference requires only tree traversal and leaf-wise updates, yielding $O(Tw \log n)$ amortized complexity over $T$ steps. We further compare the fixed and retrained variants of DistMatch under online deployment in Appendix E.2.7, and find that retraining is unnecessary. This is a key advantage of DistMatch: a single training step suffices to maintain efficient inference throughout the online sequence, unlike SPCI and HopCPT, which incur cumulative superlinear online costs.

The QRF component introduces an additional per-update cost $O(|\mathcal{L}| \log |\mathcal{L}|)$ when new residuals are added to a leaf, where $|\mathcal{L}|$ is the leaf size; Appendix E.2.6 further analyzes the trade-off between computational cost and alternative leaf storage strategies, showing that DistMatch remains robust under small leaf sizes without accumulating leaf samples.

## 7. Conclusion

We introduce DistMatch, a novel binning-based sequential CP method that recursively partitions residuals into distributionally bounded groups using the KS statistic. We theoretically establish that the resulting leaf-wise groups satisfy approximate exchangeability, which is sufficient to guarantee valid conformal coverage. Extensive experiments on real-world datasets demonstrate that DistMatch consistently outperforms existing sequential CP methods. As future work, we plan to extend DistMatch with adaptive thresholding and online learning mechanisms.

## Impact Statement

This paper presents work aimed at advancing the field of sequential conformal prediction. Our contributions promote reliable uncertainty quantification under distribution shifts via binning, which is relevant to high-stakes domains such as finance, where calibrated prediction intervals can support safer decision-making. We do not foresee negative societal impacts specific to this work, beyond the general caution that the theoretical coverage guarantees hold only insofar the stated assumptions are satisfied.

## Acknowledgement

This work was supported by Institute for Information & communications Technology Planning & Evaluation (IITP) grant funded by the Korea government (MSIT) (RS-2019-II190075, Artificial Intelligence Graduate School Program (KAIST); No. RS-2024-00509258, AI Guardians: Development of Robust, Controllable, and Unbiased Trustworthy AI Technology; No. RS-2024-00457882, AI Research Hub Project), and by the InnoCORE program of the Ministry of Science and ICT (N10250156).

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

# Appendix

## Contents

# A. Notation

*Table 4.* Notation summary.

| Symbol | Description |
|---|---|
| **Data and predictor** | |
| $X = \{(x_t, y_t)\}_{t=1}^T$ | Original dataset with input–target pairs. |
| $t, T$ | Data index and the total number of samples. |
| $x_t \in \mathbb{R}^d$ | $d$-dimensional input at index $t$. |
| $y_t \in \mathbb{R}$ | Scalar target at index $t$. |
| $\phi$ | Point predictor. |
| $\hat{y}_t = \phi(x_t)$ | Prediction at index $t$. |
| $(\cdot), (\star)$ | Arbitrary index (any index) and optimal index. |
| **Residual and exchangeability** | |
| $\varepsilon_t = y_t - \hat{y}_t$ | Residual at index $t$. |
| $P, Q$ | Probability distributions. |
| $\pi$ | A permutation of indices. |
| **KS statistic** | |
| $S$ | A set of scalar residual samples. |
| $\mathcal{I}$ | Index set of samples. |
| $\mathbb{1}$ | Indicator function. |
| $F(\cdot)$ | CDF (cumulative distribution function). |
| $D_{\text{KS}}(\cdot, \cdot)$ | Kolmogorov–Smirnov statistic (distance). |
| **Conformal prediction** | |
| $\mathbb{P}$ | Probability. |
| $\mathcal{C}$ | Conformal coverage. |
| $\alpha$ | Significance level. |
| $Z^{(c)}$ | Calibration set for signed residual, where $\varepsilon \in Z^{(c)}$. |
| **DistMatch** | |
| $(\tilde{\varepsilon}_t, \varepsilon_{t+1})$ | Pair of patch residual and target residual. |
| $w$ | Patch (window) size. |
| $\gamma$ | KS statistic threshold (tree construction threshold). |
| MG | Matching count-based gain score. |
| $\mathcal{D}$ | Node dataset in the tree, where $\{(\tilde{\varepsilon}_t, \varepsilon_{t+1})\} \in \mathcal{D}$. |
| $\mathcal{T}$ | DistMatch tree. |
| $\mathcal{T}_L, \mathcal{T}_R$ | Left and right subtrees. |
| $n, n_{\min}$ | Node dataset ($\mathcal{D}$) size and minimum leaf size. |
| $\mathcal{L}_{(\cdot)}, K$ | An arbitrary leaf and the final number of leaves. |
| $s$ | Split anchor. |
| $N(s, \mathcal{T}_R, \mathcal{T}_L)$ | Split node with anchor $s$ and subtrees. |
| $\delta$ | Quantile bin. |
| $B$ | Number of bootstrap samples. |
| $\theta$ | Bootstrapping ratio. |
| **Theory** | |
| $i, j$ | Data indices within $\mathcal{D}$. |
| $\xi$ | Failure probability. |
| $\sigma_{\text{mix}}$ | $\beta$-mixing concentration bound constant. |
| $C$ | Patch to target smoothness ratio (constant). |
| $\mathcal{K}$ | Kernel. |

# B. Proof

## B.1. Theorem 5.4

We prove each claim of Theorem 5.4 based on the construction of Algorithm 1.

**Claim 1: Approximate Local Exchangeability**   Let $\mathcal{L}_{(\cdot)}$ be an arbitrary leaf terminated by

$$D_{\mathrm{KS}}(\tilde{\varepsilon}_i, s) \leq \gamma, \tag{24}$$

where $s$ denotes the split anchor selected at line 6 of Algorithm 1, for all indices $i \in \mathcal{I}(\mathcal{L}_{(\cdot)})$.

For any two samples $i, j \in \mathcal{I}(\mathcal{L}_{(\cdot)})$, the triangle inequality of the KS distance yields

$$D_{\mathrm{KS}}(\mathcal{P}_i, \mathcal{P}_j) = D_{\mathrm{KS}}(\tilde{\varepsilon}_i, \tilde{\varepsilon}_j) \leq D_{\mathrm{KS}}(\tilde{\varepsilon}_i, s) + D_{\mathrm{KS}}(s, \tilde{\varepsilon}_j). \tag{25}$$

Substituting (24) into (25), we obtain

$$D_{\mathrm{KS}}(\mathcal{P}_i, \mathcal{P}_j) \leq 2\gamma. \tag{26}$$

Therefore, the patch-level KS diameter within the leaf is bounded as

$$\max_{i,j\in\mathcal{I}(\mathcal{L}_{(\cdot)})} D_{\mathrm{KS}}(\mathcal{P}_i, \mathcal{P}_j) \leq 2\gamma. \tag{27}$$

**Claim 2: Minimum Leaf Size.**   We show that Algorithm 1 produces only leaves whose sizes are at least $n_{\min}$. The recursion terminates and declares a leaf $\mathcal{L}_{(\cdot)}$ in one of the following cases.

**Case 1.** $|\mathcal{D}| - |I_{i^\star}| < n_{\min}$. This corresponds to the termination condition, which prevents the creation of an undersized left child. In this case, the algorithm returns a leaf $\mathcal{L}_{(\cdot)} = \mathcal{D}$, and hence

$$|\mathcal{L}_{(\cdot)}| = |\mathcal{D}| \geq n_{\min}. \tag{28}$$

**Case 2.** $|I_{i^\star}| = |\mathcal{D}|$. In this case, all samples match the selected anchor. By Algorithm 1, the algorithm returns a leaf $\mathcal{L}_{(\cdot)} = \mathcal{D}$, and thus

$$|\mathcal{L}_{(\cdot)}| = |\mathcal{D}| \geq n_{\min}. \tag{29}$$

**Case 3.** *Split occurs.* If none of the above conditions hold, the algorithm proceeds to split. The negation of the termination conditions implies

$$|I_{i^\star}| > n_{\min}, \quad |I_{i^\star}| < |\mathcal{D}|, \quad |\mathcal{D}| - |I_{i^\star}| \geq n_{\min}. \tag{30}$$

Therefore, the resulting right and left subsets satisfy

$$|\mathcal{D}_R| = |I_{i^\star}| > n_{\min}, \qquad |\mathcal{D}_L| = |\mathcal{D}| - |I_{i^\star}| \geq n_{\min}. \tag{31}$$

Both children thus meet the minimum leaf size requirement.

**Inductive Argument.** We prove by strong induction on the tree depth that all terminal leaves satisfy $|\mathcal{L}| \geq n_{\min}$.

*Base case.* The root node $\mathcal{D}_0$ corresponds to the calibration set, which satisfies $|\mathcal{D}_0| \geq n_{\min}$ by assumption.

*Inductive step.* Assume that for all nodes up to depth $\ell$, any node that becomes a leaf satisfies $|\mathcal{L}| \geq n_{\min}$. Consider a node $\mathcal{D}$ at depth $\ell + 1$. By construction, $\mathcal{D}$ arises only from a valid split in Case 3, and therefore satisfies $|\mathcal{D}| \geq n_{\min}$. If $\mathcal{D}$ terminates (Cases 1–2), then $|\mathcal{L}_{(\cdot)}| = |\mathcal{D}| \geq n_{\min}$. If it splits (Case 3), both children satisfy the same condition, and the inductive hypothesis applies to all their descendants.

Thus, by induction, all leaves produced by Algorithm 1 satisfy $|\mathcal{L}| \geq n_{\min}$.   □

## B.2. Theorem 5.6

Let $\tilde{\varepsilon}_T$ denote the unseen patch, and let

$$\mathcal{L}_{(k^\star)} := \mathcal{L}(\tilde{\varepsilon}_T) \tag{32}$$

be the leaf selected by routing $\tilde{\varepsilon}_T$ through the learned matching tree. We aim to bound the coverage error

$$\left| \mathbb{P}\left( \varepsilon_{T+1} \leq \hat{q}_{1-\alpha+\delta^\star}^{\mathcal{L}_{(k^\star)}} \right) - (1-\alpha) \right|, \tag{33}$$

where $\hat{q}_{1-\alpha+\delta^\star}^{\mathcal{L}_{(k^\star)}}$ denotes the empirical $(1-\alpha+\delta^\star)$-quantile computed using calibration targets within $\mathcal{L}_{(k^\star)}$, and $\delta^\star \in [0, \alpha]$ is the adaptive refinement parameter computed via Eq. (10).

**Step 1: Patch-Level Proximity of the Unseen Sample**    At each internal node with anchor $s_{(\cdot)}$, the routing rule is

$$\tilde{\varepsilon}_T \mapsto \begin{cases} \text{right child}, & D_{\mathrm{KS}}(\tilde{\varepsilon}_T, s_{(\cdot)}) \leq \gamma, \\ \text{left child}, & \text{otherwise.} \end{cases} \tag{34}$$

If $\tilde{\varepsilon}_T$ is routed to the matched (right-child) leaf $\mathcal{L}_{(k^\star)}$, then by construction,

$$D_{\mathrm{KS}}(\tilde{\varepsilon}_T, s_{(k^\star)}) \leq \gamma, \tag{35}$$

where $s_{(k^\star)}$ denotes the anchor that formed this leaf. Moreover, for all calibration patches $\tilde{\varepsilon}_t$ with $t \in \mathcal{I}(\mathcal{L}_{(k^\star)})$,

$$D_{\mathrm{KS}}(\tilde{\varepsilon}_t, s_{(k^\star)}) \leq \gamma. \tag{36}$$

Combining (35) and (36) via the triangle inequality yields

$$D_{\mathrm{KS}}(\tilde{\varepsilon}_T, \tilde{\varepsilon}_t) \leq 2\gamma, \qquad \forall t \in \mathcal{I}(\mathcal{L}_{(k^\star)}). \tag{37}$$

**Step 2: Propagation from Patch-Level to Target-Level Distributions**    Let $P_{t+1}$ and $P_{T+1}$ denote the distributions of the target residuals $\varepsilon_{t+1}$ and $\varepsilon_{T+1}$, respectively.

Under Assumption 5.2, the $\beta$-mixing condition implies that distributional proximity between residual patches propagates to their associated targets with a controlled error:

$$D_{\mathrm{KS}}(P_{t+1}, P_{T+1}) \leq C\, D_{\mathrm{KS}}(\tilde{\varepsilon}_t, \tilde{\varepsilon}_T) + O(\sigma_{\mathrm{mix}}), \tag{38}$$

where

$$\sigma_{\mathrm{mix}} = O\left( \sum_{m=1}^{\infty} \sqrt{\beta_m} \right), \tag{39}$$

and $C$ is a constant depending only on the regularity of the conditional distribution of $\varepsilon_{t+1}$ given $\tilde{\varepsilon}_t$. The constant $C$ depends on the local regularity of the conditional target CDF (e.g., bounded density around the target quantile), and is O(1) for common ARMA, state-space, and Lipschitz-driven nonlinear time series. Substituting (37) into (38) yields

$$D_{\mathrm{KS}}(P_{t+1}, P_{T+1}) \leq 2C\gamma + O(\sigma_{\mathrm{mix}}), \qquad \forall t \in \mathcal{I}(\mathcal{L}_{(k^\star)}). \tag{40}$$

Defining

$$\gamma^\star := 2C\gamma + O(\sigma_{\mathrm{mix}}), \tag{41}$$

we obtain

$$\max_{t \in \mathcal{I}(\mathcal{L}_{(k^\star)})} D_{\mathrm{KS}}(P_{t+1}, P_{T+1}) \leq \gamma^\star, \tag{42}$$

which establishes $\gamma^\star$-approximate local exchangeability at the target level in the sense of Definition 5.1.

**Step 3: Decomposition of the Coverage Error**   Define the empirical CDF within the selected leaf

$$\hat{F}_{\mathcal{L}_{(k^\star)}}(x) := \frac{1}{|\mathcal{L}_{(k^\star)}|} \sum_{t \in \mathcal{I}(\mathcal{L}_{(k^\star)})} \mathbb{1}\{\varepsilon_{t+1} \leq x\}, \tag{43}$$

its expectation

$$\bar{F}_{\mathcal{L}_{(k^\star)}}(x) := \frac{1}{|\mathcal{L}_{(k^\star)}|} \sum_{t \in \mathcal{I}(\mathcal{L}_{(k^\star)})} F_{t+1}(x), \tag{44}$$

and the test CDF

$$F_{T+1}(x) := \mathbb{P}(\varepsilon_{T+1} \leq x). \tag{45}$$

By the triangle inequality,

$$\sup_x \left| \hat{F}_{\mathcal{L}_{(k^\star)}}(x) - F_{T+1}(x) \right| \leq \sup_x \left| \hat{F}_{\mathcal{L}_{(k^\star)}}(x) - \bar{F}_{\mathcal{L}_{(k^\star)}}(x) \right| + \sup_x \left| \bar{F}_{\mathcal{L}_{(k^\star)}}(x) - F_{T+1}(x) \right|. \tag{46}$$

**Step 4: Bias and Variance Bounds**   From $\gamma^\star$-approximate local exchangeability (41), we have

$$\sup_x \left| \bar{F}_{\mathcal{L}_{(k^\star)}}(x) - F_{T+1}(x) \right| \leq \gamma^\star =: m_{\mathcal{L}_{(k^\star)}}. \tag{47}$$

Moreover, by concentration inequalities for $\beta$-mixing sequences (Rio, 1993), for any $\xi \in (0,1)$, with probability at least $1 - \xi$,

$$\sup_x \left| \hat{F}_{\mathcal{L}_{(k^\star)}}(x) - \bar{F}_{\mathcal{L}_{(k^\star)}}(x) \right| \leq C' \sigma_{\mathrm{mix}} \sqrt{\frac{\log(1/\xi)}{|\mathcal{L}_{(k^\star)}|}} =: \varrho_{\mathcal{L}_{(k^\star)}}(\xi), \tag{48}$$

where $C' > 0$ is a universal constant.

**Step 5: Quantile Mapping and Coverage Error**   By definition of the empirical quantile,

$$\hat{q}_{1-\alpha+\delta^\star}^{\mathcal{L}_{(k^\star)}} := \inf \left\{ y \in \mathbb{R} : \hat{F}_{\mathcal{L}_{(k^\star)}}(y) \geq 1 - \alpha + \delta^\star \right\}. \tag{49}$$

Combining the above bounds, on the event of probability at least $1 - \xi$, we obtain

$$\left| \mathbb{P}\left( \varepsilon_{T+1} \leq \hat{q}_{1-\alpha+\delta^\star}^{\mathcal{L}_{(k^\star)}} \right) - (1-\alpha) \right| \leq m_{\mathcal{L}_{(k^\star)}} + \varrho_{\mathcal{L}_{(k^\star)}}(\xi) + \delta^\star + \frac{1}{|\mathcal{L}_{(k^\star)}|}. \tag{50}$$

Substituting the explicit forms of $m_{\mathcal{L}_{(k^\star)}}$ and $\varrho_{\mathcal{L}_{(k^\star)}}(\xi)$ completes the proof.   $\square$

## B.3. Proof of Proposition 5.12

**Recap of notation.** For each bootstrap tree $b \in \{1, \dots, B\}$, let $\mathcal{L}_t^{(b)}$ denote the leaf to which $\tilde{\varepsilon}_t$ is routed in tree $b$, and let $\mathcal{L}_{\mathrm{OOD}}^{(b)}$ denote its leftmost (unmatched) leaf. Define the per-step OOD-routing event

$$E_{\mathrm{OOD},t}^{(b)} := \left\{ \tilde{\varepsilon}_t \text{ is routed to } \mathcal{L}_{\mathrm{OOD}}^{(b)} \text{ in tree } b \right\}. \tag{51}$$

For a target level $\tau$ we write the per-tree coverage functional

$$c_t^{(b)}(\tau) := \mathbb{P}\left( \varepsilon_{t+1} \leq \hat{q}_\tau^{(b)}(\tilde{\varepsilon}_t) \right), \tag{52}$$

and the ensemble quantile $\hat{q}_\tau^{\mathrm{ens}}(\tilde{\varepsilon}_t) := \frac{1}{B} \sum_{b=1}^{B} \hat{q}_\tau^{(b)}(\tilde{\varepsilon}_t)$. Finally, let $M_t := \mathbb{P}(y_{t+1} \in \hat{\mathcal{C}}_t^\alpha)$ denote the per-step marginal coverage and $Z_t := \mathbb{1}\{y_{t+1} \in \hat{\mathcal{C}}_t^\alpha\}$. Throughout, $|\mathcal{L}|_{\min} := \min_{b,t} |\mathcal{L}_t^{(b)}|$ is the smallest leaf size encountered over the online run, and $m_{\max} := \sup_{b,t} m_{\mathcal{L}_t^{(b)}}$. The proof proceeds by (i) controlling the per-step marginal coverage $M_t$, then (ii) aggregating $\{Z_t\}$ to $\{M_t\}$ over the online sequence.

**Step 1: Per-step OOD decomposition for a fixed tree.** Fix $t \in \{T, \dots, T + n_{\mathrm{test}} - 1\}$, a tree $b$, and a level $\tau$. By the law of total probability with respect to $E_{\mathrm{OOD},t}^{(b)}$,

$$c_t^{(b)}(\tau) = \mathbb{P}\left( \varepsilon_{t+1} \leq \hat{q}_\tau^{(b)} \mid \neg E_{\mathrm{OOD},t}^{(b)} \right) \mathbb{P}\left( \neg E_{\mathrm{OOD},t}^{(b)} \right) + \mathbb{P}\left( \varepsilon_{t+1} \leq \hat{q}_\tau^{(b)} \mid E_{\mathrm{OOD},t}^{(b)} \right) \mathbb{P}\left( E_{\mathrm{OOD},t}^{(b)} \right). \tag{53}$$

**Step 2: Matched-leaf bound via Theorem 5.6.** On $\neg E_{\mathrm{OOD},t}^{(b)}$, the patch $\tilde{\varepsilon}_t$ is routed to a matched leaf, so Theorem 5.6 applies to that leaf. Since the tree structure is fixed and the online QRF update only appends the observed residual to its assigned leaf, the leaf-wise estimator at step $t$ is a consistent quantile estimator built from $|\mathcal{L}_t^{(b)}| \geq n_{\min}$ samples. Hence, for any $\xi' \in (0,1)$, with probability at least $1 - \xi'$,

$$\left| \mathbb{P}\left( \varepsilon_{t+1} \leq \hat{q}_\tau^{(b)} \mid \neg E_{\mathrm{OOD},t}^{(b)} \right) - \tau \right| \leq 2C\gamma + O(\sigma_{\mathrm{mix}}) + O\left( \sigma_{\mathrm{mix}} \sqrt{\tfrac{\log(1/\xi')}{|\mathcal{L}|_{\min}}} \right), \tag{54}$$

where the bias term $2C\gamma + O(\sigma_{\mathrm{mix}})$ is the target-level divergence $m_{\mathcal{L}}$ of Eq. (15) and the last term is the finite-sample term $\varrho_{\mathcal{L}}$ of Eq. (16).

**Step 3: Worst-case bound under OOD routing.** On $E_{\mathrm{OOD},t}^{(b)}$, approximate local exchangeability may fail, so we apply the conservative bound

$$\left| \mathbb{P}\left( \varepsilon_{t+1} \leq \hat{q}_\tau^{(b)} \mid E_{\mathrm{OOD},t}^{(b)} \right) - \tau \right| \leq m_{\max}. \tag{55}$$

**Step 4: Per-tree marginal bound via bootstrap diversity.** The $B$ bootstrap trees are constructed by i.i.d. subsampling with a common ratio $\theta$, hence are identically distributed; consequently $\mathbb{P}(\neg E_{\mathrm{OOD},t}^{(b)})$ does not depend on $b$. Assumption 5.10 therefore yields, for every $b$,

$$\mathbb{P}\left( \neg E_{\mathrm{OOD},t}^{(b)} \right) \geq p_{\min}, \qquad \mathbb{P}\left( E_{\mathrm{OOD},t}^{(b)} \right) \leq 1 - p_{\min}. \tag{56}$$

Combining Steps 1–3 with $\mathbb{P}(\neg E_{\mathrm{OOD},t}^{(b)}) \leq 1$, we obtain that, with probability at least $1 - \xi'$,

$$\left| c_t^{(b)}(\tau) - \tau \right| \leq (1 - p_{\min}) m_{\max} + 2C\gamma + O(\sigma_{\mathrm{mix}}) + O\left( \sigma_{\mathrm{mix}} \sqrt{\tfrac{\log(1/\xi')}{|\mathcal{L}|_{\min}}} \right) =: \Delta(\xi'), \tag{57}$$

uniformly over $b$.

**Step 5: From per-tree quantiles to the ensemble quantile.** By Assumption 5.11 (quantile averaging stability) applied to the level-$\tau$ estimators $\{\hat{q}_\tau^{(b)}\}_{b=1}^{B}$,

$$\left| \mathbb{P}\left( \varepsilon_{t+1} \leq \hat{q}_\tau^{\mathrm{ens}}(\tilde{\varepsilon}_t) \right) - \tau \right| \leq \frac{1}{B} \sum_{b=1}^{B} \left| c_t^{(b)}(\tau) - \tau \right| + \kappa_{\mathrm{avg}}, \qquad \kappa_{\mathrm{avg}} = O(B^{-1}). \tag{58}$$

Substituting the uniform per-tree bound (57), the average is itself bounded by $\Delta(\xi')$, hence

$$\left|\mathbb{P}\big(\varepsilon_{t+1} \leq \hat{q}_\tau^{\mathrm{ens}}(\tilde{\varepsilon}_t)\big) - \tau\right| \leq \Delta(\xi') + O(B^{-1}). \tag{59}$$

This is the step that converts the (otherwise only randomized-index) guarantee into one for the averaged ensemble quantile.

**Step 6: Adaptive refinement.** The interval $\hat{\mathcal{C}}_t^\alpha$ in Eq. (21) uses the refined level $\tau = 1 - \alpha + \delta^\star$ with $\delta^\star \in [0, \alpha]$ from Eq. (10). Applying (59) at this level and using $|(1 - \alpha + \delta^\star) - (1 - \alpha)| = \delta^\star$,

$$\left|\mathbb{P}\big(\varepsilon_{t+1} \leq \hat{q}_{1-\alpha+\delta^\star}^{\mathrm{ens}}(\tilde{\varepsilon}_t)\big) - (1 - \alpha)\right| \leq \Delta(\xi') + O(B^{-1}) + \delta^\star. \tag{60}$$

Consequently the per-step marginal coverage satisfies

$$M_t \geq 1 - \alpha - \epsilon_{\mathrm{step}}(\xi'), \qquad \epsilon_{\mathrm{step}}(\xi') := \Delta(\xi') + O(B^{-1}) + \delta^\star. \tag{61}$$

**Step 7: Online aggregation via $\beta$-mixing concentration.** It remains to relate the empirical coverage $\frac{1}{n_{\mathrm{test}}} \sum_t Z_t$ to the per-step probabilities $\frac{1}{n_{\mathrm{test}}} \sum_t M_t$. The indicators $Z_t$ are determined by the residual patches $\tilde{\varepsilon}_t$, which overlap—and are thus mechanically dependent—only for lags $|t - t'| \leq w$; for lags exceeding $w$ their dependence is governed by the $\beta$-mixing coefficients of the patch process (Assumption 5.9). Partitioning the test horizon into blocks of length $w$ and coupling each block to an independent copy incurs a total variation cost of order $\beta_w$ per block (Rio, 1993; Yu, 1994). A Bernstein-type inequality for $\beta$-mixing sequences then yields, with probability at least $1 - \xi'$,

$$\left|\frac{1}{n_{\mathrm{test}}} \sum_{t=T}^{T+n_{\mathrm{test}}-1} Z_t - \frac{1}{n_{\mathrm{test}}} \sum_{t=T}^{T+n_{\mathrm{test}}-1} M_t\right| \leq O\left(\sqrt{\frac{\log(1/\xi')}{n_{\mathrm{test}}}}\right) + O\left(\sqrt{\beta_w}\right), \tag{62}$$

where the first term is the statistical fluctuation and $O(\sqrt{\beta_w})$ is the coupling residue from window-induced overlap. Since each leaf is a subset of the calibration set, $|\mathcal{L}|_{\min} \leq n_{\mathrm{test}}$ in the regimes of interest, so the statistical term in (62) is dominated by the finite-sample term of (54) and need not be tracked separately.

**Step 8: Union bound and conclusion.** We invoke (54) at every step $t$ and the concentration bound (62) once. Taking $\xi' = \xi/(n_{\mathrm{test}} + 1)$ and a union bound over the $n_{\mathrm{test}} + 1$ events replaces every $\log(1/\xi')$ by $\log(n_{\mathrm{test}}/\xi)$. Combining (61) and (62), with probability at least $1 - \xi$ over the calibration sampling and the online process,

$$\frac{1}{n_{\mathrm{test}}} \sum_{t=T}^{T+n_{\mathrm{test}}-1} \mathbb{1}\{y_{t+1} \in \hat{\mathcal{C}}_t^\alpha\} \geq 1 - \alpha - \epsilon_{\mathrm{online}}, \tag{63}$$

with

$$\epsilon_{\mathrm{online}} \leq (1 - p_{\min}) m_{\max} + 2C\gamma + O(\sigma_{\mathrm{mix}}) + O\left(\sqrt{\frac{\log(n_{\mathrm{test}}/\xi)}{|\mathcal{L}|_{\min}}}\right) + O\left(\sqrt{\beta_w}\right) + O(B^{-1}) + \delta^\star. \tag{64}$$

This is exactly the bound of Eq. (23), where $(1 - p_{\min})m_{\max}$ is the OOD fallback error, $2C\gamma + O(\sigma_{\mathrm{mix}})$ the within-leaf bias and temporal-propagation error, the $|\mathcal{L}|_{\min}$ term the leaf-wise finite-sample error, $O(\sqrt{\beta_w})$ the online-sequence coupling error, $O(B^{-1})$ the ensemble-averaging error, and $\delta^\star$ the adaptive-refinement offset. $\qquad \square$

### B.4. Discussion on Definition 5.1

B.4.1. RELATIONSHIP BETWEEN KS AND TV DISTANCES

Several recent sequential CP methods employ different notions of approximate exchangeability to handle non-stationary residuals. Most notably, NexCP with the temporal reweighting method (Barber et al., 2023) relies on Total Variation (TV) distance to quantify distributional discrepancies.

For two probability measures $P$ and $Q$ on $\mathbb{R}$ with densities $p$ and $q$, the TV distance is defined as:

$$D_{\mathrm{TV}}(P,Q) = \frac{1}{2} \int_{-\infty}^{\infty} |p(x) - q(x)|\, dx = \sup_{A \in \mathcal{B}(\mathbb{R})} |P(A) - Q(A)|, \tag{65}$$

where $\mathcal{B}(\mathbb{R})$ is the Borel $\sigma$-algebra on $\mathbb{R}$.

In contrast, Definition 5.1 employs the Kolmogorov–Smirnov (KS) distance:

$$D_{\mathrm{KS}}(P,Q) = \sup_{x \in \mathbb{R}} |F_P(x) - F_Q(x)|, \tag{66}$$

where $F_P$ and $F_Q$ are the cumulative distribution functions.

These two metrics are related by the fundamental inequality:

$$D_{\mathrm{KS}}(P,Q) \leq D_{\mathrm{TV}}(P,Q). \tag{67}$$

*Proof of* (67). For any $x_0 \in \mathbb{R}$, consider the event $A = (-\infty, x_0]$:

$$|F_P(x_0) - F_Q(x_0)| = |P((-\infty, x_0]) - Q((-\infty, x_0])| \tag{68}$$
$$\leq \sup_{A \in \mathcal{B}(\mathbb{R})} |P(A) - Q(A)| = D_{\mathrm{TV}}(P,Q). \tag{69}$$

Taking the supremum over all $x_0 \in \mathbb{R}$ yields $D_{\mathrm{KS}}(P,Q) \leq D_{\mathrm{TV}}(P,Q)$. $\qquad\square$

We emphasize that the reverse direction does not hold in general: two distributions can have nearly identical CDFs (small $D_{\mathrm{KS}}$) yet large $D_{\mathrm{TV}}$ when their densities differ on a fine scale. Thus $D_{\mathrm{KS}}$ and $D_{\mathrm{TV}}$ are not equivalent, and the choice between them is consequential for sequential CP, as we discuss next.

B.4.2. WHY KS DISTANCE IS MORE SUITABLE FOR QUANTILE-BASED SEQUENTIAL CP

**1. Direct Alignment with Quantile Estimation.** Conformal prediction fundamentally relies on quantiles of the residual distribution. The $(1-\alpha)$-quantile $q_{1-\alpha}$ satisfies:

$$F_P(q_{1-\alpha}) = 1 - \alpha, \tag{70}$$

where $F_P$ is the CDF of the residual distribution $P$.

**KS control directly bounds quantile error.** If $D_{\mathrm{KS}}(P,Q) \leq \gamma$, then for any quantile $q_\tau^P$ of distribution $P$,

$$|F_Q(q_\tau^P) - F_P(q_\tau^P)| = |F_Q(q_\tau^P) - \tau| \leq D_{\mathrm{KS}}(P,Q) \leq \gamma. \tag{71}$$

This immediately implies that the quantile $q_\tau^P$ computed from distribution $P$ provides an approximately valid $(1-\alpha)$-quantile for distribution $Q$, with controlled error $\gamma$.

**TV control provides only an indirect bound.** TV distance measures the total probability-mass difference but does not directly control pointwise CDF discrepancies. Even if $D_{\mathrm{TV}}(P,Q)$ is small, the CDFs could differ significantly at specific points relevant for quantile estimation.

**2. A Weaker Sufficient Condition for Sequential CP.** By (67), the only universally valid relation between the two metrics is $D_{\mathrm{KS}} \leq D_{\mathrm{TV}}$. Consequently, for a common tolerance $\gamma$, requiring $D_{\mathrm{TV}} \leq \gamma$ is a *strictly stronger* condition than requiring $D_{\mathrm{KS}} \leq \gamma$:

- **NexCP approach:** requires $D_{\mathrm{TV}}(P_t, P_{T+1}) \leq \gamma$ for approximate exchangeability.

- **DistMatch approach:** requires only $D_{\mathrm{KS}}(P_t, P_{T+1}) \leq \gamma$.

Crucially, valid conformal coverage depends solely on quantile (equivalently, CDF) discrepancies, which $D_{\mathrm{KS}}$ controls directly (Eq. (71); Lemma B.1). A $D_{\mathrm{TV}}$ bound therefore over-constrains the problem: it additionally suppresses density-level discrepancies that are irrelevant to quantile estimation, and—unlike $D_{\mathrm{KS}}$—cannot be evaluated without density estimation. DistMatch thus attains the same coverage guarantee under a *weaker, density-free* matching condition, which is easier to satisfy in finite samples and yields larger, more stable leaves for the same $\gamma$.

**3. Robustness under Skewed Residual Distributions.** Real-world residuals often exhibit heavy tails and skewness (Figure 2). In such settings:

**KS distance is robust to tail behavior.** The KS statistic measures the maximum vertical distance between ECDFs, which is less sensitive to extreme values than density-based metrics. Even if residuals have heavy tails, the ECDF remains well-behaved, and the KS distance can be reliably estimated from finite samples.

**TV distance requires density estimation.** Computing $D_{\mathrm{TV}} = \frac{1}{2} \int |p(x) - q(x)| \, dx$ necessitates estimating the densities $p$ and $q$, which is notoriously difficult for heavy-tailed or skewed distributions. Kernel density estimation and histogram-based methods introduce additional hyperparameters (bandwidth, bin size) and can be highly biased under skewness.

**Empirical comparison.** Figure 7 demonstrates that on skewed residual distributions (e.g., the Solar and Wind datasets), KS-based matching achieves comparable or better Winkler scores than KL divergence, Wasserstein distance, and Mutual Information, while maintaining lower computational cost (Appendix C).

**4. Non-Parametric and Assumption-Free.** **The KS statistic is distribution-free.** The two-sample KS statistic assumes no parametric form for the underlying distributions. This is crucial for sequential CP, where residual distributions may change arbitrarily over time due to distributional shifts.

**TV distance often assumes density existence.** While TV can be defined measure-theoretically, its practical estimation typically assumes absolute continuity (existence of densities), which may not hold for discrete or mixed-type residuals.

**No temporal stationarity assumption.** Unlike TV-based methods that often rely on slowly-varying densities or smooth transitions (Barber et al., 2023), the KS statistic makes no assumptions about how distributions evolve over time. This is particularly advantageous under abrupt shifts (e.g., Figure 1, Solar dataset).

B.4.3. COMPARISON WITH NEXCP'S TV-BASED APPROACH

NexCP (Barber et al., 2023) defines approximate exchangeability via temporal reweighting using TV distance. Specifically, NexCP assigns weights:

$$w_t \propto \exp(-\rho \cdot t), \tag{72}$$

where $\rho$ is a decay rate, and shows that if $D_{\mathrm{TV}}(P_t, P_{T+1}) \leq O(\rho)$, then the weighted empirical distribution approximates $P_{T+1}$.

**Key differences from DistMatch.**

1. **Global vs. local.** NexCP enforces approximate exchangeability globally across all calibration samples via continuous weights, whereas DistMatch enforces it locally within homogeneous leaves via discrete binning.

2. **Weight estimation.** NexCP requires tuning the decay rate $\rho$, which is sensitive to the rate of distribution drift. Misspecifying $\rho$ leads to either overfitting recent data (too large $\rho$) or ignoring shifts (too small $\rho$). DistMatch avoids weight estimation by using the data-driven threshold $\gamma$.

3. **Computational efficiency.** NexCP recomputes weighted quantiles at every time step over the entire calibration history, incurring $O(T^2)$ cumulative cost over $T$ steps. DistMatch builds the tree once ($O(n^2 w \log n)$) and performs online updates only within the selected leaves, achieving $O(Tw \log n)$ amortized cost.

4. **Heterogeneity handling.** NexCP's exponential weighting assumes monotonic drift, struggling with non-monotonic shifts or multi-modal distributions. DistMatch naturally handles heterogeneous shifts by partitioning into multiple leaves, each capturing a distinct distribution mode (Figure 6b).

**Empirical validation.** Table 1 shows that DistMatch consistently outperforms NexCP on datasets with complex distribution shifts (Solar, META, NVDA), achieving 60–70% narrower intervals while maintaining target coverage. This validates the advantage of KS-based local binning over TV-based global reweighting.

### B.4.4. WHY DEFINITION 5.1 IS APPROPRIATE FOR SEQUENTIAL CP

We formalize why Definition 5.1 (KS-based approximate local exchangeability) provides the appropriate notion of "approximate exchangeability" for sequential CP.

**Lemma B.1** (Quantile Validity under KS-Bounded Distributions). *Let $\{P_t\}_{t \in \mathcal{I}(\mathcal{L})}$ be a collection of distributions satisfying*

$$\max_{t \in \mathcal{I}(\mathcal{L})} D_{\mathrm{KS}}(P_t, P_{T+1}) \leq \gamma^\star, \tag{73}$$

*and let $\hat{q}_{1-\alpha}$ be the empirical $(1 - \alpha)$-quantile computed from the samples $\{\varepsilon_t\}_{t \in \mathcal{I}(\mathcal{L})}$. Then, for any $\eta \in (0, 1)$, with probability at least $1 - \eta$,*

$$\left| \mathbb{P}_{P_{T+1}}(\varepsilon \leq \hat{q}_{1-\alpha}) - (1 - \alpha) \right| \leq \gamma^\star + O\left( \sqrt{\frac{\log(1/\eta)}{|\mathcal{L}|}} \right). \tag{74}$$

*Proof.* By the definition of the empirical quantile,

$$\hat{F}_{\mathcal{L}}(\hat{q}_{1-\alpha}) = \frac{1}{|\mathcal{L}|} \sum_{t \in \mathcal{I}(\mathcal{L})} \mathbb{I}\{\varepsilon_t \leq \hat{q}_{1-\alpha}\} \approx 1 - \alpha. \tag{75}$$

The expected CDF averaged over the leaf is

$$\bar{F}_{\mathcal{L}}(x) = \frac{1}{|\mathcal{L}|} \sum_{t \in \mathcal{I}(\mathcal{L})} F_{P_t}(x). \tag{76}$$

By the KS bound,

$$|F_{P_t}(x) - F_{P_{T+1}}(x)| \leq D_{\mathrm{KS}}(P_t, P_{T+1}) \leq \gamma^\star, \quad \forall t \in \mathcal{I}(\mathcal{L}), \tag{77}$$

and averaging over the leaf yields

$$|\bar{F}_{\mathcal{L}}(x) - F_{P_{T+1}}(x)| \leq \gamma^\star. \tag{78}$$

By a bounded-difference concentration inequality for $\beta$-mixing sequences (Rio, 1993), for any $\eta \in (0, 1)$, with probability at least $1 - \eta$,

$$\sup_x |\hat{F}_{\mathcal{L}}(x) - \bar{F}_{\mathcal{L}}(x)| \leq O\left( \sqrt{\frac{\log(1/\eta)}{|\mathcal{L}|}} \right). \tag{79}$$

Combining these bounds at $x = \hat{q}_{1-\alpha}$, with probability at least $1 - \eta$,

$$\left| F_{P_{T+1}}(\hat{q}_{1-\alpha}) - (1 - \alpha) \right| \leq |F_{P_{T+1}}(\hat{q}_{1-\alpha}) - \bar{F}_{\mathcal{L}}(\hat{q}_{1-\alpha})| \tag{80}$$

$$+ |\bar{F}_{\mathcal{L}}(\hat{q}_{1-\alpha}) - \hat{F}_{\mathcal{L}}(\hat{q}_{1-\alpha})| \tag{81}$$

$$+ |\hat{F}_{\mathcal{L}}(\hat{q}_{1-\alpha}) - (1 - \alpha)| \tag{82}$$

$$\leq \gamma^\star + O\left( \sqrt{\frac{\log(1/\eta)}{|\mathcal{L}|}} \right) + O\left( \frac{1}{|\mathcal{L}|} \right) \tag{83}$$

$$= \gamma^\star + O\left( \sqrt{\frac{\log(1/\eta)}{|\mathcal{L}|}} \right), \tag{84}$$

where the last step absorbs the $O(1/|\mathcal{L}|)$ quantization term into the dominant $O(\sqrt{\log(1/\eta)/|\mathcal{L}|})$ term.  □

**Interpretation.** Lemma B.1 shows that Definition 5.1 directly ensures valid conformal coverage, with the KS bound $\gamma^\star$ translating linearly into the coverage error. This justifies using the KS distance as the splitting criterion in Algorithm 1.

### B.4.5. PRACTICAL IMPLICATIONS

The choice of KS distance in Definition 5.1 has several practical advantages for implementing sequential CP.

1. **Efficient computation.** The two-sample KS statistic can be computed in $O(n \log n)$ time by sorting, making it feasible to evaluate all pairwise comparisons during tree construction.

2. **Interpretable threshold.** The threshold $\gamma$ has a clear interpretation: it bounds the maximum difference between CDFs at any quantile level. Practitioners can set $\gamma \in [0.005, 0.1]$ to ensure visibly similar distributions (Figure 2).

3. **Stable under small samples.** The KS statistic converges at rate $O(1/\sqrt{n})$, faster than density-based estimators. This ensures reliable matching even when leaf sizes are moderate ($|\mathcal{L}| = 10$–$50$).

4. **No hyperparameter sensitivity.** Unlike kernel-based methods (e.g., MMD requires bandwidth selection) or optimal transport methods (e.g., Wasserstein distance requires metric/cost specification), the KS statistic has no tunable hyperparameters beyond the data itself.

### B.4.6. CONCLUSION

Definition 5.1 provides a theoretically grounded and practically effective notion of approximate exchangeability for sequential CP:

- **Theoretical soundness:** the KS distance directly controls quantile error (Lemma B.1), ensuring valid coverage (Theorem 5.6).

- **Weaker sufficient condition:** KS-based matching achieves valid coverage under a weaker, density-free condition than TV-based reweighting, and is easier to satisfy in finite samples.

- **Robustness:** the KS distance is distribution-free and stable under the skewed, heavy-tailed residuals common in real-world time series.

- **Efficiency:** KS-based matching enables efficient tree construction and online inference.

These advantages explain why DistMatch consistently outperforms TV-based methods (NexCP) and other reweighting approaches across diverse datasets, validating the design choice of KS-based approximate local exchangeability.

# C. Motivation of KS Statistic

There are several possible candidates for distribution-matching criteria in binning-based sequential CP. However, we justify that the KS statistic is the most suitable choice.

## C.1. Divergence-Based Methods

Kullback–Leibler (KL) divergence (Kullback & Leibler, 1951) can be considered as a distribution-matching criterion, as it measures the discrepancy between two probability distributions. However, it requires additional assumptions on the underlying distributions, such as absolute continuity and shared support, and it diverges to infinity when the distributions do not have overlapping support (Akbari et al., 2021). In addition, estimating KL divergence from finite samples often relies on density estimation techniques (e.g., histograms or kernel density estimation), which introduce sensitive hyperparameters and may bias the results (Wang et al., 2009). Furthermore, KL divergence is asymmetric with respect to the order of the distributions (Riaz et al., 2023), which can hinder exchangeability.

## C.2. Optimal Transport and Kernel-Based Methods

Optimal transport and kernel-based methods can also serve as distribution-matching criteria. For example, the Wasserstein distance (Villani, 2009), an optimal transport metric, computes the minimum cost required to transform one distribution into another, thereby distinguishing between different distributions. Kernel-based methods, such as Maximum Mean Discrepancy (MMD) (Gretton et al., 2006), instead compare distributions through their embeddings in a Reproducing Kernel Hilbert Space (RKHS). While both metrics are effective at capturing global distributional shifts, they may overlook subtle local mismatches (Zhu et al., 2025). In addition, MMD scales quadratically with the sample size (Gretton et al., 2012), and can suffer from bias and instability when distributions differ substantially (Dayal et al., 2025).

## C.3. Moment-Based Methods

Moment-based methods provide another class of distribution-matching criteria. For example, Correlation Alignment (CORAL) (Sun et al., 2017) quantifies distributional differences by aligning second-order statistics, i.e., covariance matrices, between distributions. However, it struggles to capture local distributional characteristics, as it relies solely on global covariance structure and ignores higher-order or localized variations. In contrast, Central Moment Discrepancy (CMD) (Zellinger et al., 2017) matches higher-order central moments to better capture local discrepancies. Nevertheless, CMD can become unstable when the distribution centers differ

substantially, leading to distortions in the comparison (Zhu et al., 2025). As a result, its reliability degrades under skewed or heavy-tailed distributions, which are commonly encountered in sequential CP settings.

## C.4. Comparison with KS Statistic

The aforementioned methods are unsuitable as distribution-matching criteria for three main reasons: (1) some rely on parametric or distributional assumptions, (2) some overlook local discrepancies, and (3) some incur high computational costs. In contrast, the KS statistic is a nonparametric measure that directly quantifies distributional similarity without requiring additional temporal assumptions or density estimation. Furthermore, by leveraging the maximum deviation between ECDFs, it captures subtle local differences with $O(n \log n)$ computational complexity and exhibits stable behavior under distribution shifts and outliers.

## C.5. KS Statistic and $p$-value

The Kolmogorov–Smirnov (KS) test is a $p$-value–based hypothesis test that explicitly requires strong temporal assumptions, such as independence or global stationarity (Massey, 1951; Dos Reis et al., 2016), which are often violated in real-world time series. In contrast, the KS statistic itself depends only on ECDFs and can therefore be used as a direct measure of distributional similarity without invoking formal hypothesis testing. Motivated by this property, several recent works leverage the KS statistic rather than the KS test, thereby relaxing global temporal assumptions. For example, Gong & Huang (2012) proposes a KS-statistic–based segmentation method in which segmentation and feature selection become insensitive to class imbalance and distribution skew, as the KS statistic is robust to differences in class proportions and distribution shapes. This robustness allows the method to partition complex imbalanced datasets into simpler subproblems and to select relevant features without bias toward the majority class, improving performance in settings where class imbalance would otherwise degrade standard algorithms. Another line of work explores KS-based online compression (Gibson et al., 2018), which enables direct comparison of empirical distributions without relying on $p$-values. This property yields stable similarity decisions under non-stationary streaming data and varying block sizes, where $p$-value–based tests become unreliable. DistMatch similarly employs the KS statistic as a pure measure of distributional similarity between residual sets, with the additional advantage of effectively distinguishing skewed distributions. Consequently, the KS statistic is particularly well-suited for residual-based binning in sequential conformal prediction.

### C.5.1. COMPARISON WITH OTHER TREE-BASED METHODS

Standard tree-based methods (Breiman, 2001; Chen & Guestrin, 2016) operate in a supervised manner, maximizing information gain (IG) by splitting nodes to reduce label impurity (e.g., entropy or the Gini index) or prediction error. In contrast, DistMatch operates without labels and therefore redefines IG as a matching-count–based gain (MG) that measures distributional similarity, with the tree optimized to maximize MG by selecting an optimal input as an anchor. This formulation preserves the core principle of tree-based learning—maximizing IG—while enabling unsupervised, distribution-driven splitting.

## D. Extended Related Work

Online conformal inference provides an alternative approach to conformal prediction (CP) by relying on a backbone model that simultaneously produces point predictions and corresponding prediction intervals. This framework accounts for potential distribution shifts by dynamically adjusting quantile levels through online updates of the significance parameter. For example, Adaptive Conformal Inference (ACI) (Gibbs & Candes, 2021) calibrates quantile predictions by incorporating feedback from observed miscoverage, using ground-truth outcomes $y_t$ revealed sequentially. Building on this idea, several recent variants enhance ACI via smooth gradient-based updates (Bhatnagar et al., 2023) or adaptive weighting schemes informed by recent prediction losses (Gibbs & Candès, 2024). To overcome the limitations of binary feedback, more recent work introduces continuous feedback mechanisms, such as PID control (Angelopoulos et al., 2023) and error-aware quantile functions (Wu et al., 2025), enabling finer-grained calibration of uncertainty. Nevertheless, a fundamental limitation of these approaches is their dependence on the quality of the backbone model's predicted intervals: when the initial intervals are poorly estimated, subsequent calibration is also likely to be inaccurate. Moreover, because these methods primarily rely on miscoverage rates as feedback for adjusting quantiles, they offer limited control over interval width, often resulting in overly conservative prediction intervals.

## E. Extended Experiments

### E.1. Experimental Details

#### E.1.1. DATASETS

To quantitatively evaluate DistMatch, we use five datasets: electricity consumption (Elec.), solar radiation (Solar), wind energy (Wind), Meta stock prices (META), and NVIDIA stock prices (NVDA). In addition, we include the Weather dataset to assess DistMatch's performance on extreme val-

ues across diverse domains.

For the dataset setup, we follow the implementation of Xu & Xie (2021). The Solar dataset consists of hourly solar radiation measurements from Atlanta in 2018, obtained from the National Solar Radiation Database (NSRDB)[2], where the radiation level is used as the target variable. The Wind dataset includes hourly wind power generation data from the Hackberry Wind Farm in Austin in 2019[3], with wind energy generation as the target variable. The Elec. dataset corresponds to electricity pricing data collected at 30-minute intervals between 1996 and 1999 from the Elec2 dataset[4], capturing electricity transfers between New South Wales and Victoria in Australia, which serve as the target variable.

To evaluate DistMatch under highly non-stationary conditions, we additionally collect four months of 5-minute-level stock price data for Meta (META) and Nvidia (NVDA), spanning from 2024/12/06 to 2025/04/02. The dataset includes Open, High, Low, Close, and Volume, along with technical indicators such as RSI, momentum, Bollinger Bands, and moving averages, with the Close price used as the target variable.

Lastly, to evaluate performance across diverse time-series domains with extreme values, we include a daily Weather dataset[5] spanning from 2020/01/01 to 2025/08/01. The target variable is precipitation (rain). It exhibits heavy-tailed target distributions with extreme values and is used to assess the robustness of DistMatch in capturing such outliers.

#### E.1.2. SYNTHETIC DATASET

The synthetic data consists of three components with two predefined clusters, as follows:

**Piecewise trend.** For $m \in \{0, 1, 2, 3, 4\}$ and local index $u \in \{0, \ldots, 279\}$ with $t = 280m + u$, the trend $\mu_t$ is

$$\mu_t = \begin{cases} -1, & 0 \le u < 80, \\ 4, & 80 \le u < 160, \\ 4 - 0.1\,(u - 160), & 160 \le u < 240, \\ -4 + 0.2\,(u - 240), & 240 \le u < 280. \end{cases} \quad (85)$$

Let $T$ be the total length. Define a base 280-length motif repeated 5 times:

$$T = 5(80 + 80 + 80 + 40) = 1400. \quad (86)$$

**AR(1) noise.** Let $\eta_t \sim \mathcal{N}(0, \sigma^2)$ i.i.d. with $\sigma = 0.2$ and $\phi = 0.8$. Define $e_0 \sim \mathcal{N}(0, \sigma^2)$ and

$$e_t = \phi e_{t-1} + \eta_t, \qquad t = 1, \ldots, T - 1. \quad (87)$$

[2]https://nsrdb.nrel.gov/
[3]https://github.com/Duvey314/austin-green-energypredictor
[4]https://www.kaggle.com/yashsharan/the-elec2-dataset
[5]https://weather.com/

*Table 5.* Optimized hyperparameters for each method across datasets in Table 1.

| Data | DistMatch | | KOWCPI | HopCPT | SPCI | EnbPI | NexCP |
|---|---|---|---|---|---|---|---|
| Elec. | node_split_threshold $(\gamma) = 0.05$ 
 patch_window_size$(w) = 100$ | | bandwidth $(h) = 8$ | | | | |
| Solar | node_split_threshold $(\gamma) = 0.005$ 
 patch_window_size $(w) = 100$ | | bandwidth $(h) = 8$ | learning_rate $= 0.001$ 
 drop_out $= 0.5$ 
 time_encode $= 400$ | window_ size $= 100$ | window_ size $= 100$ | decay $(\rho) = 0.99$ |
| Wind | node_split_threshold $(\gamma) = 0.1$ 
 patch_window_size $(w) = 100$ | | bandwidth $(h) = 8$ | | | | |
| META | node_split_threshold $(\gamma) = 0.1$ 
 patch_window_size $(w) = 100$ | | bandwidth $(h) = 5$ | | | | |
| NVDA | node_split_threshold $(\gamma) = 0.1$ 
 patch_window_size $(w) = 100$ | | bandwidth $(h) = 3$ | | | | |

*Table 6.* Hyperparameter ranges used for tuning.

| Method | Hyperparameter | Value |
|---|---|---|
| DistMatch | node_split_ threshold $(\gamma)$ | 0.005, 0.01, 0.025, 0.05, 0.075, 0.1 |
| | patch_window_size $(w)$ | 5, 10, 25, 50, 100, 150 |
| HopCPT | learning_rate | 0.001, 0.01 |
| | drop_out | 0, 0.25, 0.5 |
| | time_encode | yes, no |
| SPCI | window_size | 10, 25, 50, 75 100, 125, 150, 200 |
| EnbPI | window_size | 10, 25, 50, 75 100, 125, 150, 200 |
| NexCP | decay $(\rho)$ | 0.90, 0.95, 0.98, 0.99 0.993, 0.995, 0.999 |
| KOWCPI | bandwidth $(h)$ | 8, 10, 12, 15, 20 |

**Grouped Gaussian noise.** Partition $\{0, \ldots, T - 1\}$ into $G = 3$ contiguous groups with sizes $n_g \in \{\lfloor T/G \rfloor, \lceil T/G \rceil\}$ (first groups take the remainder), and for each $t$ in group $g$,

$$
\begin{aligned}
g_t &\sim \mathcal{N}(m_g, s_g^2), \\
(m_1, m_2, m_3) &= (-0.2, 0, 0.2), \\
(s_1, s_2, s_3) &= (0.1, 0.3, 0.2),
\end{aligned}
\tag{88}
$$

independently across $t$ (conditional on the grouping).

**Observed series.** The (noise-added) series is

$$
x_t = \mu_t + e_t + g_t, \qquad t = 0, \ldots, T - 1. \tag{89}
$$

### E.1.3. HYPERPARAMETERS

As shown in Table 6, DistMatch has two primary tunable hyperparameters: the KS threshold $\gamma$ and the patch window size $w$. Other hyperparameters—such as the number of trees $B$, the bootstrapping ratio $\theta$, and the minimum number of samples per leaf $n_{\min}$—are fixed to $B = 10$, $\theta = 0.9$,

and $n_{\min} = 0$. Empirically, most leaf nodes (except for the leftmost) contain, on average, more than 10 samples, rendering the effect of $n_{\min}$ negligible; thus, it is set to its minimal default value 0. For the tunable hyperparameters, the node-splitting threshold $\gamma$ is selected via greedy search on the calibration set across all datasets, and the patch length $w$ is set to 100 for a conservative upper bound to satisfy Theorem 5.6. For all baseline methods, hyperparameters are also tuned via greedy search, except when optimal settings are already reported in Auer et al. (2023). The optimized hyperparameters used for Table 1 are reported in Table 5.

### E.1.4. POINT PREDICTORS

To demonstrate the robustness of DistMatch across arbitrary point predictors—a core goal of CP—we evaluate it using three models: Random Forest (Breiman, 2001), Light Gradient Boosting Machine (LGBM) (Ke et al., 2017), and Temporal Convolutional Network (TCN) (Lea et al., 2017). The first two are tree-based ensemble methods, whereas the third is a deep neural network. Random Forest constructs an ensemble of decision trees in which each internal node represents a data-driven decision rule. Although LGBM is also tree-based, it differs in its tree construction strategy by employing Gradient-based One-Side Sampling (GOSS) and Exclusive Feature Bundling (EFB) to efficiently select splits, rather than relying on naive IG maximization. In contrast, TCN is a residual deep network built on causal convolutions, making it particularly effective at capturing temporal dependencies in time series data.

### E.1.5. EVALUATION METRICS

The two primary metrics for evaluating conformal prediction methods are coverage and interval width. Coverage measures whether the ground-truth value lies within the prediction interval, while interval width quantifies the tightness of that interval. Following Auer et al. (2023), we allow a miscoverage fluctuation margin of $0.25\alpha$. Ultimately, we compare conformal prediction methods based on

their ability to construct narrow prediction intervals while satisfying the coverage constraint, since excessively wide intervals—although achieving high coverage—have limited practical utility.

In addition to coverage and interval width, we use the Winkler score (WS) (Winkler, 1972; Auer et al., 2023) to jointly account for both interval width and coverage in a single metric, defined as:

$$\text{WS}_\alpha = \text{IW} + \frac{2}{\alpha} \cdot \lambda, \tag{90}$$

$$\lambda = \begin{cases} y - \mathcal{C}_\alpha^{\text{u}} & \text{if } y > \mathcal{C}_\alpha^{\text{u}}, \\ \mathcal{C}_\alpha^{\text{l}} - y & \text{if } y < \mathcal{C}_\alpha^{\text{l}}, \\ 0 & \text{otherwise}. \end{cases} \tag{91}$$

Here, $\mathcal{C}_\alpha^{\text{u}}$ and $\mathcal{C}_\alpha^{\text{l}}$ denote the upper and lower bounds of the prediction region, respectively, and the interval width is defined as:

$$\text{IW}_t^\alpha(x_{t+1}) = \mathcal{C}_\alpha^{\text{u}}(x_{t+1}) - \mathcal{C}_\alpha^{\text{l}}(x_{t+1}). \tag{92}$$

The Winkler score increases when the prediction interval is either far from the ground truth or excessively wide, and is maximized when both occur simultaneously, indicating ineffective uncertainty quantification despite nominal coverage. Following Auer et al. (2023), we report the normalized Winkler score, computed over normalized prediction regions.

### E.2. Additional Experiments

#### E.2.1. QUANTILE REGRESSION

*Table 7.* Comparison between DistMatch and SPCI using a *Quantile Linear* as the quantile regressor. RF is used as the point predictor, and the significance level is set to $\alpha = 0.1$.

| Model | DistMatch | | SPCI | |
|---|---|---|---|---|
| Metric | Cov. ↑ | Width ↓ | Cov. ↑ | Width ↓ |
| Elec. | **0.90** | **0.54** | 0.18 | 0.13 |

While DistMatch effectively clusters residuals into approximately exchangeable subsets within each leaf, it requires a quantile regressor for final quantile estimation. Following SPCI (Xu & Xie, 2023), we adopt Quantile Regression Forests (QRF) and justify their suitability as follows. Transformer-based quantile models (Lee et al., 2024), for instance, apply learned nonlinear mappings that do not preserve the original residual distribution. Such transformations can disrupt the approximate exchangeability achieved by DistMatch's binning of distributionally homogeneous samples. Similarly, although Flow Matching (Lee et al., 2026) guarantees the marginal distribution, it adapts the conditional flow at each time point based on the specific historical context. Consequently, it becomes order-sensitive and violates the exchangeability assumption required for

conformal prediction. Quantile Linear Regression (Koenker & Bassett Jr, 1978), on the other hand, estimates quantiles asymmetrically via the pinball loss but relies on a linear mapping of potentially nonlinear residual distributions, leading to substantial information loss (Meinshausen, 2006). In contrast, QRF preserves the local empirical distribution within each leaf—formed through DistMatch's KS-based binning—by employing tree-based partitioning without imposing parametric transformations. Therefore, QRF serves as a natural and well-aligned quantile estimator for DistMatch, as it maintains the original distributional structure within each approximately exchangeable leaf. Specifically, we additionally note that the key difference between QRF and reweighting lies in whether they are used to enforce approximate exchangeability. In reweighting methods, weights are assigned globally across heterogeneous calibration samples to approximate exchangeability itself. In contrast, DistMatch ensures approximate exchangeability prior to QRF via KS-based binning without weighting. The QRF weights are then used solely to estimate the future residual, not to correct for exchangeability.

As shown in Table 7, we evaluate the performance of Quantile Linear Regression in place of QRF for both DistMatch and SPCI. While DistMatch still meets the target coverage, it yields wider prediction intervals compared to the QRF-based results. In contrast, SPCI fails to achieve the target coverage, with a substantial drop in empirical coverage, which limits its practical applicability across different choices of quantile regressors.

#### E.2.2. COMPARISON WITH ONLINE CONFORMAL INFERENCE METHODS

*Table 8.* Comparison between DistMatch and the online conformal inference method, Adaptive Conformal Inference (ACI). LGBM is used as the backbone model, as it supports both point prediction and uncertainty estimation, and the significance level is set to $\alpha = 0.1$.

| Model | DistMatch | | ACI (2021) | |
|---|---|---|---|---|
| Metric | Cov. ↑ | Width ↓ | Cov. ↑ | Width ↓ |
| Elec. | 0.93 | **0.25** | 0.94 | 0.53 |
| Solar | 0.89 | **85.71** | 0.89 | 93.25 |
| Wind | 0.89 | **60.75** | 0.90 | 134.07 |

To assess DistMatch against online conformal inference methods, we compare it with Adaptive Conformal Inference (ACI) (Gibbs & Candes, 2021). As shown in Table 8, ACI produces prediction intervals that are more than twice as wide as those of DistMatch in Elec. and Wind, despite achieving a comparable coverage rate. This empirical result supports the observation that uncertainty calibration–based methods heavily depend on the quality of the backbone model's uncertainty estimates and often fail to produce meaningfully tight intervals, as discussed in Appendix Sec-

*Table 9.* Quantitative evaluation of sequential CP algorithms on Weather (Rain) datasets. The significance level $\alpha$ is set to 0.1. We utilize Random Forest (Forest), LightGBM (LGBM), and TCN for point predictors. The smallest (best) widths and Winkler (Win.) scores are highlighted with **bold** font. The "Cov." with a miscoverage rate higher than $0.25\alpha$ are not considered and are thus greyed out.

| | UC Model | | DistMatch | KOWCPI | HopCPT | SPCI | EnbPI | NexCP | CP | Copula |
|---|---|---|---|---|---|---|---|---|---|---|
| Rain | Forest | Win. ↓ | $\mathbf{6.80^{\pm 0.25}}$ | $7.19^{\pm 0.00}$ | $7.30^{\pm 0.54}$ | $7.29^{\pm 0.04}$ | 7.45 | 7.06 | 7.04 | 7.05 |
| | | Cov. ↑ | $0.88^{\pm 0.01}$ | $0.86^{\pm 0.00}$ | $0.91^{\pm 0.03}$ | $0.83^{\pm 0.01}$ | 0.85 | 0.89 | 0.90 | 0.90 |
| | | Width ↓ | $\mathbf{0.06^{\pm 0.00}}$ | $0.06^{\pm 0.00}$ | $0.08^{\pm 0.02}$ | $0.05^{\pm 0.00}$ | 0.05 | 0.06 | 0.07 | 0.07 |
| | LGBM | Win. ↓ | $5.24^{\pm 0.05}$ | $5.10^{\pm 0.00}$ | $\mathbf{4.95^{\pm 0.09}}$ | $5.13^{\pm 0.02}$ | 5.34 | 5.00 | 5.02 | 5.04 |
| | | Cov. ↑ | $0.89^{\pm 0.00}$ | $0.90^{\pm 0.00}$ | $0.90^{\pm 0.01}$ | $0.85^{\pm 0.01}$ | 0.85 | 0.88 | 0.90 | 0.90 |
| | | Width ↓ | $\mathbf{0.04^{\pm 0.00}}$ | $0.05^{\pm 0.00}$ | $0.06^{\pm 0.00}$ | $0.04^{\pm 0.00}$ | 0.04 | 0.05 | 0.05 | 0.05 |
| | TCN | Win. ↓ | $4.90^{\pm 0.07}$ | $64.44^{\pm 0.07}$ | $\mathbf{4.52^{\pm 0.12}}$ | $4.91^{\pm 0.02}$ | 4.95 | 4.81 | 4.94 | 4.86 |
| | | Cov. ↑ | $0.89^{\pm 0.00}$ | $0.90^{\pm 0.01}$ | $0.92^{\pm 0.01}$ | $0.86^{\pm 0.00}$ | 0.86 | 0.90 | 0.93 | 0.92 |
| | | Width ↓ | $\mathbf{0.04^{\pm 0.00}}$ | $0.05^{\pm 0.00}$ | $0.05^{\pm 0.01}$ | $0.04^{\pm 0.00}$ | 0.04 | **0.04** | 0.05 | 0.05 |

tion D. Note that ACI is clipped at every step to ensure that $\alpha_t \in [0, 1]$. Whenever $\alpha_t$ goes beyond this range, the clipped value is used in the subsequent step.

### E.2.3. ADDITIONAL DATASETS ON EXTREME VALUES

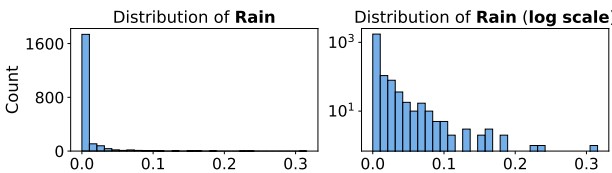

*Figure 9.* The distributions of Weather (Rain) dataset exhibit non-normality and heavy tails. The right plots show the same data as the left, with the log scale of the $y$-axis.

To evaluate DistMatch under extreme-value regimes, we additionally conduct experiments on the Weather dataset, predicting rainfall. As shown in Figure 9, the target variables exhibit heavy-tailed, non-normal distributions. Quantitatively, as reported in Table 9, HopCPT and DistMatch show competitive performance. HopCPT attains higher empirical coverage but at the cost of substantially wider prediction intervals. In contrast, DistMatch maintains the target coverage while producing considerably narrower intervals.

### E.2.4. OOD-LEAF ANALYSIS

To empirically validate Proposition 5.12, we analyze the leftmost leaf across all datasets. In Table 10, the $\mathcal{H}$-ratio denotes the within-leaf homogeneity of the leftmost leaf, while the OOD-ratio represents the proportion of out-of-distribution (OOD) samples relative to the total calibration set. Table 10 shows that datasets can be grouped by the magnitude of their $\mathcal{H}$-ratio according to their characteristics, and that the more nonstationary the data, the lower the $\mathcal{H}$-ratio of the leftmost leaf. As shown in Proposition 5.12, this means that the more nonstationary the data, the lower the probability that an unseen sample is assigned to the matched

*Table 10.* The table presents: 1) $\mathcal{H}$-ratio: Homogeneity ratio (ratio of values under the similar distribution in the leftmost node), 2) $K$-leaves: average number of leaves per tree in a bootstrapped DistMatch, and 3) OOD-ratio: $n_{\text{ood}} / n_{\text{calib}}$. G denotes the group clustered by homogeneity ratio, where (a), (b), and (c) correspond to each group.

| G | Data | $\mathcal{H}$-**ratio** | $K$-**Leaves** | **OOD-ratio (%)** |
|---|---|---|---|---|
| (a) | Solar | 0.93 | 5.10 | 0.04(0.30/800) |
| | Rain | 0.88 | 6.50 | 0.10(0.80/816) |
| (b) | Elec. | 0.64 | 26.10 | 0.41(5.70/1378) |
| | Wind | 0.46 | 55.70 | 0.00(0.00/3500) |
| (c) | META | 0.27 | 38.50 | 0.96(19.2/2000) |
| | NVDA | 0.13 | 65.50 | 1.57(38.4/2445) |

leaf among the $B$ bootstrapped trees; at the same time, this keeps the other (right) leaves homogeneous.

### E.2.5. TRADE-OFF ANALYSIS ON CALIBRATION SIZE

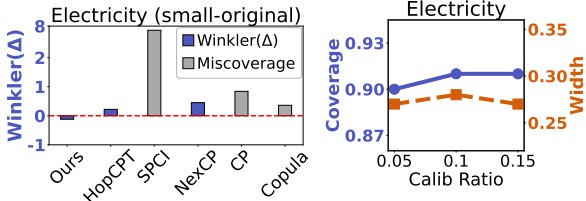

*Figure 10.* The left subplot shows Winkler score differences (log-scaled) between half and full datasets, and the right shows performance under varying calibration ratios. Both use an RF predictor at $\alpha = 0.1$.

To assess robustness under data scarcity, we evaluate DistMatch on half-sized datasets and under varying calibration ratios, while keeping the training set fixed. As shown in Figure 10, despite more dispersed residuals in the half-sized setting, DistMatch maintains reliable coverage with competitive Winkler scores, whereas SPCI, Copula, and standard CP fail to meet the coverage target, and HopCPT and NexCP exhibit higher Winkler scores. When varying the calibra-

tion ratio, DistMatch continues to preserve both coverage and interval width even with substantially fewer calibration samples.

### E.2.6. TRADE-OFF ANALYSIS ON WITHIN-LEAF SAMPLE SIZE

*Table 11.* Comparison between the original DistMatch, DistMatch-E (early-discarding variant), and DistMatch-O (optimal-sample variant). The significance level is set to $\alpha$ is set to 0.1, and an RF point predictor is used.

| | Model | DistMatch | DistMatch-E | DistMatch-O |
|---|---|---|---|---|
| **Elec.** | Win. ↓ | **1.97** | 2.03 | 2.12 |
| | Cov. ↑ | 0.92 | 0.92 | 0.92 |
| | Width ↓ | **0.27** | 0.28 | 0.32 |
| **Solar** | Win. ↓ | **1.54** | 1.57 | 1.64 |
| | Cov. ↑ | 0.91 | 0.91 | 0.90 |
| | Width ↓ | **60.00** | 64.82 | 59.22 |
| **Wind** | Win. ↓ | 2.15 | **2.13** | 2.19 |
| | Cov. ↑ | 0.90 | 0.89 | 0.86 |
| | Width ↓ | 69.04 | **67.35** | 65.27 |

We study the trade-off between limiting the number of samples ($\leq 300$) retained within each leaf and using the full set of residuals without restriction. The early-discarding variant is denoted as DistMatch-E, while the variant that selects an optimal subset under a fixed leaf sample budget is denoted as DistMatch-O. As shown in Table 11, retaining all residuals within each leaf yields the best overall performance, whereas early discarding achieves the best results on the *Wind* dataset and second best for other datasets. Importantly, DistMatch-E achieves better performance than DistMatch-O, demonstrating that recent samples are often more informative than similar samples, a fact that many existing methods implicitly rely on.

### E.2.7. ANALYSIS ON RETRAINING DISTMATCH IN ONLINE DEPLOYMENT

DistMatch operates robustly under severe distribution shifts in online deployment and does not require frequent updates. As a result, the training cost is incurred only once, after which it enables efficient, low-cost inference.

To empirically validate, we compare two variants of DistMatch with different online update strategies. All variants are evaluated on the same five datasets as in Table 1, under identical settings.

- **DistMatch-update:** The entire matching tree is retrained from scratch every time 300 unseen samples accumulate.

- **DistMatch-leaf only update:** The learned tree structure is fixed; however, as samples accumulate within a leaf, any sample satisfying Eq. (7)—i.e., qualifying as

a new split anchor—can trigger further partitioning of that leaf, enabling local structural adaptation without full tree retraining.

*Table 12.* Comparison between the DistMatch updating strategy variants in the online deployment setting. The significance level $\alpha$ is set to 0.1, and an RF point predictor is used.

| Model | DistMatch-default | | DistMatch-update | | DistMatch-leaf only update | |
|---|---|---|---|---|---|---|
| Metric | Cov. ↑ | Width ↓ | Cov. ↑ | Width ↓ | Cov. ↑ | Width ↓ |
| Elec. | 0.92 | **0.27** | 0.91 | **0.27** | 0.61 | 0.18 |
| Solar | 0.91 | **60.00** | 0.90 | 61.19 | 0.84 | 59.21 |
| Wind | 0.90 | 69.04 | 0.88 | **67.49** | 0.60 | 57.24 |
| META | 0.90 | **4.28** | 0.93 | 11.30 | 0.47 | 4.14 |
| NVDA | 0.90 | **2.71** | 0.96 | 3.55 | 0.55 | 1.22 |

As shown in Table 12, the fixed tree achieves the best performance. The updated variants do not yield improvements, as updates increase tree complexity and depth, increasing the variance term in Theorem 5.6 while leaving the bias unchanged. In contrast, the fixed tree balances tree complexity and leaf sample size by learning the tree under a limited calibration budget. This suggests that frequent retraining or structural updates are unnecessary, as fixed DistMatch already provides sufficient adaptability, as shown in Table 1 with META and NVDA.

### E.2.8. ANALYSIS ON MULTISTEP PREDICTION SETTINGS

For multi-step prediction, DistMatch extends naturally by replacing $\varepsilon_{t+1}$ with $\varepsilon_{t+h}$ for horizon $h$, while keeping the patch structure unchanged. As shown in Table 13, DistMatch outperforms SPCI in multistep prediction.

*Table 13.* Comparison between the DistMatch and SPCI in a multistep setting. The significance level $\alpha$ is set to 0.1, and an RF point predictor is used.

| Model | DistMatch | | SPCI | |
|---|---|---|---|---|
| Metric | Cov. ↑ | Width ↓ | Cov. ↑ | Width ↓ |
| Elec. | 0.90 | **0.28** | 0.89 | 0.29 |
| Solar | 0.90 | **73.16** | 0.83 | 64.78 |
| Wind | 0.90 | **69.80** | 0.83 | 61.55 |
| META | 0.88 | **12.49** | 0.60 | 10.27 |
| NVDA | 0.91 | **3.25** | 0.82 | 3.25 |

For multivariate prediction, since the KS statistic is defined for univariate distributions, DistMatch requires independent instances per output dimension. However, this is a shared limitation across sequential CP methods, as most are also designed for univariate settings. Extending DistMatch to joint multivariate distributions via multivariate nonparametric statistics remains an interesting direction for future work.

### E.2.9. EXTENDED RESULTS WITH VARIOUS SIGNIFICANCE LEVELS

We report extended results across significance levels and additional point predictors, including LGBM and TCN. The significance level $\alpha$ is the most critical hyperparameter in conformal prediction, as robustness across different values of $\alpha$ reflects a method's reliability in identifying valid prediction regions. To evaluate this robustness, we conduct experiments over multiple significance levels, $\alpha \in \{0.05, 0.10, 0.15\}$, with quantitative results reported in Tables 14, 15, and 16. In addition, qualitative results for $\alpha = 0.1$ are presented in Figures 11 and 12.

Across all settings, DistMatch consistently attains the target coverage while maintaining the narrowest or comparable prediction intervals. Its advantage is particularly pronounced on highly nonstationary datasets, where it achieves low miscoverage rates. For instance, at $\alpha = 0.05$, DistMatch is the only method that consistently satisfies the target coverage with meaningful interval widths on the *Solar* and *Stock* datasets.

## F. Limitation and Future Work

A key limitation of DistMatch is that the leftmost leaf often serves as an out-of-distribution (OOD) leaf, where quantile estimation performance may degrade. Although DistMatch leverages bootstrapping to promote generalization from a limited calibration set, it can become vulnerable when the unseen distribution undergoes substantial shifts. This observation motivates two promising directions for future work. First, we propose an online updating strategy that continuously monitors the $\mathcal{H}$-ratio of the OOD leaf. When the degree of partial OOD exceeds a predefined threshold, the leaf can be further split to restore distributional homogeneity, allowing the model to adapt to evolving test distributions flexibly. Second, we suggest learning to adaptively adjust the threshold used in the KS statistic, enabling DistMatch to handle a broader range of distributional patterns more robustly. Together, these directions aim to extend DistMatch with online adaptation mechanisms and improve its robustness under distributional shifts in the unseen set.

*Table 14.* Quantitative evaluation of sequential CP algorithms on Electricity consumption (Elec.), Solar radiation (Solar), Wind energy (Wind), META stock price (META), and NVIDIA stock price (NVDA) datasets. The significance level $\alpha$ is set to 0.1 for all experiments. We utilize Random Forest (Forest), LightGBM (LGBM), and TCN for point predictors. The smallest (best) widths and Winkler (Win.) scores are highlighted with **bold** font. The "Cov." with a miscoverage rate higher than $0.25\alpha$ are not considered and are thus greyed out.

| UC Model | | | DistMatch | KOWCPI | HopCPT | SPCI | EnbPI | NexCP | CP | Copula |
|---|---|---|---|---|---|---|---|---|---|---|
| Elec. | Forest | Win. ↓ | **1.97**$^{\pm 0.01}$ | 2.44$^{\pm 0.00}$ | 3.35$^{\pm 0.21}$ | 2.54$^{\pm 0.06}$ | 3.36 | 3.51 | 3.93 | 3.77 |
| | | Cov. ↑ | 0.92$^{\pm 0.00}$ | 0.90$^{\pm 0.00}$ | 0.91$^{\pm 0.02}$ | 0.90$^{\pm 0.00}$ | 0.85 | 0.92 | 0.97 | 0.96 |
| | | Width ↓ | **0.27**$^{\pm 0.00}$ | 0.38$^{\pm 0.00}$ | 0.50$^{\pm 0.05}$ | 0.28$^{\pm 0.00}$ | 0.45 | 0.57 | 0.71 | 0.67 |
| | LGBM | Win. ↓ | **1.73**$^{\pm 0.01}$ | 2.09$^{\pm 0.00}$ | 3.33$^{\pm 0.12}$ | 1.99$^{\pm 0.01}$ | 3.07 | 3.59 | 3.98 | 3.98 |
| | | Cov. ↑ | 0.93$^{\pm 0.01}$ | 0.89$^{\pm 0.00}$ | 0.91$^{\pm 0.01}$ | 0.93$^{\pm 0.00}$ | 0.84 | 0.92 | 0.92 | 0.94 |
| | | Width ↓ | **0.25**$^{\pm 0.00}$ | 0.30$^{\pm 0.00}$ | 0.53$^{\pm 0.02}$ | 0.29$^{\pm 0.00}$ | 0.41 | 0.60 | 0.63 | 0.66 |
| | TCN | Win. ↓ | **1.96**$^{\pm 0.01}$ | 2.48$^{\pm 0.00}$ | 2.86$^{\pm 0.13}$ | 2.06$^{\pm 0.01}$ | 2.78 | 3.78 | 4.00 | 3.99 |
| | | Cov. ↑ | 0.92$^{\pm 0.00}$ | 0.90$^{\pm 0.00}$ | 0.91$^{\pm 0.01}$ | 0.92$^{\pm 0.00}$ | 0.85 | 0.91 | 0.89 | 0.91 |
| | | Width ↓ | **0.24**$^{\pm 0.00}$ | 0.40$^{\pm 0.00}$ | 0.44$^{\pm 0.02}$ | 0.28$^{\pm 0.00}$ | 0.35 | 0.61 | 0.55 | 0.60 |
| Solar. | Forest | Win. ↓ | **1.54**$^{\pm 0.01}$ | 2.07$^{\pm 0.00}$ | 1.90$^{\pm 0.13}$ | 1.98$^{\pm 0.15}$ | 2.55 | 3.74 | 3.56 | 3.54 |
| | | Cov. ↑ | 0.91$^{\pm 0.00}$ | 0.93$^{\pm 0.00}$ | 0.92$^{\pm 0.01}$ | 0.85$^{\pm 0.01}$ | 0.88 | 0.89 | 0.86 | 0.88 |
| | | Width ↓ | **60.00**$^{\pm 0.40}$ | 109.31$^{\pm 0.00}$ | 97.04$^{\pm 8.21}$ | 47.36$^{\pm 5.14}$ | 112.57 | 215.67 | 167.91 | 187.69 |
| | LGBM | Win. ↓ | **2.00**$^{\pm 0.01}$ | 1.87$^{\pm 0.00}$ | 2.73$^{\pm 0.18}$ | 3.00$^{\pm 0.15}$ | 2.76 | 4.76 | 7.12 | 5.23 |
| | | Cov. ↑ | 0.89$^{\pm 0.00}$ | 0.87$^{\pm 0.00}$ | 0.88$^{\pm 0.03}$ | 0.74$^{\pm 0.02}$ | 0.85 | 0.86 | 0.62 | 0.75 |
| | | Width ↓ | **85.71**$^{\pm 0.27}$ | 87.84$^{\pm 0.00}$ | 138.24$^{\pm 4.86}$ | 75.68$^{\pm 0.90}$ | 130.03 | 262.85 | 176.62 | 229.04 |
| | TCN | Win. ↓ | **1.94**$^{\pm 0.00}$ | 2.54$^{\pm 0.00}$ | 2.81$^{\pm 1.05}$ | 2.67$^{\pm 0.36}$ | 3.07 | 2.85 | 2.75 | 2.75 |
| | | Cov. ↑ | 0.88$^{\pm 0.00}$ | 0.90$^{\pm 0.00}$ | 0.91$^{\pm 0.02}$ | 0.79$^{\pm 0.02}$ | 0.87 | 0.89 | 0.89 | 0.89 |
| | | Width ↓ | **66.21**$^{\pm 0.00}$ | 105.47$^{\pm 0.24}$ | 154.04$^{\pm 50.57}$ | 56.11$^{\pm 3.49}$ | 103.25 | 136.94 | 115.03 | 115.94 |
| Wind. | Forest | Win. ↓ | **2.15**$^{\pm 0.01}$ | 3.54$^{\pm 0.00}$ | 3.72$^{\pm 0.31}$ | 2.19$^{\pm 0.00}$ | 4.37 | 3.98 | 4.40 | 4.10 |
| | | Cov. ↑ | 0.90$^{\pm 0.00}$ | 0.89$^{\pm 0.00}$ | 0.90$^{\pm 0.01}$ | 0.83$^{\pm 0.00}$ | 0.85 | 0.89 | 0.74 | 0.82 |
| | | Width ↓ | **69.04**$^{\pm 0.03}$ | 139.25$^{\pm 0.00}$ | 151.66$^{\pm 12.82}$ | 63.14$^{\pm 0.18}$ | 145.63 | 171.67 | 147.57 | 159.00 |
| | LGBM | Win. ↓ | **2.03**$^{\pm 0.01}$ | 3.85$^{\pm 0.00}$ | 3.01$^{\pm 0.35}$ | 2.08$^{\pm 0.00}$ | 3.68 | 4.23 | 4.51 | 4.34 |
| | | Cov. ↑ | 0.89$^{\pm 0.00}$ | 0.51$^{\pm 0.00}$ | 0.89$^{\pm 0.01}$ | 0.82$^{\pm 0.00}$ | 0.85 | 0.89 | 0.77 | 0.82 |
| | | Width ↓ | **60.75**$^{\pm 0.20}$ | 46.52$^{\pm 0.00}$ | 122.64$^{\pm 15.85}$ | 57.40$^{\pm 0.22}$ | 123.32 | 188.70 | 174.51 | 180.76 |
| | TCN | Win. ↓ | **2.05**$^{\pm 0.01}$ | 4.19$^{\pm 0.00}$ | 5.44$^{\pm 0.81}$ | 2.66$^{\pm 0.41}$ | 5.87 | 5.32 | 5.08 | 5.39 |
| | | Cov. ↑ | 0.89$^{\pm 0.00}$ | 0.91$^{\pm 0.00}$ | 0.87$^{\pm 0.02}$ | 0.86$^{\pm 0.03}$ | 0.82 | 0.90 | 0.81 | 0.90 |
| | | Width ↓ | **64.33**$^{\pm 0.20}$ | 173.35$^{\pm 0.00}$ | 207.37$^{\pm 28.94}$ | 76.77$^{\pm 6.50}$ | 188.37 | 219.21 | 188.85 | 215.85 |
| META | Forest | Win. ↓ | **0.12**$^{\pm 0.00}$ | 0.61$^{\pm 0.00}$ | 0.25$^{\pm 0.08}$ | **0.12**$^{\pm 0.00}$ | 0.19 | 0.25 | 0.29 | 0.27 |
| | | Cov. ↑ | 0.90$^{\pm 0.02}$ | 0.91$^{\pm 0.00}$ | 0.83$^{\pm 0.10}$ | 0.96$^{\pm 0.00}$ | 0.83 | 0.90 | 0.98 | 0.96 |
| | | Width ↓ | **4.28**$^{\pm 0.03}$ | 26.27$^{\pm 0.00}$ | 8.08$^{\pm 0.65}$ | 5.00$^{\pm 0.04}$ | 5.77 | 11.25 | 14.77 | 12.96 |
| | LGBM | Win. ↓ | **0.15**$^{\pm 0.00}$ | 0.54$^{\pm 0.00}$ | 0.31$^{\pm 0.00}$ | **0.15**$^{\pm 0.00}$ | 0.26 | 0.53 | 0.71 | 0.64 |
| | | Cov. ↑ | 0.91$^{\pm 0.01}$ | 0.94$^{\pm 0.00}$ | 0.88$^{\pm 0.00}$ | 0.95$^{\pm 0.00}$ | 0.78 | 0.90 | 1.00 | 0.99 |
| | | Width ↓ | **5.63**$^{\pm 0.05}$ | 26.90$^{\pm 0.00}$ | 12.86$^{\pm 0.11}$ | 6.93$^{\pm 0.02}$ | 7.46 | 26.24 | 38.08 | 34.00 |
| | TCN | Win. ↓ | **0.10**$^{\pm 0.00}$ | **0.10**$^{\pm 0.00}$ | 0.13$^{\pm 0.04}$ | 0.13$^{\pm 0.04}$ | 0.12 | 0.13 | 0.10 | 0.10 |
| | | Cov. ↑ | 0.88$^{\pm 0.00}$ | 0.91$^{\pm 0.00}$ | 0.92$^{\pm 0.01}$ | 0.87$^{\pm 0.01}$ | 0.87 | 0.91 | 0.92 | 0.91 |
| | | Width ↓ | **3.05**$^{\pm 0.00}$ | 3.25$^{\pm 0.01}$ | 4.71$^{\pm 1.77}$ | 4.07$^{\pm 1.30}$ | 3.42 | 5.23 | 3.69 | 3.58 |
| NVDA | Forest | Win. ↓ | **0.49**$^{\pm 0.00}$ | 0.72$^{\pm 0.00}$ | 1.59$^{\pm 0.01}$ | 1.05$^{\pm 0.00}$ | 1.01 | 1.42 | 2.94 | 2.94 |
| | | Cov. ↑ | 0.90$^{\pm 0.00}$ | 0.89$^{\pm 0.00}$ | 0.75$^{\pm 0.00}$ | 0.78$^{\pm 0.01}$ | 0.79 | 0.88 | 0.90 | 0.89 |
| | | Width ↓ | **2.71**$^{\pm 0.01}$ | 4.42$^{\pm 0.00}$ | 6.93$^{\pm 0.01}$ | 3.23$^{\pm 0.01}$ | 3.53 | 8.98 | 16.30 | 14.52 |
| | LGBM | Win. ↓ | **0.30**$^{\pm 0.00}$ | 0.57$^{\pm 0.00}$ | 0.64$^{\pm 0.00}$ | 0.36$^{\pm 0.01}$ | 0.55 | 0.56 | 0.79 | 0.63 |
| | | Cov. ↑ | 0.93$^{\pm 0.00}$ | 0.90$^{\pm 0.00}$ | 0.88$^{\pm 0.00}$ | 0.95$^{\pm 0.01}$ | 0.83 | 0.91 | 0.98 | 0.96 |
| | | Width ↓ | **1.68**$^{\pm 0.03}$ | 2.82$^{\pm 0.00}$ | 3.12$^{\pm 0.02}$ | 2.30$^{\pm 0.03}$ | 2.19 | 2.96 | 6.06 | 4.18 |
| | TCN | Win. ↓ | 0.33$^{\pm 0.00}$ | 0.32$^{\pm 0.00}$ | 0.32$^{\pm 0.07}$ | 1.53$^{\pm 1.86}$ | 0.25 | **0.28** | 1.44 | 0.59 |
| | | Cov. ↑ | 0.92$^{\pm 0.00}$ | 0.83$^{\pm 0.00}$ | 0.89$^{\pm 0.03}$ | 0.62$^{\pm 0.25}$ | 0.87 | 0.89 | 0.34 | 0.66 |
| | | Width ↓ | 1.79$^{\pm 0.01}$ | 1.18$^{\pm 0.00}$ | 1.58$^{\pm 0.77}$ | 1.35$^{\pm 0.24}$ | 0.99 | **1.29** | 1.18 | 2.05 |

*Table 15.* Quantitative evaluation of sequential CP algorithms on Electricity consumption (Elec.), Solar radiation (Solar), Wind energy (Wind), META stock price (META), and NVIDIA stock price (NVDA) datasets. The significance level $\alpha$ is set to 0.05 for all experiments. We utilize Random Forest (Forest), LightGBM (LGBM), and TCN for point predictors. The smallest (best) widths and Winkler (Win.) scores are highlighted with **bold** font. The "Cov." with a miscoverage rate higher than $0.25\alpha$ are not considered and are thus greyed out.

| | UC Model | | DistMatch | KOWCPI | HopCPT | SPCI | EnbPI | NexCP | CP | Copula |
|---|---|---|---|---|---|---|---|---|---|---|
| Elec. | Forest | Win. ↓ | **2.41**$^{\pm0.02}$ | 2.97$^{\pm0.00}$ | 3.77$^{\pm0.34}$ | 3.53$^{\pm0.14}$ | 3.83 | 3.87 | 3.98 | 3.97 |
| | | Cov. ↑ | 0.95$^{\pm0.00}$ | 0.93$^{\pm0.00}$ | 0.96$^{\pm0.01}$ | 0.93$^{\pm0.00}$ | 0.90 | 0.96 | 0.97 | 0.96 |
| | | Width ↓ | **0.34**$^{\pm0.00}$ | 0.44$^{\pm0.00}$ | 0.59$^{\pm0.05}$ | 0.37$^{\pm0.00}$ | 0.50 | 0.66 | 0.68 | 0.68 |
| | LGBM | Win. ↓ | **2.12**$^{\pm0.01}$ | 2.61$^{\pm0.00}$ | 3.66$^{\pm0.08}$ | 2.59$^{\pm0.02}$ | 3.39 | 3.93 | 4.46 | 4.48 |
| | | Cov. ↑ | 0.95$^{\pm0.00}$ | 0.91$^{\pm0.00}$ | 0.95$^{\pm0.01}$ | 0.96$^{\pm0.00}$ | 0.91 | 0.96 | 0.95 | 0.97 |
| | | Width ↓ | **0.31**$^{\pm0.00}$ | 0.34$^{\pm0.00}$ | 0.60$^{\pm0.01}$ | 0.40$^{\pm0.00}$ | 0.46 | 0.69 | 0.71 | 0.78 |
| | TCN | Win. ↓ | **2.35**$^{\pm0.02}$ | 2.80$^{\pm0.01}$ | 3.36$^{\pm0.12}$ | 2.60$^{\pm0.02}$ | 3.27 | 4.23 | 4.65 | 4.46 |
| | | Cov. ↑ | 0.94$^{\pm0.00}$ | 0.94$^{\pm0.00}$ | 0.96$^{\pm0.00}$ | 0.96$^{\pm0.00}$ | 0.90 | 0.96 | 0.92 | 0.96 |
| | | Width ↓ | **0.30**$^{\pm0.01}$ | 0.46$^{\pm0.00}$ | 0.52$^{\pm0.02}$ | 0.38$^{\pm0.00}$ | 0.41 | 0.71 | 0.63 | 0.75 |
| Solar. | Forest | Win. ↓ | **2.05**$^{\pm0.03}$ | 2.71$^{\pm0.00}$ | 2.08$^{\pm0.18}$ | 2.90$^{\pm0.37}$ | 3.14 | 3.68 | 3.72 | 3.69 |
| | | Cov. ↑ | 0.94$^{\pm0.00}$ | 0.95$^{\pm0.00}$ | 0.95$^{\pm0.01}$ | 0.88$^{\pm0.01}$ | 0.92 | 0.97 | 0.97 | 0.97 |
| | | Width ↓ | **81.22**$^{\pm1.09}$ | 134.92$^{\pm0.00}$ | 113.57$^{\pm6.80}$ | 64.71$^{\pm6.97}$ | 136.15 | 225.51 | 236.53 | 230.23 |
| | LGBM | Win. ↓ | **2.43**$^{\pm0.05}$ | 2.19$^{\pm0.00}$ | 3.11$^{\pm0.21}$ | 4.18$^{\pm0.12}$ | 3.21 | 5.10 | 7.10 | 5.62 |
| | | Cov. ↑ | 0.94$^{\pm0.00}$ | 0.91$^{\pm0.00}$ | 0.92$^{\pm0.02}$ | 0.80$^{\pm0.01}$ | 0.91 | 0.92 | 0.67 | 0.86 |
| | | Width ↓ | **110.02**$^{\pm0.51}$ | 100.78$^{\pm0.00}$ | 160.95$^{\pm7.32}$ | 93.38$^{\pm1.35}$ | 145.81 | 294.64 | 232.84 | 262.50 |
| | TCN | Win. ↓ | **2.64**$^{\pm0.00}$ | 3.21$^{\pm0.04}$ | 3.30$^{\pm1.12}$ | 3.96$^{\pm0.61}$ | 3.66 | 3.34 | 3.29 | 3.26 |
| | | Cov. ↑ | 0.94$^{\pm0.00}$ | 0.95$^{\pm0.00}$ | 0.94$^{\pm0.02}$ | 0.84$^{\pm0.02}$ | 0.91 | 0.94 | 0.93 | 0.94 |
| | | Width ↓ | **95.42**$^{\pm0.00}$ | 146.45$^{\pm0.94}$ | 183.78$^{\pm59.36}$ | 74.56$^{\pm3.62}$ | 140.83 | 172.96 | 156.12 | 165.80 |
| Wind. | Forest | Win. ↓ | **2.61**$^{\pm0.03}$ | 3.82$^{\pm0.00}$ | 4.08$^{\pm0.27}$ | 2.68$^{\pm0.01}$ | 4.72 | 4.15 | 4.57 | 4.27 |
| | | Cov. ↑ | 0.94$^{\pm0.00}$ | 0.95$^{\pm0.00}$ | 0.95$^{\pm0.01}$ | 0.88$^{\pm0.00}$ | 0.90 | 0.95 | 0.85 | 0.91 |
| | | Width ↓ | **83.25**$^{\pm0.21}$ | 158.42$^{\pm0.00}$ | 170.69$^{\pm10.62}$ | 75.44$^{\pm0.17}$ | 163.59 | 183.19 | 164.72 | 174.67 |
| | LGBM | Win. ↓ | **2.50**$^{\pm0.00}$ | 6.48$^{\pm0.00}$ | 3.23$^{\pm0.27}$ | 2.56$^{\pm0.01}$ | 3.87 | 4.35 | 4.75 | 4.48 |
| | | Cov. ↑ | 0.94$^{\pm0.00}$ | 0.53$^{\pm0.00}$ | 0.94$^{\pm0.00}$ | 0.89$^{\pm0.00}$ | 0.90 | 0.95 | 0.83 | 0.88 |
| | | Width ↓ | **74.31**$^{\pm0.00}$ | 52.94$^{\pm0.00}$ | 138.35$^{\pm12.26}$ | 72.20$^{\pm0.37}$ | 138.23 | 195.74 | 183.41 | 189.16 |
| | TCN | Win. ↓ | **2.58**$^{\pm0.11}$ | 4.51$^{\pm0.00}$ | 5.94$^{\pm0.84}$ | 3.25$^{\pm0.55}$ | 6.33 | 5.79 | 5.58 | 5.89 |
| | | Cov. ↑ | 0.94$^{\pm0.00}$ | 0.94$^{\pm0.00}$ | 0.92$^{\pm0.02}$ | 0.91$^{\pm0.02}$ | 0.88 | 0.95 | 0.88 | 0.94 |
| | | Width ↓ | **82.35**$^{\pm3.11}$ | 194.58$^{\pm0.00}$ | 233.94$^{\pm29.23}$ | 91.52$^{\pm6.37}$ | 208.71 | 245.43 | 201.84 | 239.63 |
| META | Forest | Win. ↓ | **0.16**$^{\pm0.01}$ | 0.71$^{\pm0.00}$ | 0.33$^{\pm0.11}$ | 0.16$^{\pm0.00}$ | 0.24 | 0.30 | 0.38 | 0.32 |
| | | Cov. ↑ | 0.95$^{\pm0.01}$ | 0.94$^{\pm0.00}$ | 0.89$^{\pm0.10}$ | 0.97$^{\pm0.00}$ | 0.89 | 0.95 | 0.99 | 0.98 |
| | | Width ↓ | **5.49**$^{\pm0.09}$ | 31.63$^{\pm0.00}$ | 10.79$^{\pm0.13}$ | 6.00$^{\pm0.01}$ | 7.09 | 13.24 | 19.91 | 16.12 |
| | LGBM | Win. ↓ | **0.18**$^{\pm0.00}$ | 0.62$^{\pm0.00}$ | 0.36$^{\pm0.00}$ | 0.19$^{\pm0.00}$ | 0.31 | 0.58 | 0.87 | 0.74 |
| | | Cov. ↑ | 0.95$^{\pm0.00}$ | 0.97$^{\pm0.00}$ | 0.93$^{\pm0.00}$ | 0.98$^{\pm0.00}$ | 0.86 | 0.95 | 1.00 | 1.00 |
| | | Width ↓ | **7.07**$^{\pm0.06}$ | 31.32$^{\pm0.00}$ | 15.19$^{\pm0.06}$ | 8.42$^{\pm0.06}$ | 8.83 | 28.63 | 47.04 | 40.12 |
| | TCN | Win. ↓ | **0.13**$^{\pm0.00}$ | 0.13$^{\pm0.00}$ | 0.16$^{\pm0.05}$ | 0.16$^{\pm0.04}$ | 0.15 | 0.16 | 0.14 | **0.13** |
| | | Cov. ↑ | 0.94$^{\pm0.00}$ | 0.96$^{\pm0.00}$ | 0.96$^{\pm0.00}$ | 0.94$^{\pm0.00}$ | 0.92 | 0.96 | 0.97 | 0.96 |
| | | Width ↓ | **3.91**$^{\pm0.00}$ | 4.27$^{\pm0.03}$ | 6.44$^{\pm2.24}$ | 5.34$^{\pm1.65}$ | 4.35 | 6.60 | 5.22 | 4.83 |
| NVDA | Forest | Win. ↓ | **0.66**$^{\pm0.01}$ | 0.89$^{\pm0.00}$ | 1.76$^{\pm0.01}$ | 1.60$^{\pm0.00}$ | 1.15 | 1.61 | 3.65 | 3.58 |
| | | Cov. ↑ | 0.95$^{\pm0.00}$ | 0.94$^{\pm0.00}$ | 0.83$^{\pm0.00}$ | 0.84$^{\pm0.00}$ | 0.86 | 0.92 | 0.92 | 0.92 |
| | | Width ↓ | **3.32**$^{\pm0.03}$ | 5.60$^{\pm0.00}$ | 7.98$^{\pm0.00}$ | 3.72$^{\pm0.01}$ | 4.19 | 10.41 | 18.76 | 17.22 |
| | LGBM | Win. ↓ | **0.39**$^{\pm0.00}$ | 0.74$^{\pm0.00}$ | 0.84$^{\pm0.00}$ | 0.44$^{\pm0.00}$ | 0.77 | 0.74 | 1.00 | 0.90 |
| | | Cov. ↑ | 0.96$^{\pm0.00}$ | 0.96$^{\pm0.00}$ | 0.94$^{\pm0.00}$ | 0.98$^{\pm0.00}$ | 0.90 | 0.96 | 0.99 | 0.99 |
| | | Width ↓ | **2.10**$^{\pm0.02}$ | 4.17$^{\pm0.00}$ | 4.25$^{\pm0.02}$ | 2.79$^{\pm0.03}$ | 2.98 | 4.14 | 7.81 | 6.74 |
| | TCN | Win. ↓ | 0.43$^{\pm0.00}$ | 0.41$^{\pm0.00}$ | 0.41$^{\pm0.07}$ | 2.82$^{\pm3.71}$ | 0.36 | **0.38** | 2.18 | 0.65 |
| | | Cov. ↑ | 0.96$^{\pm0.00}$ | 0.92$^{\pm0.00}$ | 0.95$^{\pm0.02}$ | 0.66$^{\pm0.27}$ | 0.92 | 0.94 | 0.37 | 0.82 |
| | | Width ↓ | 2.19$^{\pm0.01}$ | 1.57$^{\pm0.00}$ | 2.02$^{\pm0.89}$ | 1.62$^{\pm0.21}$ | 1.33 | **1.72** | 1.55 | 2.47 |

*Table 16.* Quantitative evaluation of sequential CP algorithms on Electricity consumption (Elec.), Solar radiation (Solar), Wind energy (Wind), META stock price (META), and NVIDIA stock price (NVDA) datasets. The significance level $\alpha$ is set to 0.15 for all experiments. We utilize Random Forest (Forest), LightGBM (LGBM), and TCN for point predictors. The smallest (best) widths and Winkler (Win.) scores are highlighted with **bold** font. The "Cov." with a miscoverage rate higher than $0.25\alpha$ are not considered and are thus greyed out.

| | UC Model | | DistMatch | KOWCPI | HopCPT | SPCI | EnbPI | NexCP | CP | Copula |
|---|---|---|---|---|---|---|---|---|---|---|
| Elec. | Forest | Win. ↓ | $\mathbf{1.71^{\pm0.01}}$ | $2.19^{\pm0.00}$ | $3.07^{\pm0.14}$ | $2.07^{\pm0.05}$ | 3.01 | 3.38 | 3.46 | 3.44 |
| | | Cov. ↑ | $0.88^{\pm0.01}$ | $0.87^{\pm0.00}$ | $0.86^{\pm0.02}$ | $0.86^{\pm0.01}$ | 0.80 | 0.87 | 0.90 | 0.89 |
| | | Width ↓ | $\mathbf{0.23^{\pm0.00}}$ | $0.34^{\pm0.00}$ | $0.45^{\pm0.04}$ | $\mathbf{0.23^{\pm0.00}}$ | 0.41 | 0.54 | 0.58 | 0.57 |
| | LGBM | Win. ↓ | $\mathbf{1.53^{\pm0.01}}$ | $1.85^{\pm0.00}$ | $3.12^{\pm0.17}$ | $1.69^{\pm0.01}$ | 2.85 | 3.33 | 3.62 | 3.61 |
| | | Cov. ↑ | $0.88^{\pm0.00}$ | $0.86^{\pm0.00}$ | $0.86^{\pm0.01}$ | $0.91^{\pm0.00}$ | 0.80 | 0.89 | 0.88 | 0.90 |
| | | Width ↓ | $\mathbf{0.21^{\pm0.00}}$ | $0.27^{\pm0.00}$ | $0.47^{\pm0.02}$ | $0.24^{\pm0.00}$ | 0.37 | 0.53 | 0.54 | 0.57 |
| | TCN | Win. ↓ | $\mathbf{1.68^{\pm0.00}}$ | $2.21^{\pm0.00}$ | $2.56^{\pm0.10}$ | $1.72^{\pm0.01}$ | 2.57 | 3.48 | 3.61 | 3.60 |
| | | Cov. ↑ | $0.89^{\pm0.00}$ | $0.86^{\pm0.00}$ | $0.87^{\pm0.00}$ | $0.90^{\pm0.00}$ | 0.82 | 0.86 | 0.83 | 0.87 |
| | | Width ↓ | $\mathbf{0.20^{\pm0.00}}$ | $0.35^{\pm0.00}$ | $0.38^{\pm0.01}$ | $0.22^{\pm0.00}$ | 0.32 | 0.53 | 0.46 | 0.50 |
| Solar. | Forest | Win. ↓ | $\mathbf{1.34^{\pm0.01}}$ | $1.78^{\pm0.00}$ | $1.70^{\pm0.09}$ | $1.57^{\pm0.09}$ | 2.26 | 3.12 | 3.11 | 3.11 |
| | | Cov. ↑ | $0.86^{\pm0.01}$ | $0.92^{\pm0.00}$ | $0.89^{\pm0.01}$ | $0.82^{\pm0.02}$ | 0.84 | 0.85 | 0.84 | 0.85 |
| | | Width ↓ | $48.84^{\pm0.62}$ | $92.69^{\pm0.00}$ | $81.84^{\pm8.43}$ | $\mathbf{37.96^{\pm3.79}}$ | 96.99 | 163.79 | 153.50 | 158.96 |
| | LGBM | Win. ↓ | $1.72^{\pm0.01}$ | $\mathbf{1.71^{\pm0.00}}$ | $2.48^{\pm0.17}$ | $2.51^{\pm0.06}$ | 2.56 | 4.53 | 6.60 | 5.00 |
| | | Cov. ↑ | $0.86^{\pm0.00}$ | $0.83^{\pm0.00}$ | $0.83^{\pm0.03}$ | $0.68^{\pm0.01}$ | 0.80 | 0.80 | 0.57 | 0.68 |
| | | Width ↓ | $\mathbf{71.97^{\pm0.09}}$ | $78.42^{\pm0.00}$ | $121.92^{\pm5.26}$ | $65.76^{\pm1.35}$ | 117.00 | 241.77 | 138.30 | 200.55 |
| | TCN | Win. ↓ | $\mathbf{1.61^{\pm0.00}}$ | $2.09^{\pm0.01}$ | $2.50^{\pm0.92}$ | $2.12^{\pm0.26}$ | 2.54 | 2.52 | 2.36 | 2.37 |
| | | Cov. ↑ | $0.84^{\pm0.00}$ | $0.85^{\pm0.00}$ | $0.87^{\pm0.02}$ | $0.75^{\pm0.03}$ | 0.81 | 0.84 | 0.85 | 0.84 |
| | | Width ↓ | $\mathbf{56.81^{\pm0.00}}$ | $78.59^{\pm0.20}$ | $134.49^{\pm42.16}$ | $46.19^{\pm3.88}$ | 80.51 | 109.36 | 93.61 | 92.98 |
| Wind. | Forest | Win. ↓ | $\mathbf{1.91^{\pm0.00}}$ | $3.34^{\pm0.00}$ | $3.49^{\pm0.33}$ | $1.94^{\pm0.00}$ | 4.24 | 3.85 | 4.31 | 3.97 |
| | | Cov. ↑ | $0.85^{\pm0.00}$ | $0.85^{\pm0.00}$ | $0.86^{\pm0.02}$ | $0.77^{\pm0.00}$ | 0.80 | 0.84 | 0.65 | 0.74 |
| | | Width ↓ | $\mathbf{58.82^{\pm0.12}}$ | $125.15^{\pm0.00}$ | $137.75^{\pm13.58}$ | $55.03^{\pm0.17}$ | 129.82 | 161.90 | 132.58 | 147.28 |
| | LGBM | Win. ↓ | $\mathbf{1.76^{\pm0.01}}$ | $2.97^{\pm0.00}$ | $2.84^{\pm0.36}$ | $1.83^{\pm0.01}$ | 3.57 | 4.15 | 4.39 | 4.25 |
| | | Cov. ↑ | $0.85^{\pm0.00}$ | $0.47^{\pm0.00}$ | $0.84^{\pm0.01}$ | $0.74^{\pm0.00}$ | 0.79 | 0.83 | 0.72 | 0.76 |
| | | Width ↓ | $\mathbf{52.53^{\pm0.21}}$ | $41.75^{\pm0.00}$ | $111.41^{\pm17.53}$ | $49.06^{\pm0.20}$ | 109.07 | 181.94 | 164.77 | 172.01 |
| | TCN | Win. ↓ | $\mathbf{1.86^{\pm0.06}}$ | $3.92^{\pm0.00}$ | $5.09^{\pm0.78}$ | $2.34^{\pm0.31}$ | 5.58 | 5.02 | 4.79 | 5.07 |
| | | Cov. ↑ | $0.85^{\pm0.01}$ | $0.88^{\pm0.00}$ | $0.82^{\pm0.03}$ | $0.82^{\pm0.03}$ | 0.75 | 0.85 | 0.77 | 0.84 |
| | | Width ↓ | $\mathbf{58.57^{\pm2.38}}$ | $156.54^{\pm0.00}$ | $187.49^{\pm28.70}$ | $67.76^{\pm6.44}$ | 170.92 | 202.55 | 179.63 | 198.86 |
| META | Forest | Win. ↓ | $\mathbf{0.10^{\pm0.00}}$ | $0.53^{\pm0.00}$ | $0.21^{\pm0.06}$ | $0.11^{\pm0.00}$ | 0.17 | 0.23 | 0.26 | 0.24 |
| | | Cov. ↑ | $0.85^{\pm0.01}$ | $0.89^{\pm0.00}$ | $0.78^{\pm0.09}$ | $0.93^{\pm0.00}$ | 0.76 | 0.85 | 0.95 | 0.91 |
| | | Width ↓ | $\mathbf{3.62^{\pm0.05}}$ | $22.18^{\pm0.00}$ | $6.69^{\pm0.65}$ | $4.38^{\pm0.03}$ | 4.96 | 10.35 | 12.72 | 11.33 |
| | LGBM | Win. ↓ | $\mathbf{0.13^{\pm0.00}}$ | $0.49^{\pm0.00}$ | $0.28^{\pm0.00}$ | $0.14^{\pm0.00}$ | 0.23 | 0.51 | 0.65 | 0.58 |
| | | Cov. ↑ | $0.86^{\pm0.00}$ | $0.90^{\pm0.00}$ | $0.84^{\pm0.01}$ | $0.91^{\pm0.01}$ | 0.71 | 0.86 | 0.99 | 0.94 |
| | | Width ↓ | $\mathbf{4.76^{\pm0.03}}$ | $23.83^{\pm0.00}$ | $11.57^{\pm0.08}$ | $6.19^{\pm0.05}$ | 6.58 | 24.91 | 34.73 | 30.31 |
| | TCN | Win. ↓ | $\mathbf{0.08^{\pm0.00}}$ | $\mathbf{0.08^{\pm0.00}}$ | $0.11^{\pm0.04}$ | $0.11^{\pm0.04}$ | 0.10 | 0.11 | 0.09 | 0.09 |
| | | Cov. ↑ | $0.84^{\pm0.00}$ | $0.87^{\pm0.00}$ | $0.87^{\pm0.01}$ | $0.80^{\pm0.01}$ | 0.82 | 0.87 | 0.87 | 0.87 |
| | | Width ↓ | $\mathbf{2.57^{\pm0.00}}$ | $2.69^{\pm0.02}$ | $3.87^{\pm1.55}$ | $3.38^{\pm1.05}$ | 2.85 | 4.43 | 3.02 | 2.96 |
| NVDA | Forest | Win. ↓ | $\mathbf{0.45^{\pm0.01}}$ | $0.64^{\pm0.00}$ | $1.45^{\pm0.01}$ | $0.84^{\pm0.00}$ | 0.92 | 1.33 | 2.59 | 2.58 |
| | | Cov. ↑ | $0.84^{\pm0.01}$ | $0.83^{\pm0.00}$ | $0.67^{\pm0.00}$ | $0.74^{\pm0.00}$ | 0.72 | 0.82 | 0.88 | 0.85 |
| | | Width ↓ | $\mathbf{2.07^{\pm0.01}}$ | $3.74^{\pm0.00}$ | $6.12^{\pm0.01}$ | $2.90^{\pm0.02}$ | 3.08 | 8.14 | 14.94 | 12.53 |
| | LGBM | Win. ↓ | $\mathbf{0.25^{\pm0.00}}$ | $0.49^{\pm0.00}$ | $0.55^{\pm0.00}$ | $0.32^{\pm0.01}$ | 0.47 | 0.47 | 0.57 | 0.50 |
| | | Cov. ↑ | $0.90^{\pm0.00}$ | $0.83^{\pm0.00}$ | $0.82^{\pm0.00}$ | $0.89^{\pm0.01}$ | 0.78 | 0.86 | 0.96 | 0.92 |
| | | Width ↓ | $\mathbf{1.39^{\pm0.01}}$ | $2.29^{\pm0.00}$ | $2.61^{\pm0.15}$ | $2.01^{\pm0.05}$ | 1.82 | 2.43 | 4.02 | 3.08 |
| | TCN | Win. ↓ | $0.28^{\pm0.00}$ | $0.27^{\pm0.00}$ | $0.28^{\pm0.07}$ | $1.09^{\pm1.17}$ | 0.22 | $\mathbf{0.24}$ | 1.10 | 0.56 |
| | | Cov. ↑ | $0.89^{\pm0.00}$ | $0.76^{\pm0.00}$ | $0.83^{\pm0.04}$ | $0.56^{\pm0.25}$ | 0.81 | 0.84 | 0.33 | 0.54 |
| | | Width ↓ | $1.52^{\pm0.01}$ | $0.97^{\pm0.00}$ | $1.34^{\pm0.70}$ | $1.16^{\pm0.22}$ | 0.82 | $\mathbf{1.10}$ | 0.96 | 1.76 |

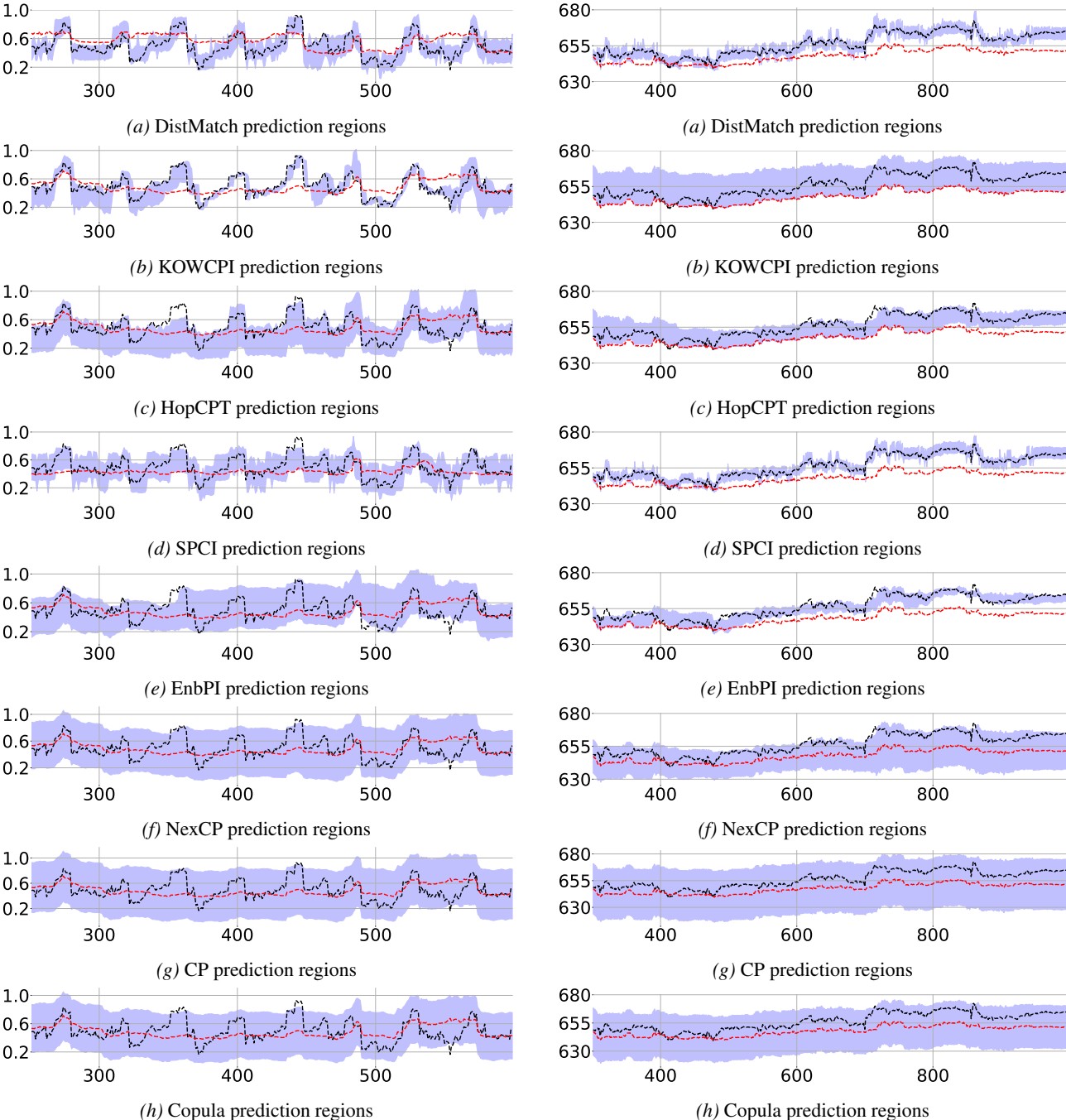

*Figure 11.* Qualitative evaluation of the DistMatch, KOWCPI, HopCPT, SPCI, EnbPI, NexCP, CP, and Copula on *Elec.* Dataset with $\alpha = 0.1$. DistMatch provides narrower regions while maintaining target coverage by accounting for distribution shifts and abrupt changes.

*Figure 12.* Qualitative evaluation of the DistMatch, HopCPT, SPCI, EnbPI, NexCP, CP, and Copula on META Dataset with target $\alpha = 0.1$, and LightGBM point predictor. DistMatch provides narrower regions while maintaining target coverage by accounting for distribution shifts and abrupt changes.

