# OpenReview forum: "DistMatch: Adaptive Binning  via Distribution Matching for Robust Sequential Conformal Prediction"
_ICML.cc/2026/Conference — ICML 2026 regular_

### Official Review · Reviewer_PDh4 · 2026-03-04

**Soundness:** 3
**Presentation:** 2
**Significance:** 2
**Originality:** 3
**Overall Recommendation:** 4
**Confidence:** 3

**Summary:**

The paper addresses the problem of computing Conformal Prediction sets in the non-exchangeable time-dependent setup. The proposed algorithm learns a tree to select past observations similar to the test point. The tree is grown using the Kolmogorov–Smirnov statistic between sliding windows at different positions as an objective function.

**Compliance With Llm Reviewing Policy:**

Affirmed.

**Key Questions For Authors:**

- Why does the data need to be sequential? Would a similar approach be possible for general distribution shifts?

- The earth mover distance is also density-free. Could it be used in the proposed scheme?

- The tree is trained without using the unseen test residual. How is exchangeability between past and present guaranteed? Is there any regularity assumption on the time-varying distributions?

- What is the difference between binning and weighting? Can binning be seen as an extreme case of weighting, where weights take values in $\{0, 1 \}$?

- The key motivation for the method, compared to reweighting approaches, is to preserve the empirical distribution. But patch-dependent weights are introduced in Equation 8. Why is the claimed patch-invariance better than standard sample-by-sample reweighting?

**Limitations:**

yes

**Strengths And Weaknesses:**

**Strengths**
- The tree-based approach with a distributional objective can be used more broadly in the CP community.
- Experimental results show good empirical improvement compared to existing methods.

**Weaknesses**
- It is unclear whether the proposed tree-based grouping aims to find locally i.i.d. or locally exchangeable groups.
- The intuitive explanation of the method [1] in the introduction could be improved. For example, the authors could define the idea of *pairing a residual with a patch* more precisely.
- Kernel approaches are criticized because practitioners need to choose the bandwidth. But the proposed method also has a free hyperparameter.
- The KS distance is density-free but requires sorting the data. The authors should discuss the time complexity of their approach.
- After binning, the algorithm resembles Mondrian Conformal Prediction. Specifying what is special about the time-dependent setup would help appreciate the paper’s contribution.
- The main method comparison includes naive (non-adaptive) CP but not ACI, which is the standard baseline for online CP [2].

[1] *DistMatch first pairs each patch residual with a corresponding target residual and performs partitioning based on patches. [...] DistMatch ensures local approximate exchangeability at the target level via patch-to-target propagation under patchwise local $\beta$-mixing and patch-to-target smoothness condition, thereby enabling valid coverage.*
[2] Gibbs and Candes (2021).

---

> ### Author Rebuttal · Authors · 2026-03-31
>
> We thank Reviewer **PDh4** for the thoughtful review.
>
> 1. **locally exchangeable groups**:  As clarified in Definition 5.1, our goal is to construct locally exchangeable groups, since temporal dependence and distribution shifts make local i.i.d. overly restrictive in practice. Notably, i.i.d. is a special case ($\gamma=0$), while allowing $\gamma>0$ accommodates a mild shift between samples.
> 2. **Patch-Target sentence**: We will revise the description as follows:
> Specifically, DistMatch represents the local distributional context at each time step using a residual patch $\tilde{\varepsilon_t}=$ { $\varepsilon_{t-w+1}, \ldots, \varepsilon_t$}, a sliding window $w$ of recent residuals. Each patch is paired with its subsequent target residual $\varepsilon_{t+1}$, capturing the conditional relationship between recent and future residual behavior.
> 3. **Kernel approach limitation:** We acknowledge that DistMatch has hyperparameters, namely $\gamma$ and $w$, but these parameters operate transparently with clear interpretations and directly control the coverage error via Lemma B.5. In contrast, in kernel-based approaches, the bandwidth does not transparently control coverage, and even a small misspecification can assign non-negligible weights to unrelated samples. Please also refer to our responses to Reviewers **mR1N** and **DeFN** for details on hyperparameter settings.
> 4. **Sorting complexity**: As discussed in Table 3 and Section 6.5, the sorting cost is negligible in practice: patch-level ECDFs are precomputed once in $\mathcal{O}(n w \log w)$ and cached, reducing each comparison to $\mathcal{O}(w)$, and this cost is incurred only during calibration.
> 5. **Mondrian Conformal Prediction:** While DistMatch shares the high-level idea of binning with Mondrian CP, the two methods differ fundamentally in two aspects. First, Mondrian CP requires global exchangeability and a symmetric score function, whereas DistMatch operates under local $\beta$-mixing and uses asymmetric signed residuals to capture directional quantile information. Next, Mondrian CP does not exploit temporal structure, whereas DistMatch leverages patch–target pairs to capture local temporal dependence, with coverage guaranteed via patch-to-target propagation under Assumption 5.2. These distinctions make DistMatch specifically designed for the sequential, non-exchangeable setting.
> 6. **ACI comparison:** We note that ACI is already discussed in Appendix D as part of our extended related work, and an empirical comparison with DistMatch is provided in Appendix E.2.2 (Table 8). The results show that DistMatch achieves comparable coverage while producing prediction intervals up to twice as narrow as ACI across all datasets.
> 7. **Sequential setting:** The sequential structure is integral to both the design and theoretical guarantees of DistMatch. The patch–target pair $(\tilde{\varepsilon_t}, \varepsilon_{t+1})$ leverages temporal adjacency, where the current patch informs the distribution of the next residual, formalized via the $\beta$-mixing condition in Assumption 5.2. However, as discussed in Section 1 (paragraph 3), binning-based approaches have been actively studied in non-sequential CP settings. In this regard, our framework can be extended to general distribution shift settings by replacing temporal pairs with alternative context representations.
> 8. **Earth mover distance:** Earth Mover’s Distance is equivalent to the 1D Wasserstein distance, which we have already discussed in Appendix C and evaluated in Figure 7-(a). As shown, the KS statistic achieves a better Winkler score while incurring significantly lower computational cost.
> 9. **Calibration-test shift**: Please refer to our responses to Reviewer **7RTn** for distribution shift, Reviewer **DeFN** for online deployment, and Reviewer **mR1N** for periodicity.
> 10. **Is binning an extreme version of weighting?:** Formally, binning can be treated as a discrete form of reweighting where weights take values in ${0,1}$: a sample receives weight 1 if it is assigned to the selected leaf and 0 otherwise. However, our structure differs fundamentally from continuous weighting, as it “does not estimate weights”. Please refer to our response to Reviewer **mR1N** for a detailed comparison between binning and retrieval weights.
> 11. **Difference between QRF and weighting:** The key difference between QRF and reweighting lies in whether they are used to enforce approximate exchangeability. In reweighting methods, weights are assigned globally across heterogeneous calibration samples to approximate exchangeability itself. In contrast, DistMatch ensures approximate exchangeability prior to QRF via KS-based binning without weighting. The QRF weights are then used solely to estimate the future residual, “not to correct for exchangeability”.

---

> > ### Author Rebuttal · Reviewer_PDh4 · 2026-04-02
> >
> > I am happy with your answers and tend to keep my positive rating.
> >
> > Is ACI clipped in your implementation?  What happens when $\alpha_t$ temporarily assumes meaningless values?

---

> > > ### Author Response · Authors · 2026-04-04
> > >
> > > We thank Reviewer **PDh4** for maintaining the positive rating.
> > >
> > > The ACI is clipped at every step to ensure that $\alpha_t \in [0, 1]$. Whenever $\alpha_t$ goes beyond this range, the clipped value is used in the subsequent step.

---

### Official Review · Reviewer_DeFN · 2026-03-08

**Soundness:** 2
**Presentation:** 3
**Significance:** 2
**Originality:** 3
**Overall Recommendation:** 3
**Confidence:** 3

**Summary:**

This paper proposes DistMatch, a sequential conformal prediction framework for time series based on distribution matching over residual patches. The main idea is to group historical residual patch-target pairs into approximately exchangeable local bins using KS-based matching trees, and then perform leaf-wise quantile inference within the matched bin for online prediction interval construction. The paper also explicitly considers the online setting by designing a fixed matching tree together with leaf-wise updates, which is practically relevant.

**Compliance With Llm Reviewing Policy:**

Affirmed.

**Key Questions For Authors:**

Questions:
- How robust is the method when the calibration segment is substantially shorter?
- In long online deployment, would removing leaf samples and trimming or updating tree structure produce better results? Maybe this can be a baseline in the ablation study.
- How would the framework extend to multi-step or multivariate settings?

**Limitations:**

yes

**Strengths And Weaknesses:**

Strengthes:

- The proposed method is generally novel, and the motivation is clear to me. The idea of using patches to as a local distribution tackles the hardness of estimating the distribution on single samples.
- The online setting is explicitly considered and designed for.
- The writing is easy to follow and comprehensively presented.

Weakness:

- Reliance on long residual patches (w=100 in this paper). This may limit application on shorter series or small calibration budget. Meanwhile, w can not be too small to maintain the KS statistics and matching algorithm valid.
- Sensitivity to hyper-paramter. Beyond w, the KS threshold γ also determines the quality of algorithms, and needs to be optimized using only the calibration set.
- Higher computational cost. The tree construction is computationally heavier than some baselines, and the quantile regressor needs to be trained on every leaf. Although the authors designs online adaptations cheap, the overall overhead is heavy.
- Potential weakness of the online design. Fixed tree structure is favorable for fast online computation, but may limit adaptation under emerging distribution shift over residuals. Also, the historical samples accumulate within leaves, which may create both computational and statistical issues over long deployments.

---

> ### Author Rebuttal · Authors · 2026-03-31
>
> We thank Reviewer **DeFN** for the thoughtful review.
>
> 1. **Small patch and calibration size:**
>     1. **Patch:** We adopt $w=100$ as a conservative default to satisfy Theorem 5.5, but smaller patch sizes can be used in practice. Figure 8 shows that $C$ remains bounded across $w$, and Figure 7 demonstrates strong performance even with $w=25$ on highly non-stationary datasets. Please also refer to our response to Reviewer **7RTn** for the validity of the $\beta$-mixing assumption.
>     2. **Calibration:** DistMatch does not require a large calibration set for training; we use only 10% of the full dataset as the calibration set. In addition, as discussed in Figure 10 (Appendix E.2.5), DistMatch maintains robust performance under limited training and calibration budgets (noisier residuals), as well as under varying calibration sizes with a fixed training set.
> 2. **Online deployment:**
>     1. **Summary**: DistMatch operates robustly under severe distribution shifts in online deployment and does not require frequent updates. As a result, the training cost is incurred only once, after which it enables efficient, low-cost inference.
>     2. **Online updating**: We compare two variants of DistMatch with different online update strategies. All variants are evaluated on the same five datasets as in Table 1, under identical settings.
>         - DistMatch-update: The entire matching tree is retrained from scratch every time 300 unseen samples accumulate.
>         - DistMatch-leaf only update: The learned tree structure is fixed; however, as samples accumulate within a leaf, any sample satisfying Eq. 7—i.e., qualifying as a new split anchor—can trigger further partitioning of that leaf, enabling local structural adaptation without full tree retraining.
>
>         As shown in the table below, the fixed tree achieves the best performance. The updated variants do not yield improvements, as updates increase tree complexity and depth, increasing the variance term in Theorem 5.5 while leaving the bias unchanged. In contrast, fixed tree balances tree complexity and leaf sample size by learning the tree under a limited calibration budget. This suggests that frequent retraining or structural updates are unnecessary, as fixed DistMatch already provides sufficient adaptability (as shown in Table 1 with META and NVDA). Please also refer to our response to Reviewer **mR1N** regarding periodicity and to Reviewer **7RTn** regarding distribution shifts.
>
>         |  | DistMatch-default | DistMatch-update | DistMatch-leaf only update |
>         | --- | --- | --- | --- |
>         | Elec. | 1.97 / 0.92 / 0.27 | 2.02 / 0.91 / 0.27 | 3.78 / 0.61 / 0.18 |
>         | Solar | 1.54 / 0.91 / 60.00 | 1.67 / 0.90 / 61.19 | 1.84 / 0.84 / 59.21 |
>         | Wind | 2.15 / 0.90 / 69.04 | 2.11 / 0.88 / 67.49 | 4.28 / 0.60 / 57.24 |
>         | META | 0.12 / 0.90 / 4.28 | 0.25 / 0.93 / 11.30 | 0.26 / 0.47 / 4.14 |
>         | NVDA | 0.49 / 0.90 / 2.71 | 0.51 / 0.96 / 3.55 | 0.63 / 0.55 / 1.22 |
>
>         * Winkler / Coverage / Width
>
>     3. **Computational cost**: We acknowledge that tree construction is more computationally intensive than some baselines; however, as shown above, frequent updates are unnecessary, making it a one-time training cost.
>
>         For QRF, since it operates in a leaf-wise manner, we do not retrain all QRFs at every step; instead, only the QRF corresponding to the leaf to which an unseen sample is assigned is updated, making it similar to SPCI. Regarding the accumulation of samples in DistMatch leaves, we have conducted an ablation study. As shown in Table 11 (Appendix E.2.6), DistMatch and DistMatch-E (discarding early samples) achieve nearly identical performance, indicating that lightweight leaves can be maintained without sacrificing accuracy while reducing QRF cost.
>
> 3. **Multistep or multivariate settings:**
>     1. For multi-step prediction, DistMatch extends naturally by replacing $\varepsilon_{t+1}$ with $\varepsilon_{t+h}$ for horizon $h$, while keeping the patch structure unchanged. As shown in the Table below, DistMatch outperforms SPCI in multistep prediction.
>
>         |  | DistMatch | SPCI |
>         | --- | --- | --- |
>         | Elec. | 2.17 / 0.90 / 0.28 | 2.32 / 0.89 / 0.29 |
>         | Solar | 1.91 / 0.90 / 73.16 | 2.74 / 0.83 / 64.78 |
>         | Wind | 2.13 / 0.90 / 69.80 | 2.24 / 0.83 / 61.55 |
>         | META | 0.30 / 0.88 / 12.49 | 0.57 / 0.60 / 10.27 |
>         | NVDA | 0.52 / 0.91 / 3.25 | 1.00 / 0.82 / 3.25 |
>
>     2. For multivariate prediction, since the KS statistic is defined for univariate distributions, DistMatch requires independent instances per output dimension. However, this is a shared limitation across sequential CP methods, as most are also designed for univariate settings. Extending DistMatch to joint multivariate distributions via multivariate nonparametric statistics remains an interesting direction for future work.
>
> 4. **Hyper-parameter:** Please refer to our response to Reviewer **mR1N**.

---

> > ### Author Rebuttal · Reviewer_DeFN · 2026-04-05
> >
> > Thank you for addressing my concerns. I have no further questions.

---

> > > ### Author Response · Authors · 2026-04-06
> > >
> > > We thank Reviewer **DeFN** for the constructive discussion.

---

### Official Review · Reviewer_7RTn · 2026-03-11

**Soundness:** 3
**Presentation:** 4
**Significance:** 3
**Originality:** 4
**Overall Recommendation:** 5
**Confidence:** 3

**Summary:**

This paper proposes DistMatch, which uses binary tree with Kolmogorov-Smirnov statistics and conduct quantile random forest on the leaf of the tree for sequential conformal prediction

**Compliance With Llm Reviewing Policy:**

Affirmed.

**Final Justification:**

The authors rebuttal was very helpful in addressing my concerns. One remaining point is that the authors introduced a new assumption during the rebuttal. I think this is acceptable, as it does not appear to change the direction of the other theoretical results in the paper, though this may require the judgment of the other reviewers or the AC. Since all my concerns have been addressed and the paper introducted novel approach with theoretical guarantees, I will raise my score to accept.

**Key Questions For Authors:**

- Assumption 5.2

Assumption 5.2, which imposes beta-mixing on patches of residuals derived from non-i.i.d. sequences, appears to be a strong assumption, yet it is not fully justified in the manuscript. A more thorough discussion of this assumption—including its motivation, practical plausibility, and the method’s sensitivity to possible violations—would substantially strengthen the paper.


- Sensitivity under distribution shift


Since quantile regression is performed within a memory (leaf), I suspect that DistMatch may struggle under distribution shift scenarios in which new patches are dissimilar to previously observed ones. Although the method updates leaves online, it may still require some time for the left-most leaf to be properly adapted. If multiple distribution shifts occur within a sequence, this update mechanism may be insufficient to keep the left-most leaf well calibrated throughout. In such settings, methods that directly update the quantile regression model itself may be more suitable.

This concern may already be reflected in the real-data example on Meta. Based on Figure 12, the sequence appears to exhibit a weak distributional shifts rather than repetitive patterns, which seems to represent a relatively weak setting for the proposed method. Consistent with this, the results show that SPCI performs competitively. Since distribution shift is a common phenomenon in sequential data, additional experiments specifically designed to evaluate this setting would substantially strengthen the paper.



- Minor typos

Reference [Gong et al., A kolmogorov-smirnov statistic based segmentation approach to learning from imbalanced datasets] is duplicated.

**Limitations:**

Yes

**Strengths And Weaknesses:**

- Soundness

The proposed method is theoretically sound and is supported by an intuitive motivation based on the use of the KS statistic.

- Presentation

The paper is well structured, and the figures effectively support the reader’s understanding. The appendix also provides useful additional details that help clarify the method.


- Significance

The paper addresses an important and actively studied problem: conformal prediction beyond exchangeability for sequential data.


- Originality

The method is novel in its use of the KS statistic and a growing binary tree to identify local patches of residuals.

---

> ### Author Rebuttal · Authors · 2026-03-31
>
> We thank Reviewer **7RTn** for the thoughtful review.
>
> 1. **$\beta$-mixing:**
>     1. **Summary**: Assumption 5.2 is reasonable because residual patch sequences with large $w$ inherit and often strengthen the mixing properties of the underlying process, leading to rapidly decaying dependence in practice.
>     2. **Practical plausibility.** Given a well-specified predictor $\phi$, residuals capture the noise component and inherit its mixing properties. The patch sequence benefits from aggregation, as each patch’s ECDF is estimated from $w$ samples, yielding more stable distributional comparisons and reducing the effective variance in the KS-based matching criterion.
>     3. **Sensitivity to violations.** Assumption 5.2 (i) enables the patch-to-target propagation bound in Theorem 5.5 and (ii) controls the variance term $\varrho_{\mathcal{L}}(\xi)$ via mixing concentration. If violated, slower decay increases the $\mathcal{O}(\sigma_{\text{mix}})$ term, leading to larger coverage error. DistMatch mitigates this by adopting a large patch size $w=100$ as the default, which empirically weakens inter-patch dependence and reduces the effective $\sigma_{\text{mix}}$ term.
>     4. **Empirical validation.** To support Assumption 5.2, we employ an ACF-based proxy to estimate the $\beta$-mixing coefficients $\hat{\beta}_1$ of the residual patch sequences {$\tilde{\varepsilon}_t$} on five representative datasets, using the same settings as in Table 1, for patch sizes $w \in$ {$5,10,25,50,100$}. As shown in the table below, $\hat{\beta}_1$ at $w=100$ is consistently lower than at $w=5$ across all datasets, confirming that larger patches induce weaker inter-patch temporal dependence. In addition, for datasets with moderate temporal dependence, shorter patch sizes already exhibit low $\hat{\beta}_1$ values, suggesting that $w=100$ is a conservative upper bound and that shorter patch sizes may suffice in practice. This is consistent with Figure 8, where $C$ remains bounded across a wide range of $w$, jointly validating Assumption 5.2.
>
>
>         |  | w=5 | w=10 | w=25 | w=50 | w=100 |
>         | --- | --- | --- | --- | --- | --- |
>         | Elec. | 1.000 | 0.743 | 0.845 | 0.640 | 0.441 |
>         | Wind | 1.000 | 0.793 | 0.301 | 0.490 | 0.268 |
>         | Solar | 0.483 | 0.689 | 0.655 | 0.264 | 0.124 |
>         | META | 1.000 | 1.000 | 0.796 | 0.422 | 0.100 |
>         | NVDA | 1.000 | 1.000 | 1.000 | 1.000 | 0.807 |
> 2. **DistMatch and SPCI under sever shifts**:
>     1. **Summary**: The left-most leaf in DistMatch differs from SPCI, as it contains only a small residual set of unmatched samples; even when a test sample falls into it, the bootstrap ensemble ensures matching to a right leaf in at least one tree, limiting fallback impact and maintaining stable coverage under severe distribution shifts.
>     2. **Left-most leaf**: We clarify that the left-most leaf in DistMatch is fundamentally distinct from SPCI, even though both use QRF-based online updates. In SPCI, updates are applied uniformly across the entire heterogeneous calibration set, mixing multiple distributions. In contrast, DistMatch routes samples satisfying Definition 5.1 to homogeneous right leaves, leaving the left-most leaf with only a small residual set of unmatched samples—empirically around 1% across most datasets (Table 10, Appendix E.2.4)—and thus operating on a much more concentrated subset than SPCI.
>
>         Furthermore, even when an unseen test sample is routed to the left-most leaf, DistMatch remains robust via the bootstrap ensemble (Appendix B.3). By constructing $B$ trees with diverse anchors, at least one tree routes the sample to a matched right leaf with probability $p_{\min}$, and the fallback error is scaled by $(1 - p_{\min})$ (Proposition B.4). As $B$ increases, anchor diversity improves, reducing worst-case fallback error and ensuring that overall coverage remains stable even under severe distribution shifts.
>
>         Please additionally refer to our response to Reviewer **mR1N** regarding periodicity (distribution shift).
>
>     3. **META results**: Regarding the META dataset, this distinction is clearly reflected in the quantitative results: SPCI achieves high coverage (0.96) but at the cost of substantially wider intervals (width: 5.00), whereas DistMatch attains the target coverage (0.90) with considerably tighter intervals (width: 4.28), as reported in Table 1. Figure 12 illustrates this trade-off qualitatively.
>     4. **Experiments**: As discussed in our response to Reviewer **DeFN**, we compare the fixed DistMatch tree with interval-based updates and leaf-wise updates. As shown in the table, the fixed tree achieves the best performance. This indicates that frequent retraining or structural updates are unnecessary, as the fixed DistMatch already provides sufficient adaptability even under multiple severe shifts (as shown in Table 1 with META and NVDA).
> 3. We will fix the typo.

---

> > ### Author Rebuttal · Reviewer_7RTn · 2026-04-02
> >
> > Thanks for your rebuttal and additional experiments.
> >
> > - Assumption 5.2
> >
> > The authors state that “Assumption 5.2 is reasonable because residual patch sequences with large $w$ inherit and often strengthen the mixing properties of the underlying process, leading to rapidly decaying dependence in practice.” However, as far as I understand, the paper does not explicitly assume mixing properties for the underlying process itself. If so, this justification would seem to require additional assumptions on either the underlying process or the residual process in order to be valid.
> >
> >
> > In addition, I find the reported ACF results across varying $w$ is helpful as an empirical indication that dependence decrease as the patch size increases. However, I do not think this is sufficient to empirically validate the $\beta$-mixing assumption, since ACF only captures a limited notion of dependence. Moreover, the reported values still appear fairly large, which makes it difficult to consider them as strong evidence supporting the assumption.

---

> > > ### Author Response · Authors · 2026-04-04
> > >
> > > We thank Reviewer **7RTn** for the insightful follow-up questions.
> > >
> > > 1. **Local stationarity**
> > >
> > >     We acknowledge that the paper did not explicitly state the mixing properties of the underlying process. To clarify, the fundamental assumption we implicitly rely on is **Local Stationarity** (Dahlhaus, 1997) [1], which is standard when sliding window analyses are applied to nonstationary time series.
> > >
> > >     Since the observed process is assumed to be locally stationary and the predictor $\phi$ is a smooth mapping, the residual process $\{\varepsilon_t = y_t - \phi(x_t)\}$ also exhibits smoothly varying statistical properties over time, thereby inheriting local stationarity. Under the Dahlhaus framework, this implies that the joint distribution of two well-separated, non-overlapping windows (or patches) factorizes in the limit as the total length $T \to \infty$.
> > >
> > >     While Dahlhaus's framework establishes asymptotic independence across rescaled time, it provides both physical and theoretical grounding for imposing a $\beta$-mixing assumption on the patch sequence {$\tilde{\varepsilon_t}$} in our finite-sample setting. Specifically, the asymptotic factorization of well-separated windows in Dahlhaus suggests that cross-window cumulants diminish as the distance increases. In a finite-sample setting, this behavior is reflected in the property that the joint distribution of two distant patches approaches the product of their marginals. Therefore, it suggests that the residual process does not exhibit infinitely long memory, thereby supporting the decay of the $\beta$-mixing coefficients $\beta(m)$ as the lag $m$ increases.
> > >
> > >     Even when patches are generated with overlap (e.g., via a sliding window with step size 1), the dependence between two patches $\tilde{\varepsilon_{t_1}}$ and $\tilde{\varepsilon_{t_2}}$ decays rapidly as the lag $m = |t_1 - t_2|$ grows larger than the patch size $w$. Once $m > w$, the two patches become non-overlapping and share no common residual elements. Beyond this point, the dependence is no longer driven by artificial overlap, but instead reflects only the underlying process's decaying memory. Therefore, under local stationarity, this remaining dependence diminishes as the separation between patches increases, which is consistent with the decaying dependence behavior required by $\beta$-mixing.
> > >
> > > 2. **TV-based evidence**
> > >
> > >     We also appreciate the reviewer’s point that ACF only captures linear dependence and may not be sufficient to validate $\beta$-mixing, which concerns the full distribution.
> > >
> > >     To provide stronger empirical evidence, we conduct non-parametric independence tests based on Total Variation (TV) distances. Specifically, to capture the full distribution of residual patches, each patch is mapped to its sorted quantile vector (serving as a sufficient statistic for the ECDF). We then discretize this high-dimensional space into a finite set of states using K-means clustering. Based on this discretization, we compute the TV distance between the empirical joint distribution and the product of marginals of two patches across various lags $m$. As shown in the table below, the TV distance generally decreases as the lag $m$ increases across various datasets and patch sizes, providing empirical support for the $\beta$-mixing condition.
> > >
> > >     |  | w=5 | w=10 | w=25 | w=50 | w=100 |
> > >     | --- | --- | --- | --- | --- | --- |
> > >     | Elec. | 0.409→0.221 | 0.218→0.272 | 0.269→0.259 | 0.376→0.231 | 0.333→0.000 |
> > >     | Wind | 0.348→0.123 | 0.268→0.167 | 0.109→0.101 | 0.213→0.126 | 0.101→0.052 |
> > >     | Solar | 0.161→0.172 | 0.195→0.186 | 0.329→0.257 | 0.286→0.047 | 0.222→0.000 |
> > >     | META | 0.471→0.189 | 0.370→0.146 | 0.252→0.080 | 0.140→0.091 | 0.106→0.047 |
> > >     | NVDA | 0.530→0.280 | 0.469→0.286 | 0.366→0.246 | 0.282→0.144 | 0.106→0.062 |
> > >
> > >     * Note: Values in the cells represent the TV distance at lag $m=1 \to$ lag $m=10$.
> > >
> > > 3. **Related work**
> > >
> > >     Local stationarity is not unique to DistMatch. SPCI explicitly cites [2] for locally stationary time series to justify its sliding-window QRF; KOWCPI applies a Nadaraya-Watson kernel whose validity rests on the assumption that nearby points in feature space share similar residual distributions — a local homogeneity condition equivalent to local stationarity; HopCPT assumes that past Hopfield-retrieved patterns remain predictive at test time, which requires the feature-residual relationship to be locally stable across time.
> > >
> > > 4. We will revise the manuscript to:
> > >     - Explicitly state that the underlying process is assumed to be locally stationary.
> > >     - Include the TV-based independence test results.
> > >
> > > [1] Dahlhaus, R. Fitting time series models to nonstationary processes. The Annals of Statistics, 25(1):1–37, 1997.
> > >
> > > [2] Zhou, Z. and Wu, W. B. Local linear quantile estimation for nonstationary time series. The Annals of Statistics, 37 (5B):2696–2729, 2009.

---

### Official Review · Reviewer_mR1N · 2026-03-12

**Soundness:** 3
**Presentation:** 3
**Significance:** 3
**Originality:** 4
**Overall Recommendation:** 4
**Confidence:** 3

**Summary:**

This paper focuses on the question of how to approximately recover residual exchangeability in sequential conformal prediction under non-stationary time series. The authors argue that existing methods rely heavily on reweighting, yet weight estimation is often inaccurate. They therefore propose DistMatch, which recursively constructs a binary matching tree based on the Kolmogorov–Smirnov (KS) statistic, assigns residual patches to a set of approximately exchangeable leaf nodes, and then performs local quantile regression and online updating within each leaf.

**Compliance With Llm Reviewing Policy:**

Affirmed.

**Final Justification:**

The authors answered my question well, and I considered raising the score to 4.

**Key Questions For Authors:**

1. How would the method perform on time series without periodicity?
2. Does each iteration necessarily produce a right leaf node?
3. The paper would benefit from a more explicit discussion of why KS-based binary partitioning is more stable than learned retrieval weights.

**Limitations:**

Yes. The authors have adequately discussed the limitations and potential negative societal impacts of their work.

**Strengths And Weaknesses:**

The core claim of the paper is that, rather than correcting the residual distribution through continuous weighting, one can directly construct locally approximately exchangeable subsets by binning or partitioning. The paper further argues that the KS statistic is a suitable criterion for both distribution matching and splitting, especially for skewed residuals.

The method consists of three main components: (1) residual patching; (2) recursive partitioning through a KS-based matching tree; and (3) leaf-wise quantile regression with online updates. The advantage of this design is that the roles of the components are clearly separated: patching captures local temporal information, the KS tree promotes distributional homogeneity, and the leaf-level regression enables adaptive inference.

The theoretical analysis is one of the most prominent strengths of the paper. The authors not only introduce a KS-based notion of local approximate exchangeability, but also attempt to show how matching at the patch level propagates to the target residual level. This goes beyond methods that are only empirically effective.

The experimental narrative is well aligned with the paper’s objective: the method outperforms existing sequential conformal prediction approaches on real datasets, with particular emphasis on robustness under severe distribution shift.

The core idea is clear and distinctive; the focus on robustness under shift is practically meaningful; and the idea of replacing weight estimation with binning is relatively novel.

The method may incur substantial computational and tuning overhead; and it may be sensitive to the threshold, patch size, and stopping criterion of the tree.

---

> ### Author Rebuttal · Authors · 2026-03-31
>
> We thank Reviewer **mR1N** for the thoughtful review.
>
> 1. **Hyper-parameter**:
>     1. **Summary**: Although DistMatch involves several hyperparameters, they operate transparently as discussed in the Theory section, and the method performs robustly within the suggested parameter range, which is also theoretically grounded.
>     2. **Threshold**: As shown in Figure 7-(a), DistMatch is robust to the choice of $\gamma$, as it balances the bias–variance trade-off in Theorem 5.5: increasing $\gamma$ relaxes the exchangeability bound (higher bias) while enlarging within-leaf sample sizes (lower variance), whereas decreasing $\gamma$ tightens the exchangeability bound (lower bias) but reduces within-leaf sample sizes (higher variance), resulting in stable Winkler scores over a wide range.
>     3. **Patch size**: For the patch size $w$, as shown in Figure 7-(a), all three datasets perform robustly at a patch size of 100, which we also theoretically justify in Theorem 5.5 and Figure 8. Please also refer to our responses to Reviewer **DeFN** regarding the patch and calibration sizes, and to Reviewer **7RTn** for the $\beta$-mixing assumption.
>     4. **Stopping criterion**: For the stopping criterion, we set $n_{\min}=0$ (Appendix E.1.3, line 1283); nevertheless, leaves contain, on average, more than 10 samples (Figure 8), since the matching criterion (Eq. 7) naturally prevents extremely small leaves from forming—a sample can only be routed to a leaf if it distributionally matches the anchor, which intrinsically limits fragmentation. For computational cost, please refer to our response to Reviewer **DeFN**.
> 2. **Periodicity**:
>     1. **Summary**: DistMatch operates effectively without requiring periodicity, due to the use of the $\gamma$-bound, bootstrapping, and online QRF updating.
>     2. **Under mild shift**: DistMatch does not rely on finding identical samples; instead, as defined in Definition 5.1, it admits any sample whose residual distribution lies within the $\gamma$-approximate exchangeability bound, allowing mild shifts within this bound across all right leaves. This enables robust handling of distributional differences between the calibration and test sets.
>     3. **Severe shift:** Even under severe distribution shifts, DistMatch maintains robustness through two complementary mechanisms discussed in Appendix B.3. First, DistMatch induces diverse split anchors (Assumption B.2) via building an ensemble of $B$ trees using subsampled calibration sets (ratio $\theta$). Even if an unseen patch fails to match in one tree, it can be matched in another. The final quantile is averaged across trees (Proposition B.4), stabilizing coverage.
>
>         Second, online QRF updating (Section 4.2.4): after each step, the observed residual $\varepsilon_{T+1}$ is added to the assigned leaf and the leaf-wise QRF is updated, enabling local quantiles to adapt to gradual distribution shifts without modifying the tree.
>
>         Together, anchor diversity maintains coverage, while online updates progressively adapt quantiles, preserving validity over the online sequence. Please additionally refer to our response to Reviewer **7RTn** for the left-most leaf.
>
>     4. **Experiments**: As discussed in our response to Reviewer **DeFN**, we compare the fixed DistMatch tree with interval-based updates and leaf-wise updates. As shown in the table, the fixed tree achieves the best performance. This indicates that frequent retraining or structural updates are unnecessary, as the fixed DistMatch already provides sufficient adaptability under distribution shifts (as shown in Table 1 with META and NVDA).
> 3. **Right leaf**: The formation of a right leaf node is ensured whenever at least one group in the calibration set satisfies the $\gamma$ threshold. Since $\gamma$ is a tunable hyperparameter, it can be adapted to the data, thereby avoiding degenerate cases. Please also refer to our response to Reviewer **7RTn** for the distinction between DistMatch and SPCI.
> 4. **Binning vs retrieval weights**:
>
>     First, binning preserves the empirical residual distribution. Retrieval-based reweighting uses continuous weights that can distort empirical quantiles and impair coverage, whereas binning treats samples uniformly within each leaf, preserving the distribution.
>
>     Second, binning avoids weight estimation error. Retrieval methods rely on similarity weights, where small errors can degrade quantile estimation—especially under abrupt shifts. KS-based partitioning replaces this with a binary routing rule using a nonparametric statistic.
>
>     Third, the KS bound provides explicit control over coverage error: the threshold $\gamma$ directly determines the error bound (Lemma B.5). In contrast, reweighting methods (e.g., NexCP) depend on tuning $\rho$, whose effect on coverage is indirect and sensitive to drift.
>
>     Overall, KS-based binning is more stable and theoretically transparent than learned retrieval weights.

---

> > ### Author Rebuttal · Reviewer_mR1N · 2026-04-04
> >
> > The authors answered my question well, and I considered raising the score to 4.

---

> > > ### Author Response · Authors · 2026-04-05
> > >
> > > We extend our gratitude to Reviewer **mR1N** for raising the score.

---

### Decision · Program_Chairs · 2026-04-30

**Decision:**

Accept (regular)

**Comment:**

This paper introduces a novel sequential conformal prediction method (DistMatch) that leverages a Kolmogorov-Smirnov (KS) statistic-based binary tree to group residual patches. By enforcing approximate exchangeability through discrete binning rather than continuous reweighting, the authors provide a highly effective, theoretically grounded, and robust solution for uncertainty quantification in non-stationary time series.

The initial reviews were generally positive, praising the originality of the KS-based binning approach and the strong empirical results. However, the reviewers raised a few critical questions that needed to be addressed before acceptance. The primary concerns revolved around the theoretical justification for the $\beta$-mixing condition (Assumption 5.2), the potential computational burden of the tree structure during long online deployments, and the sensitivity to hyperparameters like the patch size.

The authors provided an exceptional and thorough rebuttal that directly resolved these issues. To address the theoretical gaps, they explicitly connected the $\beta$-mixing condition to the concept of local stationarity and backed this up with convincing non-parametric Total Variation (TV) independence tests. On the practical side, their new ablation studies clearly demonstrated that a fixed tree structure is computationally lightweight and actually outperforms frequent retraining strategies. They also provided sufficient evidence that the method remains stable even with significantly smaller patch sizes.

Following the rebuttal phase and the subsequent discussion, all reviewers acknowledged that their concerns were fully resolved and unanimously supported accepting the paper. The submission is technically solid, well-written, and offers a practical contribution that will be useful to the conformal prediction community.